# Private estimation algorithms for stochastic block models and mixture models

**Hongjie Chen**
ETH Zürich

**Vincent Cohen-Addad**
Google Research

**Tommaso d'Orsi**
Bocconi*

**Alessandro Epasto**
Google Research

**Jacob Imola**
UC San Diego

**David Steurer**
ETH Zürich

**Stefan Tiegel**
ETH Zürich

## Abstract

We introduce general tools for designing efficient private estimation algorithms, in the high-dimensional settings, whose statistical guarantees almost match those of the best known non-private algorithms. To illustrate our techniques, we consider two problems: recovery of stochastic block models and learning mixtures of spherical Gaussians.

For the former, we present the first efficient $(\varepsilon, \delta)$-differentially private algorithms for both weak recovery and exact recovery. Previously known algorithms achieving comparable guarantees required quasi-polynomial time. We complement these results with an information-theoretic lower bound that highlights how the guarantees of our algorithms are almost tight.

For the latter, we design an $(\varepsilon, \delta)$-differentially private algorithm that recovers the centers of the $k$-mixture when the minimum separation is at least $O(k^{1/t}\sqrt{t})$. For all choices of $t$, this algorithm requires sample complexity $n \geqslant k^{O(1)}d^{O(t)}$ and time complexity $(nd)^{O(t)}$. Prior work required either an additional additive $\Omega(\sqrt{\log n})$ term in the minimum separation or an explicit upper bound on the Euclidean norm of the centers.

## 1  Introduction

Computing a model that best matches a dataset is a fundamental question in machine learning and statistics. Given a set of $n$ samples from a model, how to find the most likely parameters of the model that could have generated this data? This basic question has been widely studied for several decades, and recently revisited in the context where the input data has been partially corrupted (i.e., where few samples of the data have been adversarially generated—see for instance [37, 18, 22, 20]). This has led to several recent works shedding new lights on classic model estimation problems, such as the Stochastic Block Model (SBM) [28, 46, 44, 25, 21, 40] and the Gaussian Mixture Model (GMM) [30, 36, 9, 10] (see Definitions 1.1 and 1.2).

Privacy in machine learning and statistical tasks has recently become of critical importance. New regulations, renewed consumer interest as well as privacy leaks, have led the major actors to adopt privacy-preserving solutions for the machine learning [1, 2, 3]. This new push has resulted in a flurry of activity in algorithm design for private machine learning, including very recently for SBMs and GMMs [55, 32, 16, 60]. Despite this activity, it has remain an open challenge to fully understand how privacy requirements impact model estimation problems and in particular their recovery thresholds and the computational complexity. This is the problem we tackle in this paper.

---

*Much of this work was done while the author was at ETH Zürich.

37th Conference on Neural Information Processing Systems (NeurIPS 2023).

While other notions of privacy exist (e.g. $k$-anonymity), the de facto privacy standard is the differential privacy (DP) framework of Dwork, McSherry, Nissim, and Smith [24]. In this framework, the privacy quality is governed by two parameters, $\varepsilon$ and $\delta$, which in essence tell us how the probability of seeing a given output changes (both multiplicatively and additively) between two datasets that differ by any individual data element. This notion, in essence, quantifies the amount of information *leaked* by a given algorithm on individual data elements. The goal of the algorithm designer is to come up with differentially private algorithms for $\varepsilon$ being a small constant and $\delta$ being of order $1/n^{\Theta(1)}$.

Differentially private analysis of graphs usually considers two notions of neighboring graphs. The weaker notion of *edge-DP* defines two graphs to be neighboring if they differ in one edge. Under the stronger notion of *node-DP*, two neighboring graphs can differ arbitrarily in the set of edges connected to a single vertex. Recently, there is a line of work on node-DP parameter estimation in random graph models, e.g. Erdös-Rényi models [61] and Graphons [12, 13]. However, for the more challenging task of graph clustering, node-DP is sometimes impossible to achieve.[2] Thus it is a natural first step to study edge-DP graph clustering.

Very recently, Seif, Nguyen, Vullikanti, and Tandon [55] were the first to propose differentially private algorithms for the Stochastic Block Model, with edge-DP. Concretely, they propose algorithms achieving exact recovery (exact identification of the planted clustering) while preserving privacy of individual edges of the graph. The proposed approach either takes $n^{\Theta(\log n)}$ time when $\varepsilon$ is constant, or runs in polynomial time when $\varepsilon$ is $\Omega(\log n)$.

Gaussian Mixture Models have also been studied in the context of differential privacy by [32, 16, 60] using the subsample-and-aggregate framework first introduced in [52] (see also recent work for robust moment estimation in the differential privacy setting [35, 8, 29]). The works of [32, 16] require an explicit bound on the euclidean norm of the centers as the sample complexity of these algorithms depends on this bound. For a mixture of $k$ Gaussians, if there is a non-private algorithm that requires the minimum distance between the centers to be at least $\Delta$, then [16, 60] can transform this non-private algorithm into a private one that needs the minimum distance between the centers to be at least $\Delta + \sqrt{\log n}$, where $n$ is the number of samples.

In this paper, we tackle both clustering problems (graph clustering with the SBM and metric clustering with the GMM) through a new general privacy-preserving framework that brings us significantly closer to the state-of-the-art of non-private algorithms. As we will see, our new perspective on the problems appear to be easily extendable to many other estimation algorithms.

**From robustness to privacy**    In recent years a large body of work (see [19, 22, 40, 18, 37, 14, 36, 30] and references therein) has advanced our understanding of parameter estimation in the presence of adversarial perturbations. In these settings, an adversary looks at the input instance and modifies it arbitrarily, under some constraints (these constraints are usually meant to ensure that it is still information theoretically possible to recover the underlying structure). As observed in the past [23, 35, 41], the two goals of designing privacy-preserving machine learning models and robust model estimation are tightly related. The common objective is to design algorithms that extract global information without over-relaying on individual data samples.

Concretely, robust parameter estimation tends to morally follow a two-steps process: *(i)* argue that typical inputs are well-behaved, in the sense that they satisfy some property which can be used to accurately infer the desired global information, *(ii)* show that adversarial perturbations cannot significantly alter the quality of well-behaved inputs, so that it is still possible to obtain an accurate estimate. Conceptually, the analysis of private estimation algorithms can also be divided in two parts: *utility*, which is concerned with the accuracy of the output, and *privacy*, which ensures there is no leak of sensitive information. In particular, the canonical differential privacy definition can be interpreted as the requirement that, for any distinct inputs $Y, Y'$, the change in the output is *proportional* to the distance[3] between $Y$ and $Y'$.

It is easy to see this as a generalization of robustness: while robust algorithm needs the output to be stable for typical inputs, private algorithms requires this stability for *any possible input*. Then,

---

[2]In particular, we cannot hope to achieve *exact recovery*. We could isolate a vertex by removing all of its adjacent edges. Then it is impossible to cluster this vertex correctly. See below for formal definitions.

[3]The notion of distance is inherently application dependent. For example, it could be Hamming distance.

stability of the output immediately implies adding a small amount of noise to the output yields privacy. If the added noise is small enough, then utility is also preserved.

Our work further tightens this connection between robustness and privacy through a simple yet crucial insight: if two strongly convex functions over constrained sets –where both the function and the set may depend on the input– are point-wise close (say in a $\ell_2$-sense), their minimizers are also close (in the same sense). The alternative perspective is that projections of points that are close to each other, onto convex sets that are point-wise close, must also be close. This observation subsumes previously known sensitivity bounds in the empirical risk minimization literature (in particular in the output-perturbation approach to ERM, see Section 2 for a comparison).

The result is a clean, user-friendly, *framework to turn robust estimation algorithms into private algorithms*, while keeping virtually the same guarantees. We apply this paradigm to stochastic block models and Gaussian mixture models, which we introduce next.

**Stochastic block model**  The stochastic block model is an extensively studied statistical model for community detection in graphs (see [4] for a survey).

**Model 1.1** (Stochastic block model). *In its most basic form, the stochastic block model describes the distribution[4] of an $n$-vertex graph $\mathbf{G} \sim \mathsf{SBM}_n(d, \gamma, x)$, where $x$ is a vector of $n$ binary[5] labels, $d \in \mathbb{N}$, $\gamma > 0$, and for every pair of distinct vertices $i, j \in [n]$ the edge $\{i, j\}$ is independently added to the graph $\mathbf{G}$ with probability $(1 + \gamma \cdot x_i \cdot x_j)\frac{d}{n}$.*

For balanced label vector $x$, i.e., with roughly the same number of $+1$'s and $-1$'s, parameter $d$ roughly corresponds to the average degree of the graph. Parameter $\gamma$ corresponds to the *bias* introduced by the community structure. Note that for distinct vertices $i, j \in [n]$, the edge $\{i, j\}$ is present in $\mathbf{G}$ with probability $(1 + \gamma)\frac{d}{n}$ if the vertices have the same label $x_i = x_j$ and with probability $(1 - \gamma)\frac{d}{n}$ if the vertices have different labels $x_i \neq x_j$.[6]

Given a graph $\mathbf{G}$ sampled according to this model, the goal is to recover the (unknown) underlying vector of labels as well as possible. In particular, for a chosen algorithm returning a partition $\hat{x}(\mathbf{G}) \in \{\pm 1\}^n$, there are two main objective of interest: *weak recovery* and *exact recovery*. The former amounts to finding a partition $\hat{x}(\mathbf{G})$ correlated with the true partition. The latter instead corresponds to actually recovering the true partition with high probability. As shown in the following table, by now the statistical and computational landscape of these problems is well understood [17, 42, 48, 49, 28]:

| | **Objective** | can be achieved (and efficiently so) *iff* |
|---|---|---|
| *weak recovery* | $\mathbb{P}_{\mathbf{G} \sim \mathsf{SBM}_n(d, \gamma, x)} \left( \frac{1}{n} \lvert \langle x, \hat{x}(\mathbf{G}) \rangle \rvert \geqslant \Omega_{d, \gamma}(1) \right) \geqslant 1 - o(1)$ | $\gamma^2 \cdot d \geqslant 1$ |
| *exact recovery* | $\mathbb{P}_{\mathbf{G} \sim \mathsf{SBM}_n(d, \gamma, x)} \left( \hat{x}(\mathbf{G}) \in \{x, -x\} \right) \geqslant 1 - o(1)$ | $\frac{d}{\log n}\left(1 - \sqrt{1 - \gamma^2}\right) \geqslant 1$ |

**Learning mixtures of spherical Gaussians**  The Gaussian Mixture Model we consider is the following.

**Model 1.2** (Mixtures of spherical Gaussians). *Let $D_1, \ldots, D_k$ be Gaussian distributions on $\mathbb{R}^d$ with covariance $\mathrm{Id}$ and means $\mu_1, \ldots, \mu_k$ satisfying $\lVert \mu_i - \mu_j \rVert \geqslant \Delta$ for any $i \neq j$. Given a set $\mathbf{Y} = \{\mathbf{y}_1, \ldots, \mathbf{y}_n\}$ of $n$ samples from the uniform mixture over $D_1, \ldots, D_k$, estimate $\mu_1, \ldots, \mu_k$.*

It is known that when the minimum separation is $\Delta = o(\sqrt{\log k})$, superpolynomially many samples are required to estimate the means up to small constant error [54]. Just above this threshold, at separation $k^{O(1/\gamma)}$ for any constant $\gamma$, there exist efficient algorithms based on the sum-of-squares hierarchy recovering the means up to accuracy $1/\operatorname{poly}(k)$ [30, 36, 59]. In the regime

---

[4]We use **bold** characters to denote random variables.

[5]More general versions of the stochastic block model allow for more than two labels and general edge probabilities depending on the label assignment. However, many of the algorithmic phenomena of the general version can in their essence already be observed for the basic version that we consider in this work.

[6]At times we may write $d_n$, $\gamma_n$ to emphasize that these may be functions of $n$. We write $o(1), \omega(1)$ for functions tending to zero (resp. infinity) as $n$ grows.

where $\Delta = O(\sqrt{\log k})$ these algorithms yield the same guarantees but require quasipolynomial time. Recently, [39] showed how to efficiently recover the means as long as $\Delta = O(\log(k)^{1/2+c})$ for any constant $c > 0$.

## 1.1 Results

**Stochastic block model**  We present here the first $(\varepsilon, \delta)$-differentially private efficient algorithms for exact recovery. In all our results on stochastic block models, we consider the *edge privacy* model, in which two input graphs are adjacent if they differ on a single edge (cf. Definition C.1).

**Theorem 1.3** (Private exact recovery of SBM). *Let $x \in \{\pm 1\}^n$ be balanced[7]. For any $\gamma, d, \varepsilon, \delta > 0$ satisfying*

$$\frac{d}{\log n}\left(1 - \sqrt{1 - \gamma^2}\right) \geqslant \Omega(1) \quad and \quad \frac{\gamma d}{\log n} \geqslant \Omega\left(\frac{1}{\varepsilon^2} \cdot \frac{\log(1/\delta)}{\log n} + \frac{1}{\varepsilon}\right),$$

*there exists an $(\varepsilon, \delta)$-differentially edge private algorithm that, on input $\mathbf{G} \sim \mathsf{SBM}_n(d, \gamma, x)$, returns $\hat{x}(\mathbf{G}) \in \{x, -x\}$ with probability $1 - o(1)$. Moreover, the algorithm runs in polynomial time.*

For any constant $\varepsilon > 0$, Theorem 1.3 states that $(\varepsilon, \delta)$-differentially private exact recovery is possible, in polynomial time, already a constant factor close to the non-private threshold. Previous results [55] could only achieve comparable guarantees in time $O(n^{O(\log n)})$. It is also important to observe that the theorem provides a trade-off between signal-to-noise ratio of the instance (captured by the expression on the left-hand side with $\gamma, d$) and the privacy parameter $\varepsilon$. In particular, we highlight two regimes: for $d \geqslant \Omega(\log n)$ one can achieve exact recovery with high probability and privacy parameters $\delta = n^{-\Omega(1)}, \varepsilon = O(1/\gamma + 1/\gamma^2)$. For $d \geqslant \omega(\log n)$ one can achieve exact recovery with high probability and privacy parameters $\varepsilon = o(1), \delta = n^{-\omega(1)}$. Theorem 1.3 follows by a result for private weak recovery and a boosting argument (cf. Theorem C.3 and Appendix C.2).

Further, we present a second, exponential-time, algorithm based on the exponential mechanism [43] which improves over the above in two regards. First, it gives *pure* differential privacy. Second, it provides utility guarantees for a larger range of graph parameters. In fact, we will also prove a lower bound which shows that its privacy guarantees are information theoretically optimal.[8] All hidden constants are absolute and do not depend on any graph or privacy parameters unless stated otherwise. In what follows we denote by $\mathrm{err}(\hat{x}, x)$ the minimum of the hamming distance of $\hat{x}$ and $x$, and the one of $-\hat{x}$ and $x$, divided by $n$.

**Theorem 1.4** (Slightly informal, see Theorem C.18 in the supplements for full version). *Let $\gamma\sqrt{d} \geqslant \Omega(1)$, $x \in \{\pm 1\}^n$ be balanced, and $\zeta \geqslant \exp\left(-\Omega\left(\gamma^2 d\right)\right)$. For any $\varepsilon \geqslant \Omega\left(\frac{\log(1/\zeta)}{\gamma d}\right)$, there exists an algorithm which on input $\mathbf{G} \sim \mathsf{SBM}_n(\gamma, d, x)$ outputs an estimate $\hat{x}(\mathbf{G}) \in \{\pm 1\}^n$ satisfying $\mathrm{err}(\hat{x}(\mathbf{G}), x) \leqslant \zeta$ with probability at least $1 - \zeta$. In addition, the algorithm is $\varepsilon$-differentially edge private. Further, we can achieve error $\Theta\left(1/\sqrt{\log(1/\zeta)}\right)$ with probability $1 - e^{-n}$.*

A couple of remarks are in order. First, our algorithm works across all degree-regimes in the literature and matches known non-private thresholds and rates up to constants.[9] In particular, for $\gamma^2 d = \Theta(1)$, we achieve weak/partial recovery with either constant or exponentially high success probability. Recall that the optimal non-private threshold is $\gamma^2 d > 1$. For the regime, where $\gamma^2 d = \omega(1)$, it is known that the optimal error rate is $\exp(-(1 - o(1))\gamma^2 d)$ [64] even non-privately which we match up to constants - here $o(1)$ denotes a function that tends to zero as $\gamma^2 d$ tends to infinity. Moreover, our algorithm achieves exact recovery as soon as $\gamma^2 d = \Omega(\log n)$ since then $\zeta < \frac{1}{n}$. This also matches known non-private threshholds up to constants [5, 47]. We remark that [55] gave an $\varepsilon$-DP exponential time algorithm which achieved exact recovery and has inverse polynomial success probability in the utility case as long as $\varepsilon \geqslant \Omega(\frac{\log n}{\gamma d})$. We recover this result as a special case (with slightly worse constants). In fact, their algorithm is also based on the exponential mechanism, but their analysis only applies to the setting of exact recovery, while our result holds much more generally. Another

---

[7]A vector $x \in \{\pm 1\}^n$ is said to be balanced if $\sum_{i=1}^n x_i = 0$.

[8]It is optimal in the "small error" regime, otherwise it is almost optimal. See Theorem C.22 for more detail.

[9]For ease of exposition we did not try to optimize these constants.

crucial difference is that we show how to privatize a known boosting technique frequently used in the non-private setting, allowing us to achieve error guarantees which are optimal up to constant factors.

It is natural to ask whether, for a given set of parameters $\gamma, d, \zeta$ one can obtain better privacy guarantees than Theorem 1.4. Our next result implies that our algorithmic guarantees are almost tight.

**Theorem 1.5** (Informal, see Theorem C.22 in the supplements for full version). *Suppose there exists an $\varepsilon$-differentially edge private algorithm such that for any balanced $x \in \{\pm 1\}^n$, on input $\mathbf{G} \sim \mathsf{SBM}_n(d, \gamma, x)$, outputs $\hat{x}(\mathbf{G}) \in \{\pm 1\}^n$ satisfying*

$$\mathbb{P}\left(\mathrm{err}(\hat{x}(\mathbf{G}), x) < \zeta\right) \geqslant 1 - \eta\,.$$

*Then,*

$$\varepsilon \geqslant \Omega\left(\frac{\log(1/\zeta)}{\gamma d} + \frac{\log(1/\eta)}{\zeta n \gamma d}\right)\,. \tag{1.1}$$

This lower bound is tight for $\varepsilon$-DP exact recovery. By setting $\zeta = 1/n$ and $\eta = 1/\mathrm{poly}(n)$, Theorem 1.5 implies no $\varepsilon$-DP exact recovery algorithm exists for $\varepsilon \leqslant O(\frac{\log n}{\gamma d})$. There exist $\varepsilon$-DP algorithms (Algorithm C.19 in the supplements and the algorithm in [55]) exactly recover the community for any $\varepsilon \geqslant \Omega(\frac{\log n}{\gamma d})$.

Notice Theorem 1.5 is a lower bound for a large range of error rates (partial to exact recovery). For failure probability $\eta = \zeta$, the lower bound simplifies to $\varepsilon \geqslant \Omega(\frac{\log(1/\zeta)}{\gamma d})$ and hence matches Theorem 1.4 up to constants. For exponentially small failure probability, $\eta = e^{-n}$, it becomes $\varepsilon \geqslant \Omega(\frac{1}{\zeta \gamma d})$. To compare, Theorem 1.4 requires $\varepsilon \geqslant \Omega(\frac{1}{\zeta^2 \gamma d})$ in this regime, using the substitution $\sqrt{\log(1/\zeta)} \to \zeta$.

Further, while formally incomparable, this $\varepsilon$-DP lower bound also suggests that the guarantees obtained by our efficient $(\varepsilon, \delta)$-DP algorithm in Theorem 1.3 might be close to optimal. Note that setting $\zeta = \frac{1}{n}$ in Theorem 1.5 requires the algorithm to exactly recover the partitioning. In this setting, Theorem 1.3 implies that there is an efficient $(\varepsilon, n^{-\Theta(1)})$-DP exact recovery algorithm for $\varepsilon \leqslant O(\sqrt{\frac{\log n}{\gamma d}})$. Theorem 1.5 states any $\varepsilon$-DP exact recovery algorithm requires $\varepsilon \geqslant \Omega(\frac{\log n}{\gamma d})$. Further, for standard privacy parameters that are required for real-world applications, such as $\varepsilon \approx 1$ and $\delta = n^{-10}$, Theorem 1.3 requires that $\gamma d \geqslant \Omega(\log n)$. Theorem 1.5 shows that for pure-DP algorithms with the same setting of $\varepsilon$ this is also necessary. We leave it as fascinating open questions to bridge the gap between upper and lower bounds in the context of $(\varepsilon, \delta)$-DP.

**Learning mixtures of spherical Gaussians** Our algorithm for privately learning mixtures of $k$ spherical Gaussians provides statistical guarantees matching those of the best known non-private algorithms.

**Theorem 1.6** (Privately learning mixtures of spherical Gaussians). *Consider an instance of Model 1.2. Let $t > 0$ be such that $\Delta \geqslant O\left(\sqrt{t} k^{1/t}\right)$. For $n \geqslant \Omega\left(k^{O(1)} \cdot d^{O(t)}\right), k \geqslant (\log n)^{1/5}$, there exists an algorithm, running in time $(nd)^{O(t)}$, that outputs vectors $\hat{\boldsymbol{\mu}}_1, \ldots, \hat{\boldsymbol{\mu}}_k$ satisfying*

$$\max_{\ell \in [k]} \left\|\hat{\boldsymbol{\mu}}_\ell - \mu_{\pi(\ell)}\right\|_2 \leqslant O(k^{-12})\,,$$

*with high probability, for some permutation $\pi : [k] \to [k]$. Moreover, for $\varepsilon \geqslant k^{-10}, \delta \geqslant n^{-10}$, the algorithm is $(\varepsilon, \delta)$-differentially private[10] for any input $Y$.*

The conditions $\varepsilon \geqslant k^{-10}, \delta \geqslant n^{-10}$ in Theorem 1.6 are not restrictive and should be considered a formality. Moreover, setting $\varepsilon = 0.01$ and $\delta = n^{-10}$ already provides meaningful privacy guarantees in practice. The condition that $k \geqslant (\log n)^{1/5}$ is a technical requirement by our proofs.

Prior to this work, known differentially private algorithms could learn a mixture of $k$-spherical Gaussian either if: (1) they were given a ball of radius $R$ containing all centers [32, 16];[11] or (2) the minimum separation between centers needs an additional additive $\Omega(\sqrt{\log n})$ term [16, 60][12].

---

[10]Two input datasets are adjacent if they differ on a single sample. See Definition D.1 in the supplements.

[11]In [32, 16] the sample complexity of the algorithm depends on this radius $R$.

[12]For $k \leqslant n^{o(1)}$ our algorithm provides a significant improvement as $\sqrt{\log k} = o(\sqrt{\log n})$.

To the best of our knowledge, Theorem 1.6 is the first to get the best of both worlds. That is, our algorithm requires no explicit upper bounds on the means (this also means the sample complexity does not depend on $R$) and only minimal separation assumption $O(\sqrt{\log k})$. Furthermore, we remark that while previous results only focused on mixtures of Gaussians, our algorithm also works for the significantly more general class of mixtures of Poincaré distributions. Concretely, in the regime $k \geqslant \sqrt{\log d}$, our algorithm recovers the state-of-the-art guarantees provided by non-private algorithms which are based on the sum-of-squares hierarchy [36, 30, 59]:[13]

- If $\Delta \geqslant k^{1/t^*}$ for some constant $t^*$, then by choosing $t \geqslant \Omega(t^*)$ the algorithm recovers the centers, up to a $1/\operatorname{poly}(k)$ error, in time $\operatorname{poly}(k, d)$ and using only $\operatorname{poly}(k, d)$ samples.
- If $\Delta \geqslant \Omega(\sqrt{\log k})$ then choosing $t = O(\log k)$ the algorithm recovers the centers, up to a $1/\operatorname{poly}(k)$ error, in quasi-polynomial time $\operatorname{poly}(k^{O(t)}, d^{O(t^2)})$ and using a quasi-polynomial number of samples $\operatorname{poly}(k, d^{O(t)})$.

For simplicity of exposition we will limit the presentation to mixtures of spherical Gaussians. We reiterate that separation $\Omega(\sqrt{\log k})$ is information-theoretically necessary for algorithms with polynomial sample complexity [54].

Subsequently and independently of our work, the work of [6] gives an algorithm that turns any non-private GMM learner into a private one based on the subsample and aggregate framework. They apply this reduction to the classical result of [45] to give the first finite-sample $(\varepsilon, \delta)$-DP algorithm that learns mixtures of unbounded Gaussians, in particular, the covariance matrices of their mixture components can be arbitrary.

## 2 Techniques

We present here our general tools for designing efficient private estimation algorithms in the high-dimensional setting whose statistical guarantees almost match those of the best know non-private algorithms. The algorithms we design have the following structure in common: First, we solve a convex optimization problem with constraints and objective function depending on our input $Y$. Second, we round the optimal solution computed in the first step to a solution $X$ for the statistical estimation problem at hand.

We organize our privacy analyses according to this structure. In order to analyze the first step, we prove a simple sensitivity bound for strongly convex optimization problems, which bounds the $\ell_2$-sensitivity of the optimal solution in terms of a uniform sensitivity bound for the objective function and the feasible region of the optimization problem.

For bounded problems –such as recovery of stochastic block models– we use this sensitivity bound, in the second step, to show that introducing small additive noise to standard rounding algorithms is enough to achieve privacy.

For unbounded problems –such as learning GMMs– we use this sensitivity bound to show that on adjacent inputs, either most entries of $X$ only change slightly, as in the bounded case, or few entries vary significantly. We then combine different privacy techniques to hide both type of changes.

**Privacy from sensitivity of strongly convex optimization problems**  Before illustrating our techniques with some examples, it is instructive to explicit our framework. Here we have a set of inputs $\mathcal{Y}$ and a family of strongly convex functions $\mathcal{F}(\mathcal{Y})$ and convex sets $\mathcal{K}(\mathcal{Y})$ parametrized by these inputs. The generic *non-private* algorithm based on convex optimization we consider works as follows:

1. Compute $\hat{X} := \operatorname{argmin}_{X \in \mathcal{K}(Y)} f_Y(X)$;
2. Round $\hat{X}$ into an integral solution.

For an estimation problem, a distributional assumption on $\mathcal{Y}$ is made. Then one shows how, for typical inputs $\mathbf{Y}$ sampled according to that distribution, the above scheme recovers the desired structured information.

---

[13]We remark that [39] give a polynomial time algorithm for separation $\Omega(\log(k)^{1/2+c})$ for constant $c > 0$ in the non-private setting but for a less general class of mixture distributions.

We can provide a privatized version of this scheme by arguing that, under reasonable assumptions on $\mathcal{F}(\mathcal{Y})$ and $\mathcal{K}(\mathcal{Y})$, the output of the function $\mathrm{argmin}_{X \in \mathcal{K}(Y)} f_Y(X)$ has low $\ell_2$-sensitivity. The consequence of this crucial observation is that one can combine the rounding step 2 with some standard privacy mechanism and achieve differential privacy. That is, the second step becomes:

2. Add random noise $\mathbf{N}$ and round $\hat{X} + \mathbf{N}$ into an integral solution.

Our sensitivity bound is simple, yet it generalizes previously known bounds for strongly convex optimization problems (we provide a detailed comparison later in the section). For adjacent $Y, Y' \in \mathcal{Y}$, it requires the following properties:

*(i)* For each $X \in \mathcal{K}(Y) \cap \mathcal{K}(Y')$ it holds $|f_Y(X) - f_{Y'}(X)| \leqslant \alpha$;

*(ii)* For each $X \in \mathcal{K}(Y)$ its projection $Z$ onto $\mathcal{K}(Y) \cap \mathcal{K}(Y')$ satisfies $|f_Y(X) - f_{Y'}(Z)| \leqslant \alpha$.

Here we think of $\alpha$ as some small quantity (relatively to the problem parameters). Notice, we may think of *(i)* as Lipschitz-continuity of the function $g(Y, X) = f_Y(X)$ with respect to $Y$ and of *(ii)* as a bound on the change of the constrained set on adjacent inputs. In fact, these assumptions are enough to conclude low $\ell_2$ sensitivity. Let $\hat{X}$ and $\hat{X}'$ be the outputs of the first step on inputs $Y, Y'$. Then using (i) and (ii) above and the fact that $\hat{X}$ is an optimizer, we can show that there exists $Z \in \mathcal{K}(Y) \cap \mathcal{K}(Y')$ such that

$$|f_Y(\hat{X}) - f_Y(Z)| + |f_{Y'}(\hat{X}') - f_{Y'}(Z)| \leqslant O(\alpha).$$

By $\kappa$-strong convexity of $f_Y$, $f_{Y'}$ this implies

$$\left\| \hat{X} - Z \right\|_2^2 + \left\| \hat{X}' - Z \right\|_2^2 \leqslant O(\alpha/\kappa)$$

which ultimately means $\|\hat{X} - \hat{X}'\|_2^2 \leqslant O(\alpha/\kappa)$ (see Lemma B.1). Thus, starting from our assumptions on the point-wise distance of $f_Y$, $f_{Y'}$ we were able to conclude low $\ell_2$-sensitivity of our output!

**A simple application: weak recovery of stochastic block models** The ideas introduced above, combined with existing algorithms for weak recovery of stochastic block models, immediately imply a private algorithm for the problem. To illustrate this, consider Model 1.1 with parameters $\gamma^2 d \geqslant C$, for some large enough constant $C > 1$. Let $x \in \{\pm 1\}^n$ be balanced. Here $Y$ is an $n$-by-$n$ matrix corresponding to the rescaled centered adjacency matrix of the input graph:

$$Y_{ij} = \begin{cases} \frac{1}{\gamma d}\left(1 - \frac{d}{n}\right) & \text{if } ij \in E(G) \\ -\frac{1}{\gamma n} & \text{otherwise.} \end{cases}$$

The basic semidefinite program [28, 46] can be recast[14] as the strongly convex constrained optimization question of finding the orthogonal projection of the matrix $Y$ onto the set $\mathcal{K} := \left\{ X \in \mathbb{R}^{n \times n} \mid X \succeq \mathbf{0}, X_{ii} = \frac{1}{n} \; \forall i \right\}$. That is

$$\hat{X} := \mathrm{argmin}_{X \in \mathcal{K}} \|Y - X\|_{\mathrm{F}}^2.$$

Let $f_Y(X) := \|X\|_{\mathrm{F}}^2 - 2\langle X, Y \rangle$ and notice that $\hat{X} = \mathrm{argmin}_{X \in \mathcal{K}} f_Y(X)$. It is a standard fact that, if our input was $\mathbf{G} \sim \mathsf{SBM}_n(d, \gamma, x)$, then with high probability $X(\mathbf{G}) = \mathrm{argmin}_{X \in \mathcal{K}} f_{Y(\mathbf{G})}(X)$ would have leading eigenvalue-eigenvector pair satisfying

$$\lambda_1(\mathbf{G}) \geqslant 1 - O(1/\gamma^2 d) \quad \text{and} \quad \langle v_1(\mathbf{G}), x/\|x\|\rangle^2 \geqslant 1 - O\left(1/\gamma^2 d\right).$$

This problem fits perfectly the description of the previous paragraph. Note that since the constraint set does not depend on $Y$, Property (ii) reduces to Property (i). Thus, it stands to reason that the projections $\hat{X}, \hat{X}'$ of $Y, Y'$ are close whenever the input graphs generating $Y$ and $Y'$ are adjacent. By Hölder's Inequality with the entry-wise infinity and $\ell_1$-norm, we obtain $|f_Y(X) - f_{Y'}(X)| \leqslant 2 \|X\|_\infty \|Y - Y'\|_1$. By standard facts about positive semidefinite matrices, we have $\|X\|_\infty \leqslant \frac{1}{n}$

---

[14]The objective function in [28, 46] is linear in $X$ instead of quadratic. However, both programs have similar utility guarantees and the utility proof of our program is an adaption of that in [28] (see Lemma C.8). We use the quadratic objective function to achieve privacy via strong convexity.

for all $X \in \mathcal{K}$. Also, $Y$ and $Y'$ can differ on at most 2 entries and hence $\|Y - Y'\|_1 \leqslant O(\frac{1}{\gamma d})$. Thus, $\|\hat{X} - \hat{X}'\|_F^2 \leqslant O(\frac{1}{n\gamma d})$.

The rounding step is now straightforward. Using the Gaussian mechanism we return the leading eigenvector of $\hat{X} + \mathbf{N}$ where $\mathbf{N} \sim N(0, \frac{1}{n\gamma d} \cdot \frac{\log(1/\delta)}{\varepsilon^2})^{n \times n}$. This matrix has Frobeinus norm significantly larger than $\hat{X}$ but its spectral norm is only

$$\|\mathbf{N}\| \leqslant \frac{\sqrt{n \log(1/\delta)}}{\varepsilon} \cdot \sqrt{\frac{1}{n\gamma d}} \leqslant \frac{1}{\varepsilon} \cdot \sqrt{\frac{\log(1/\delta)}{\gamma d}}.$$

Thus by standard linear algebra, for typical instances $\mathbf{G} \sim \mathsf{SBM}_n(d, \gamma, x)$, the leading eigenvector of $\hat{X}(\mathbf{G}) + \mathbf{N}$ will be highly correlated with the true community vector $x$ whenever the average degree $d$ is large enough. In conclusion, a simple randomized rounding step is enough!

**Remark 2.1** (From weak recovery to exact recovery). *In the non-private setting, given a weak recovery algorithm for the stochastic block model, one can use this as an initial estimate for a boosting procedure based on majority voting to achieve exact recovery. We show that this can be done privately. See Appendix C.2 in the supplements.*

**An advanced application: learning mixtures of Gaussians** In the context of SBMs our argument greatly benefited from two key properties: first, on adjacent inputs $Y - Y'$ was bounded in an appropriate norm; and second, the convex set $\mathcal{K}$ was fixed. In the context of learning mixtures of spherical Gaussians as in Model 1.2, *both* this properties are *not* satisfied (notice how one of this second properties would be satisfied assuming bounded centers!). So additional ideas are required.

The first observation, useful to overcome the first obstacle, is that before finding the centers, one can first find the $n$-by-$n$ membership matrix $W(Y)$ where $W(Y)_{ij} = 1$ if $i, j$ where sampled from the same mixture component and 0 otherwise. The advantage here is that, on adjacent inputs, $\|W(Y) - W(Y')\|_F^2 \leqslant 2n/k$ and thus one recovers the first property.[15] Here early sum-of-squares algorithms for the problem [30, 36] turns out to be convenient as they rely on minimizing the function $\|W\|_F^2$ subject to the following system of polynomial inequalities in variables $z_{11}, \ldots, z_{1k}, \ldots, z_{nk}$, with $W_{ij} = \sum_\ell z_{i\ell} z_{j\ell}$ for all $i, j \in [n]$ and a parameter $t > 0$.

$$
\begin{cases}
z_{i\ell}^2 = z_{i\ell} & \forall i \in [n], \ell \in [k] \quad \text{(indicators)} \\[2mm]
\sum_{\ell \in [k]} z_{i\ell} \leqslant 1 & \forall i \in [n] \quad \text{(cluster membership)} \\[2mm]
z_{i\ell} \cdot z_{i\ell'} = 0 & \forall i \in [n], \ell \in [k] \quad \text{(unique membership)} \\[2mm]
\sum_i z_{i\ell} = n/k & \forall \ell \in [k] \quad \text{(size of clusters)}^{16} \\[2mm]
\mu_\ell' = \frac{k}{n} \sum_i z_{i\ell} \cdot y_i & \forall \ell \in [k] \quad \text{(means of clusters)} \\[2mm]
\frac{k}{n} \sum_i z_{i\ell} \langle y_i - \mu_\ell', u \rangle^{2t} \leqslant (2t)^t \cdot \|u\|_2^t & \forall u \in \mathbb{R}^d, \ell \in [k] \quad \text{(subgaussianity of $t$-moment)}
\end{cases}
$$
$$(\mathcal{P}(Y))$$

For the scope of this discussion,[17] we may disregard computational issues and assume we have access to an algorithm returning a point from the convex hull $\mathcal{K}(Y)$ of all solutions to our system of inequalities.[18] Each indicator variable $z_{i\ell} \in \{0, 1\}$ is meant to indicate whether sample $y_i$ is believed

---

[15]Notice for typical inputs $\mathbf{Y}$ from Model 1.2 one expect $\|W(\mathbf{Y})\|_F^2 \approx n^2/k$.

[16]Formally, we would replace the constraint on the size of the clusters by one which requires them to be of size $(1 \pm \alpha)\frac{n}{k}$, for some small $\alpha$.

[17]While this is far from being true, it turns out that having access to a pseudo-distribution satisfying $\mathcal{P}(Y)$ is enough for our subsequent argument to work, albeit with some additional technical work required.

[18]We remark that a priori it is also not clear how to encode the subgaussian constraint in a way that we could recover a degree-$t$ pseudo-distribution satisfying $\mathcal{P}(Y)$ in polynomial time. By now this is well understood, we discuss this in Appendix A in the supplements.

to be in cluster $C_\ell$. In the non-private setting, the idea behind the program is that –for typical $\mathbf{Y}$ sampled according to Model 1.2 with minimum separation $\Delta \geqslant k^{1/t}\sqrt{t}$– any solution $W(\mathbf{Y}) \in \mathcal{K}(\mathbf{Y})$ is close to the ground truth matrix $W^*(\mathbf{Y})$ in Frobenius norm: $\|W(\mathbf{Y}) - W^*(\mathbf{Y})\|_F^2 \leqslant 1/\operatorname{poly}(k)$. Each row $W(\mathbf{Y})_i$ may be seen as inducing a uniform distribution over a subset of $\mathbf{Y}$.[19] Combining the above bound with the fact that subgaussian distributions at small total variation distance have means that are close, we conclude the algorithm recovers the centers of the mixture.

While this program suggests a path to recover the first property, it also possesses a fatal flaw: the projection $W'$ of $W \in \mathcal{K}(Y)$ onto $\mathcal{K}(Y) \cap \mathcal{K}(Y')$ may be *far* in the sense that $|\|W\|_F^2 - \|W'\|_F^2| \geqslant \Omega(\|W\|_F^2 + \|W'\|_F^2) \geqslant \Omega(n^2/k)$. The reason behind this phenomenon can be found in the constraint $\sum_i z_{i\ell} = n/k$. The set indicated by the vector $(z_{1\ell} \ldots, z_{n\ell})$ may be subgaussian in the sense of $\mathcal{P}(Y)$ for input $Y$ but, upon changing a single sample, this may no longer be true. We work around this obstacle in two steps:

1. We replace the above constraint with $\sum_i z_{i\ell} \leqslant n/k$.

2. We compute $\hat{W} := \operatorname{argmin}_{W \text{ solving } \mathcal{P}(Y)} \|J - W\|_F^2$, where $J$ is the all-one matrix.[20]

The catch now is that the program is satisfiable for *any* input $Y$ since we can set $z_{il} = 0$ whenever necessary. Moreover, we can guarantee property *(ii)* (required by our sensitivity argument) for $\alpha \leqslant O(n/k)$, since we can obtain $W' \in \mathcal{K}(Y) \cap \mathcal{K}(Y')$ simply zeroing out the row/column in $W$ corresponding to the sample differing in $Y$ and $Y'$. Then for typical inputs $\mathbf{Y}$, the correlation with the true solution is guaranteed by the new strongly convex objective function.

We offer some more intuition on the choice of our objective function: Recall that $W_{ij}$ indicates our guess whether the $i$-th and $j$-th datapoints are sampled from the same Gaussian component. A necessary condition for $W$ to be close to its ground-truth counterpart $W^*$, is that they roughly have the same number of entries that are (close to) 1. One way to achieve this would be to add the lower bound constraing $\sum_\ell z_{i\ell} \gtrsim \frac{n}{k}$. However, such a constraint could cause privacy issues: There would be two neighboring datasets, such that the constraint set induced by one dataset is satisfiable, but the constraint set induced by the other dataset is not satisfiable. We avoid this issue by noticing that the appropriate number of entries close to 1 can also be induced by minimizing the distance of $W$ to the all-one matrix. This step is also a key difference from [35], explained in more detail below.

**From low sensitivity of the indicators to low sensitivity of the estimates**   For adjacent inputs $Y, Y'$ let $\hat{W}, \hat{W}'$ be respectively the matrices computed by the above strongly convex programs. Our discussion implies that, applying our sensitivity bound, we can show $\|\hat{W} - \hat{W}'\|_F^2 \leqslant O(n/k)$. The problem is that simply applying a randomized rounding approach here cannot work. The reason is that even tough the vector $\hat{W}_i$ induces a subgaussian distribution, the vector $\hat{W}_i + v$ for $v \in \mathbb{R}^n$, *might not*. Without the subgaussian constraint we cannot provide any meaningful utility bound. In other words, the root of our problem is that there exists heavy-tailed distributions that are arbitrarily close in total variation distance to any given subgaussian distribution.

On the other hand, our sensitivity bound implies $\|\hat{W} - \hat{W}'\|_1^2 \leqslant o(\|\hat{W}\|_1)$ and thus, all but a vanishing fraction of rows $i \in [n]$ must satisfy $\|\hat{W}_i - \hat{W}'_i\|_1 \leqslant o(\|\hat{W}_i\|_1)$. For each row $i$, let $\mu_i, \mu'_i$ be the means of the distributions induced respectively by $\hat{W}_i, \hat{W}'_i$. We are thus in the following setting:

1. For a set of $(1 - o(1)) \cdot n$ good rows $\|\mu_i - \mu'_i\|_2 \leqslant o(1)$,

2. For the set $\mathcal{B}$ of remaining bad rows, the distance $\|\mu_i - \mu'_i\|_2$ may be unbounded.

We hide differences of the first type as follows: pick a random subsample $\mathcal{S}$ of $[n]$ of size $n^c$, for some small $c > 0$, and for each picked row use the Gaussian mechanism. The subsampling step is useful as it allows us to decrease the standard deviation of the entry-wise random noise by a factor $n^{1-c}$. We hide differences of the second type as follows: Note that most of the rows are clustered together in space. Hence, we aim to privately identify the regions which contain many of the rows.

---

[19]More generally, we may think of a vector $v \in \mathbb{R}^n$ as the vector inducing the distribution given by $v/\|v\|_1$ onto the set $Y$ of $n$ elements.

[20]We remark that for technical reasons our function in Appendix D.1 in the supplements will be slightly different. We do not discuss it here to avoid obfuscating our main message.

Formally, we use a classic high dimensional $(\varepsilon, \delta)$-differentially private histogram learner on $\mathcal{S}$ and for the $k$ largest bins of highest count privately return their average (cf. Lemma A.13). The crux of the argument here is that the cardinality of $\mathcal{B} \cap \mathcal{S}$ is sufficiently small that the privacy guarantees of the histogram learner can be extended even for inputs that differ in $|\mathcal{B} \cap \mathcal{S}|$ many samples. Finally, standard composition arguments will guarantee privacy of the whole algorithm.

**Comparison with the framework of Kothari-Manurangsi-Velingker**  Both Kothari-Manurangsi-Velingker [35] and our work obtained private algorithms for high-dimensional statistical estimation problems by privatizing strongly convex programs, more specifically, sum-of-squares (SoS) programs. The main difference between KMV and our work lies in how we choose the SoS program. For the problem of robust moment estimation, KMV considered the canonical SoS program from [30, 36] which contains a minimum cardinality constraint (e.g., $\sum_l z_{il} \gtrsim \frac{n}{k}$ in the case of GMMs). Such a constraint is used to ensure good utility. However, as alluded to earlier, this is problematic for privacy: there will always exist two adjacent input datasets such that the constraints are satisfiable for one but not for the other. KMV and us resolve this privacy issue in different ways.

KMV uses an exponential mechanism to pick the lower bound of the minimum cardinality constraint. This step also ensures that solutions to the resulting SoS program will have low sensitivity. In contrast, we simply drop the minimum cardinality constraint. Then the resulting SoS program is always feasible for any input dataset! To still ensure good utility, we additionally pick an appropriate objective function. For example, in Gaussian mixture models, we chose the objective $\|W - J\|_F^2$. Our approach has the following advantages: First, the exponential mechanism in KMV requires computing $O(n)$ scores. Computing each score requires solving a large semidefinite program, which can significantly increase the running time. Second, proving that the exponential mechanism in KMV works requires several steps: 1) defining a (clever) score function, 2) bounding the sensitivity of this score function and, 3) showing existence of a large range of parameters with high score. Our approach bypasses both of these issues.

Further, as we show, our general recipe can be easily extended to other high dimensional problems of interest: construct a strongly convex optimization program and add noise to its solution. This can provide significant computational improvements. For example, in the context of SBMs, the framework of [35] would require one to sample from an exponential distribution over matrices. Constructing and sampling from such distributions is an expensive operation. However, it is well-understood that an optimal fractional solution to the basic SDP relaxation we consider can be found in *near quadratic time* using the standard matrix multiplicative weight method [7, 58], making the whole algorithm run in near-quadratic time. Whether our algorithm can be sped up to near-linear time, as in [7, 58], remains a fascinating open question.

**Comparison with previous works on empirical risk minimization**  Results along the lines of the sensitivity bound described at the beginning of the section (see Lemma B.1 for a formal statement) have been extensively used in the context of empirical risk minimization [15, 34, 57, 11, 63, 50]. Most results focus on the special case of unconstrained optimization of strongly convex functions. In contrast, our sensitivity bound applies to the significantly more general settings where both the objective functions and the constrained set may depend on the input.[21] Most notably for our settings of interest, [15] studied unconstrained optimization of (smooth) strongly convex functions depending on the input, with bounded gradient. We recover such a result for $X' = X$ in *(ii)*. In [50], the authors considered constraint optimization of objective functions where the domain (but *not* the function) may depend on the input data. They showed how one can achieve differential privacy while optimize the desired objective function by randomly perturbing the constraints. It is important to remark that, in [50], the notion of utility is based on the optimization problem (and their guarantees are tight only up to logarithmic factors). In the settings we consider, even in the special case where $f$ does not depend on the input, this notion of utility may not correspond to the notion of utility required by the estimation problem, and thus, the corresponding guarantees can turn out to be too loose to ensure the desired error bounds.

---

[21]The attentive reader may argue that one could cast convex optimization over a constrained domain as unconstrained optimization of a new convex function with the appropriate penalty terms. In practice however, this turns out to be hard to do for constraints such as Definition A.19.

## Acknowledgments and Disclosure of Funding

This project has received funding from the European Research Council (ERC) under the European Union's Horizon 2020 research and innovation programme (grant agreement No 815464).

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

# A  Preliminaries

We use **boldface** characters for random variables. We hide multiplicative factors *logarithmic* in $n$ using the notation $\tilde{O}(\cdot), \tilde{\Omega}(\cdot)$. Similarly, we hide absolute constant multiplicative factors using the standard notation $O(\cdot), \Omega(\cdot), \Theta(\cdot)$. Often times we use the letter $C$ do denote universal constants independent of the parameters at play. We write $o(1), \omega(1)$ for functions tending to zero (resp. infinity) as $n$ grows. We say that an event happens with high probability if this probability is at least $1 - o(1)$. Throughout the paper, when we say "an algorithm runs in time $O(q)$" we mean that the number of basic arithmetic operations involved is $O(q)$. That is, we ignore bit complexity issues.

**Vectors, matrices, tensors**  We use $\mathrm{Id}_n$ to denote the $n$-by-$n$ dimensional identity matrix, $J_n \in \mathbb{R}^{n \times n}$ the all-ones matrix and $\mathbf{0}_n, \mathbf{1}_n \in \mathbb{R}^n$ to denote respectively the zero and the all-ones vectors. When the context is clear we drop the subscript. For matrices $A, B \in \mathbb{R}^{n \times n}$ we write $A \succeq B$ if $A - B$ is positive semidefinite. For a matrix $M$, we denote its eigenvalues by $\lambda_1(M), \dots, \lambda_n(M)$, we simply write $\lambda_i$ when the context is clear. We denote by $\|M\|$ the spectral norm of $M$. We denote by $\mathbb{R}^{d^{\otimes t}}$ the set of real-valued order-$t$ tensors. for a $d \times d$ matrix $M$, we denote by $M^{\otimes t}$ the *$t$-fold Kronecker product* $\underbrace{M \otimes M \otimes \cdots \otimes M}_{t \text{ times}}$. We define the *flattening*, or *vectorization*, of $M$ to be the $d^t$-dimensional vector, whose entries are the entries of $M$ appearing in lexicographic order. With a slight abuse of notation we refer to this flattening with $M$, ambiguities will be clarified form context. We denote by $N\left(0, \sigma^2\right)^{d^{\otimes t}}$ the distribution over Gaussian tensors with $d^t$ entries with standard deviation $\sigma$. Given $u, v \in \{\pm 1\}^n$, we use $\mathrm{Ham}(u, v) := \sum_{i=1}^n \mathbb{1}_{[u_i \neq v_i]}$ to denote their Hamming distance. Given a vector $u \in \mathbb{R}^n$, we let $\mathrm{sign}(u) \in \{\pm 1\}^n$ denote its sign vector. A vector $u \in \{\pm 1\}^n$ is said to be *balanced* if $\sum_{i=1}^n u_i = 0$.

**Graphs**  We consider graphs on $n$ vertices and let $\mathcal{G}_n$ be the set of all graphs on $n$ vertices. For a graph $G$ on $n$ vertices we denote by $A(G) \in \mathbb{R}^{n \times n}$ its adjacency matrix. When the context is clear we simply write $A$. Let $V(G)$ (resp. $E(G)$) denote the vertex (resp. edge) set of graph $G$. Given two graphs $G, H$ on the same vertex set $V$, let $G \setminus H := (V, E(G) \setminus H(G))$. Given a graph $H$, $H' \subseteq H$ means $H'$ is a subgraph of $H$ such that $V(H') = V(H)$ and $E(H) \subseteq E(H)$. The Hamming distance between two graphs $G, H$ is defined to be the size of the symmetric difference between their edge sets, i.e. $\mathrm{Ham}(G, H) := |E(G) \triangle E(H)|$.

## A.1  Differential privacy

In this section we introduce standard notions of differential privacy [24].

**Definition A.1** (Differential privacy). *An algorithm $\mathcal{M} : \mathcal{Y} \to \mathcal{O}$ is said to be $(\varepsilon, \delta)$-differentially private for $\varepsilon, \delta > 0$ if and only if, for every $S \subseteq \mathcal{O}$ and every neighboring datasets $Y, Y' \in \mathcal{Y}$ we have*

$$\mathbb{P}\left[\mathcal{M}(Y) \in S\right] \leqslant e^\varepsilon \cdot \mathbb{P}\left[\mathcal{M}(Y') \in S\right] + \delta.$$

To avoid confusion, for each problem we will exactly state the relevant notion of neighboring datasets. Differential privacy is closed under post-processing and composition.

**Lemma A.2** (Post-processing). *If $\mathcal{M} : \mathcal{Y} \to \mathcal{O}$ is an $(\varepsilon, \delta)$-differentially private algorithm and $\mathcal{M}' : \mathcal{Y} \to \mathcal{Z}$ is any randomized function. Then the algorithm $\mathcal{M}'(\mathcal{M}(Y))$ is $(\varepsilon, \delta)$-differentially private.*

In order to talk about composition it is convenient to also consider DP algorithms whose privacy guarantee holds only against subsets of inputs.

**Definition A.3** (Differential Privacy Under Condition). *An algorithm $\mathcal{M} : \mathcal{Y} \to \mathcal{O}$ is said to be $(\varepsilon, \delta)$-differentially private under condition $\Psi$ (or $(\varepsilon, \delta)$-DP under condition $\Psi$) for $\varepsilon, \delta > 0$ if and only if, for every $S \subseteq \mathcal{O}$ and every neighboring datasets $Y, Y' \in \mathcal{Y}$ both satisfying $\Psi$ we have*

$$\mathbb{P}\left[\mathcal{M}(Y) \in S\right] \leqslant e^\varepsilon \cdot \mathbb{P}\left[\mathcal{M}(Y') \in S\right] + \delta.$$

It is not hard to see that the following composition theorem holds for privacy under condition.

**Lemma A.4** (Composition for Algorithm with Halting, [35]). *Let $\mathcal{M}_1 : \mathcal{Y} \to \mathcal{O}_1 \cup \{\perp\}$ , $\mathcal{M}_2 : \mathcal{O}_1 \times \mathcal{Y} \to \mathcal{O}_2 \cup \{\perp\}$ , ..., $\mathcal{M}_t : \mathcal{O}_{t-1} \times \mathcal{Y} \to \mathcal{O}_t \cup \{\perp\}$ be algorithms. Furthermore, let $\mathcal{M}$ denote the algorithm that proceeds as follows (with $\mathcal{O}_0$ being empty): For $i = 1 \ldots, t$ compute $o_i = \mathcal{M}_i(o_{i-1}, Y)$ and, if $o_i = \perp$, halt and output $\perp$. Finally, if the algorithm has not halted, then output $o_t$. Suppose that:*

- *For any $1 \leqslant i \leqslant t$, we say that $Y$ satisfies the condition $\Psi_i$ if running the algorithm on $Y$ does not result in halting after applying $\mathcal{M}_1, \ldots, \mathcal{M}_i$.*

- *$\mathcal{M}_1$ is $(\varepsilon_1, \delta_1)$-DP.*

- *$\mathcal{M}_i$ is $(\varepsilon_i, \delta_i)$-DP (with respect to neighboring datasets in the second argument) under condition $\Psi_{i-1}$ for all $i = \{2, \ldots, t\}$ .*

*Then $\mathcal{M}$ is $(\sum_i \varepsilon_i, \sum_i \delta_i)$-DP.*

### A.1.1 Basic differential privacy mechanisms

The Gaussian and the Laplace mechanism are among the most widely used mechanisms in differential privacy. They work by adding a noise drawn from the Gaussian (respectively Laplace) distribution to the output of the function one wants to privatize. The magnitude of the noise depends on the sensitivity of the function.

**Definition A.5** (Sensitivity of function). *Let $f : \mathcal{Y} \to \mathbb{R}^d$ be a function, its $\ell_1$-sensitivity and $\ell_2$-sensitivity are respectively*

$$\Delta_{f,1} := \max_{\substack{Y, Y' \in \mathcal{Y} \\ Y, Y' \text{ are adjacent}}} \|f(Y) - f(Y')\|_1 \qquad \Delta_{f,2} := \max_{\substack{Y, Y' \in \mathcal{Y} \\ Y, Y' \text{ are adjacent}}} \|f(Y) - f(Y')\|_2 .$$

For function with bounded $\ell_1$-sensitivity the Laplace mechanism is often the tool of choice to achieve privacy.

**Definition A.6** (Laplace distribution). *The Laplace distribution with mean $\mu$ and parameter $b > 0$, denoted by $Lap(\mu, b)$, has PDF $\frac{1}{2b} e^{-|x-\mu|/b}$ . Let $\mathrm{Lap}(b)$ denote $\mathrm{Lap}(0, b)$.*

A standard tail bound concerning the Laplace distribution will be useful throughout the paper.

**Fact A.7** (Laplace tail bound). *Let $\boldsymbol{x} \sim Lap(\mu, b)$. Then,*

$$\mathbb{P}\left[|\boldsymbol{x} - \mu| > t\right] \leqslant e^{-t/b} .$$

The Laplace distribution is useful for the following mechanism

**Lemma A.8** (Laplace mechanism). *Let $f : \mathcal{Y} \to \mathbb{R}^d$ be any function with $\ell_1$-sensitivity at most $\Delta_{f,1}$. Then the algorithm that adds $Lap\left(\frac{\Delta_{f,1}}{\varepsilon}\right)^{\otimes d}$ to $f$ is $(\varepsilon, 0)$-DP.*

It is also useful to consider the "truncated" version of the Laplace distribution where the noise distribution is shifted and truncated to be non-positive.

**Definition A.9** (Truncated Laplace distribution). *The (negatively) truncated Laplace distribution $w$ with mean $\mu$ and parameter $b$ on $\mathbb{R}$, denoted by $tLap(\mu, b)$, is defined as $Lap(\mu, b)$ conditioned on the value being non-positive.*

**Lemma A.10** (Truncated Laplace mechanism). *Let $f : \mathcal{Y} \to \mathbb{R}$ be any function with $\ell_1$-sensitivity at most $\Delta_{f,1}$. Then the algorithm that adds $tLap\left(-\Delta_{f,1}\left(1 + \frac{\log(1/\delta)}{\varepsilon}\right), \Delta_{f,1}/\varepsilon\right)$ to $f$ is $(\varepsilon, \delta)$-DP.*

The following tail bound is useful when reasoning about truncated Laplace random variables.

**Lemma A.11** (Tail bound truncated Laplace). *Suppose $\mu < 0$ and $b > 0$. Let $\boldsymbol{x} \sim tLap(\mu, b)$. Then, for $y < \mu$ we have that*

$$\mathbb{P}\left[\boldsymbol{x} < y\right] \leqslant \frac{e^{(y-\mu/b)}}{2 - e^{\mu/b}} .$$

In constrast, when the function has bounded $\ell_2$-sensitivity, the Gaussian mechanism provides privacy.

**Lemma A.12** (Gaussian mechanism). *Let $f : \mathcal{Y} \to \mathbb{R}^d$ be any function with $\ell_2$-sensitivity at most $\Delta_{f,2}$. Let $0 < \varepsilon, \delta \leqslant 1$. Then the algorithm that adds $N\left(0, \frac{\Delta_{f,2}^2 \cdot 2 \log(2/\delta)}{\varepsilon^2} \cdot \mathrm{Id}\right)$ to $f$ is $(\varepsilon, \delta)$-DP.*

### A.1.2 Private histograms

Here we present a classical private mechanism to learn a high dimensional histogram.

**Lemma A.13** (High-dimensional private histogram learner, see [33]). *Let $q, b, \varepsilon > 0$ and $0 < \delta < 1/n$. Let $\{I_i\}_{i=-\infty}^{\infty}$ be a partition of $\mathbb{R}$ into intervals of length $b$, where $I_i := \{x \in \mathbb{R} \mid q + (i-1) \cdot b \leqslant x < q + i \cdot b\}$. Consider the partition of $\mathbb{R}^d$ into sets $\{B_{i_1,\ldots,i_d}\}_{i_1,\ldots,i_d=1}^{\infty}$ where*

$$B_{i_1,\ldots,i_d} := \{x \in \mathbb{R}^d \mid \forall j \in [d], x_j \in I_{i_j}\}$$

*Let $Y = \{y_1, \ldots, y_n\} \subseteq \mathbb{R}^d$ be a dataset of $n$ points. For each $B_{i_1,\ldots,i_d}$, let $p_{i_1,\ldots,i_d} = \frac{1}{n} |\{j \in [n] \mid y_j \in B_{i_1,\ldots,i_d}\}|$. For $n \geqslant \frac{8}{\varepsilon\alpha} \cdot \log \frac{2}{\delta\beta}$, there exists an efficient $(\varepsilon, \delta)$-differentially private algorithm that returns $\hat{\boldsymbol{p}}_{1,\ldots,1}, \ldots, \hat{\boldsymbol{p}}_{i_1,\ldots,i_d}, \ldots$ satisfying*

$$\mathbb{P}\left[\max_{i_1,\ldots,i_d \in \mathbb{N}} |p_{i_1,\ldots,i_d} - \hat{\boldsymbol{p}}_{i_1,\ldots,i_d}| \geqslant \alpha\right] \leqslant \beta.$$

*Proof.* We consider the following algorithm, applied to each $i_1, \ldots, i_d \in \mathbb{N}$ on input $Y$:

1. If $p_{i_1,\ldots,i_d} = 0$ set $\hat{\boldsymbol{p}}_{i_1,\ldots,i_d} = 0$, otherwise let $\hat{\boldsymbol{p}}_{i_1,\ldots,i_d} = p_{i_1,\ldots,i_d} + \tau$ where $\tau \sim \text{Lap}\left(0, \frac{2}{n\varepsilon}\right)$.

2. If $\hat{\boldsymbol{p}}_{i_1,\ldots,i_d} \leqslant \frac{3\log(2/\delta)}{\varepsilon n}$ set $\hat{\boldsymbol{p}}_{i_1,\ldots,i_d} = 0$.

First we argue utility. By construction we get $\hat{\boldsymbol{p}}_{i_1,\ldots,i_d} = 0$ whenever $p_{i_1,\ldots,i_d} = 0$, thus we may focus on non-zero $p_{i_1,\ldots,i_d}$. There are at most $n$ non zero $p_{i_1,\ldots,i_d}$. By choice of $n, \delta$ and by Fact A.7 the maximum over $n$ independent trials $\tau \sim \text{Lap}\left(0, \frac{2}{n\varepsilon}\right)$ is bounded by $\alpha$ in absolute value with probability at least $\beta$.

It remains to argue privacy. Let $Y = \{y_1, \ldots, y_n\}, Y' = \{y_1', \ldots, y_n'\}$ be adjacent datasets. For $i_1, \ldots, i_d \in \mathbb{N}$, let

$$p_{i_1,\ldots,i_d} = |\{j \in [n] \mid y_j \in B_{i_1,\ldots,i_d}\}|$$
$$p_{i_1,\ldots,i_d}' = |\{j \in [n] \mid y_j' \in B_{i_1,\ldots,i_d}\}|.$$

Since $Y, Y'$ are adjacent there exists only two set of indices $\mathcal{I} := \{i_1, \ldots, i_d\}$ and $\mathcal{J} := \{j_1, \ldots, j_d\}$ such that $p_{\mathcal{I}} \neq p_{\mathcal{I}}'$ and $p_{\mathcal{J}} \neq p_{\mathcal{J}}'$. Assume without loss of generality $p_{\mathcal{I}} > p_{\mathcal{I}}'$. Then it must be $p_{\mathcal{I}} = p_{\mathcal{I}}' + 1/n$ and $p_{\mathcal{J}} = p_{\mathcal{J}}' - 1/n$. Thus by the standard tail bound on the Laplace distribution in Fact A.7 and by Lemma A.8, we immediately get that the algorithm is $(\varepsilon, \delta)$-differentially private. $\square$

### A.2 Sum-of-squares and pseudo-distributions

We introduce here the sum-of-squares notion necessary for our private algorithm learning mixtures of Gaussians. We remark that these notions are not needed for Appendix C.

Let $w = (w_1, w_2, \ldots, w_n)$ be a tuple of $n$ indeterminates and let $\mathbb{R}[w]$ be the set of polynomials with real coefficients and indeterminates $w, \ldots, w_n$. We say that a polynomial $p \in \mathbb{R}[w]$ is a *sum-of-squares (sos)* if there are polynomials $q_1, \ldots, q_r$ such that $p = q_1^2 + \cdots + q_r^2$.

#### A.2.1 Pseudo-distributions

Pseudo-distributions are generalizations of probability distributions. We can represent a discrete (i.e., finitely supported) probability distribution over $\mathbb{R}^n$ by its probability mass function $D : \mathbb{R}^n \to \mathbb{R}$ such that $D \geqslant 0$ and $\sum_{w \in \text{supp}(D)} D(w) = 1$. Similarly, we can describe a pseudo-distribution by its mass function. Here, we relax the constraint $D \geqslant 0$ and only require that $D$ passes certain low-degree non-negativity tests.

Concretely, a *level-$\ell$ pseudo-distribution* is a finitely-supported function $D : \mathbb{R}^n \to \mathbb{R}$ such that $\sum_w D(w) = 1$ and $\sum_w D(w)f(w)^2 \geqslant 0$ for every polynomial $f$ of degree at most $\ell/2$. (Here, the summations are over the support of $D$.) A straightforward polynomial-interpolation argument shows

that every level-$\infty$-pseudo distribution satisfies $D \geqslant 0$ and is thus an actual probability distribution. We define the *pseudo-expectation* of a function $f$ on $\mathbb{R}^d$ with respect to a pseudo-distribution $D$, denoted $\tilde{\mathbb{E}}_{D(w)} f(w)$, as

$$\tilde{\mathbb{E}}_{D(w)} f(w) = \sum_w D(w) f(w) \ . \tag{A.1}$$

The degree-$\ell$ moment tensor of a pseudo-distribution $D$ is the tensor $\mathbb{E}_{D(w)}(1, w_1, w_2, \ldots, w_n)^{\otimes \ell}$. In particular, the moment tensor has an entry corresponding to the pseudo-expectation of all monomials of degree at most $\ell$ in $w$. The set of all degree-$\ell$ moment tensors of probability distribution is a convex set. Similarly, the set of all degree-$\ell$ moment tensors of degree $d$ pseudo-distributions is also convex. Key to the algorithmic utility of pseudo-distributions is the fact that while there can be no efficient separation oracle for the convex set of all degree-$\ell$ moment tensors of an actual probability distribution, there's a separation oracle running in time $n^{O(\ell)}$ for the convex set of the degree-$\ell$ moment tensors of all level-$\ell$ pseudodistributions.

**Fact A.14** ([56, 53, 51, 38]). *For any $n, \ell \in \mathbb{N}$, the following set has a $n^{O(\ell)}$-time weak separation oracle (in the sense of [27]):*

$$\left\{ \tilde{\mathbb{E}}_{D(w)}(1, w_1, w_2, \ldots, w_n)^{\otimes d} \mid \text{degree-d pseudo-distribution } D \text{ over } \mathbb{R}^n \right\} \ . \tag{A.2}$$

This fact, together with the equivalence of weak separation and optimization [27] allows us to efficiently optimize over pseudo-distributions (approximately)—this algorithm is referred to as the sum-of-squares algorithm.

The *level-$\ell$ sum-of-squares algorithm* optimizes over the space of all level-$\ell$ pseudo-distributions that satisfy a given set of polynomial constraints—we formally define this next.

**Definition A.15** (Constrained pseudo-distributions). *Let $D$ be a level-$\ell$ pseudo-distribution over $\mathbb{R}^n$. Let $\mathcal{A} = \{f_1 \geqslant 0, f_2 \geqslant 0, \ldots, f_m \geqslant 0\}$ be a system of $m$ polynomial inequality constraints. We say that $D$ satisfies the system of constraints $\mathcal{A}$ at degree $r$, denoted $D \models_{\overline{r}} \mathcal{A}$, if for every $S \subseteq [m]$ and every sum-of-squares polynomial $h$ with $\deg h + \sum_{i \in S} \max\{\deg f_i, r\} \leqslant \ell$,*

$$\tilde{\mathbb{E}}_D h \cdot \prod_{i \in S} f_i \geqslant 0 \,.$$

*We write $D \models \mathcal{A}$ (without specifying the degree) if $D \models_{\overline{0}} \mathcal{A}$ holds. Furthermore, we say that $D \models_{\overline{r}} \mathcal{A}$ holds* approximately *if the above inequalities are satisfied up to an error of $2^{-n^\ell} \cdot \|h\| \cdot \prod_{i \in S} \|f_i\|$, where $\|\cdot\|$ denotes the Euclidean norm[22] of the coefficients of a polynomial in the monomial basis.*

We remark that if $D$ is an actual (discrete) probability distribution, then we have $D \models \mathcal{A}$ if and only if $D$ is supported on solutions to the constraints $\mathcal{A}$.

We say that a system $\mathcal{A}$ of polynomial constraints is *explicitly bounded* if it contains a constraint of the form $\{\|w\|^2 \leqslant M\}$. The following fact is a consequence of Fact A.14 and [27],

**Fact A.16** (Efficient Optimization over Pseudo-distributions). *There exists an $(n + m)^{O(\ell)}$-time algorithm that, given any explicitly bounded and satisfiable system[23] $\mathcal{A}$ of $m$ polynomial constraints in $n$ variables, outputs a level-$\ell$ pseudo-distribution that satisfies $\mathcal{A}$ approximately.*

### A.2.2 Sum-of-squares proof

Let $f_1, f_2, \ldots, f_r$ and $g$ be multivariate polynomials in $w$. A *sum-of-squares proof* that the constraints $\{f_1 \geqslant 0, \ldots, f_m \geqslant 0\}$ imply the constraint $\{g \geqslant 0\}$ consists of sum-of-squares polynomials $(p_S)_{S \subseteq [m]}$ such that

$$g = \sum_{S \subseteq [m]} p_S \cdot \Pi_{i \in S} f_i \,. \tag{A.3}$$

---

[22] The choice of norm is not important here because the factor $2^{-n^\ell}$ swamps the effects of choosing another norm.

[23] Here, we assume that the bit complexity of the constraints in $\mathcal{A}$ is $(n + m)^{O(1)}$.

We say that this proof has *degree* $\ell$ if for every set $S \subseteq [m]$, the polynomial $p_S \Pi_{i \in S} f_i$ has degree at most $\ell$. If there is a degree $\ell$ SoS proof that $\{f_i \geqslant 0 \mid i \leqslant r\}$ implies $\{g \geqslant 0\}$, we write:

$$\{f_i \geqslant 0 \mid i \leqslant r\} \,\big|\!\tfrac{}{\ell}\, \{g \geqslant 0\}. \tag{A.4}$$

Sum-of-squares proofs satisfy the following inference rules. For all polynomials $f, g \colon \mathbb{R}^n \to \mathbb{R}$ and for all functions $F \colon \mathbb{R}^n \to \mathbb{R}^m$, $G \colon \mathbb{R}^n \to \mathbb{R}^k$, $H \colon \mathbb{R}^p \to \mathbb{R}^n$ such that each of the coordinates of the outputs are polynomials of the inputs, we have:

$$\frac{\mathcal{A} \,\big|\!\tfrac{}{\ell}\, \{f \geqslant 0, g \geqslant 0\}}{\mathcal{A} \,\big|\!\tfrac{}{\ell}\, \{f + g \geqslant 0\}}, \frac{\mathcal{A} \,\big|\!\tfrac{}{\ell}\, \{f \geqslant 0\}, \mathcal{A} \,\big|\!\tfrac{}{\ell'}\, \{g \geqslant 0\}}{\mathcal{A} \,\big|\!\tfrac{}{\ell + \ell'}\, \{f \cdot g \geqslant 0\}} \qquad \text{(addition and multiplication)}$$

$$\frac{\mathcal{A} \,\big|\!\tfrac{}{\ell}\, \mathcal{B}, \mathcal{B} \,\big|\!\tfrac{}{\ell'}\, C}{\mathcal{A} \,\big|\!\tfrac{}{\ell \cdot \ell'}\, C} \qquad \text{(transitivity)}$$

$$\frac{\{F \geqslant 0\} \,\big|\!\tfrac{}{\ell}\, \{G \geqslant 0\}}{\{F(H) \geqslant 0\} \,\big|\!\tfrac{}{\ell \cdot \deg(H)}\, \{G(H) \geqslant 0\}}. \qquad \text{(substitution)}$$

Low-degree sum-of-squares proofs are sound and complete if we take low-level pseudo-distributions as models.

Concretely, sum-of-squares proofs allow us to deduce properties of pseudo-distributions that satisfy some constraints.

**Fact A.17** (Soundness). *If $D \,\big|\!\tfrac{}{r}\, \mathcal{A}$ for a level-$\ell$ pseudo-distribution $D$ and there exists a sum-of-squares proof $\mathcal{A} \,\big|\!\tfrac{}{r'}\, \mathcal{B}$, then $D \,\big|\!\tfrac{}{r \cdot r' + r'}\, \mathcal{B}$.*

If the pseudo-distribution $D$ satisfies $\mathcal{A}$ only approximately, soundness continues to hold if we require an upper bound on the bit-complexity of the sum-of-squares $\mathcal{A} \,\big|\!\tfrac{}{r'}\, B$ (number of bits required to write down the proof).

In our applications, the bit complexity of all sum of squares proofs will be $n^{O(\ell)}$ (assuming that all numbers in the input have bit complexity $n^{O(1)}$). This bound suffices in order to argue about pseudo-distributions that satisfy polynomial constraints approximately.

The following fact shows that every property of low-level pseudo-distributions can be derived by low-degree sum-of-squares proofs.

**Fact A.18** (Completeness). *Suppose $d \geqslant r' \geqslant r$ and $\mathcal{A}$ is a collection of polynomial constraints with degree at most $r$, and $\mathcal{A} \vdash \{\sum_{i=1}^n w_i^2 \leqslant B\}$ for some finite $B$.*

*Let $\{g \geqslant 0\}$ be a polynomial constraint. If every degree-$d$ pseudo-distribution that satisfies $D \,\big|\!\tfrac{}{r}\, \mathcal{A}$ also satisfies $D \,\big|\!\tfrac{}{r'}\, \{g \geqslant 0\}$, then for every $\varepsilon > 0$, there is a sum-of-squares proof $\mathcal{A} \,\big|\!\tfrac{}{d}\, \{g \geqslant -\varepsilon\}$.*

### A.2.3 Explicitly bounded distributions

We will consider a subset of subgaussian distributions denoted as certifiably subgaussians. Many subgaussians distributions are known to be certifiably subgaussian (see [36]).

**Definition A.19** (Explicitly bounded distribution). *Let $t \in \mathbb{N}$. A distribution $D$ over $\mathbb{R}^d$ with mean $\mu$ is called $2t$-explicitly $\sigma$-bounded if for each even integer $s$ such that $1 \leqslant s \leqslant t$ the following equation has a degree $s$ sum-of-squares proof in the vector variable $u$*

$$\big|\!\tfrac{2s}{u}\, \left\{ \mathop{\mathbb{E}}_{\mathbf{x} \sim D} \langle \mathbf{x} - \mu, u \rangle^{2s} \leqslant (\sigma s)^s \cdot \|u\|_2^{2s} \right\}$$

*Furthermore, we say that $D$ is explicitly bounded if it is $2t$-explicitly $\sigma$-bounded for every $t \in \mathbb{N}$. A finite set $X \subseteq \mathbb{R}^d$ is said to be $2t$-explicitly $\sigma$-bounded if the uniform distribution on $X$ is $2t$-explicitly $\sigma$-bounded.*

Sets that are $2t$-explicitly $\sigma$-bounded with large intersection satisfy certain key properties. Before introducing them we conveniently present the following definition.

**Definition A.20** (Weight vector inducing distribution). *Let $Y$ be a set of size $n$ and let $p \in [0,1]^n$ be a vector satisfying $\|p\|_1 = 1$. We say that $p$ induces the distribution $D$ with support $Y$ if*

$$\mathbb{P}_{\mathbf{y} \sim D}[\mathbf{y} = y_i] = p_i .$$

**Theorem A.21** ([36, 30]). *Let $Y \subseteq \mathbb{R}^d$ be a set of cardinality $n$. Let $p, p' \in [0,1]^n$ be weight vectors satisfying $\|p\|_1 = \|p'\|_1 = 1$ and $\|p - p'\|_1 \leqslant \beta$. Suppose that $p$ (respectively $p'$) induces a $2t$-explicitly $\sigma_1$-bounded (resp. $\sigma_2$) distribution over $Y$ with mean $\mu_{(p)}$ (resp. $\mu_{(p')}$). There exists an absolute constant $\beta^*$ such that, if $\beta \leqslant \beta^*$, then for $\sigma = \sigma_1 + \sigma_2$ :*

$$\left\| \mu_{(p)} - \mu_{(p')} \right\| \leqslant \beta^{1-1/2t} \cdot O\left(\sqrt{\sigma t}\right) .$$

In the context of learning Gaussian mixtures, we will make heavy use of the statement below.

**Theorem A.22** ([36, 30]). *Let $Y$ be a $2t$-explicitly $\sigma$-bounded set of size $n$. Let $p \in \mathbb{R}^n$ be the weight vector inducing the uniform distribution over $Y$. Let $p' \in \mathbb{R}^n$ be a unit vector satisfying $\|p - p'\|_1 \leqslant \beta$ for some $\beta \leqslant \beta^*$ where $\beta^*$ is a small constant. Then $p'$ induces a $2t$-explicitly $(\sigma + O(\beta^{1-1/2t}))$-bounded distribution over $Y$.*

# B   Stability of strongly-convex optimization

In this section, we prove $\ell_2$ sensitivity bounds for the minimizers of a general class of (strongly) convex optimization problems. In particular, we show how to translate a uniform point-wise sensitivity bound for the objective functions into a $\ell_2$ sensitivity bound for the minimizers.

**Lemma B.1** (Stability of strongly-convex optimization). *Let $\mathcal{Y}$ be a set of datasets. Let $\mathcal{K}(\mathcal{Y})$ be a family of closed convex subsets of $\mathbb{R}^m$ parametrized by $Y \in \mathcal{Y}$ and let $\mathcal{F}(\mathcal{Y})$ be a family of functions $f_Y : \mathcal{K}(Y) \to \mathbb{R}$, parametrized by $Y \in \mathcal{Y}$, such that:*

*(i) for adjacent datasets $Y, Y' \in \mathcal{Y}$ and $X \in \mathcal{K}(Y)$ there exist $Z \in \mathcal{K}(Y) \cap \mathcal{K}(Y')$ satisfying $|f_Y(X) - f_{Y'}(Z)| \leqslant \alpha$ and $|f_Y(Z) - f_{Y'}(Z)| \leqslant \alpha$.*

*(ii) $f_Y$ is $\kappa$-strongly convex in $X \in \mathcal{K}(Y)$.*

*Then for $Y, Y' \in \mathcal{Y}$, $\hat{X} := \arg\min_{X \in \mathcal{K}(Y)} f_Y(X)$ and $\hat{X}' := \arg\min_{X' \in \mathcal{K}(Y')} f_{Y'}(X')$, it holds*

$$\left\| \hat{X} - \hat{X}' \right\|_2^2 \leqslant \frac{12\alpha}{\kappa} .$$

*Proof.* Let $Z \in \mathcal{K}(Y) \cap \mathcal{K}(Y')$ be a point such that $\left| f_Y(\hat{X}) - f_{Y'}(Z) \right| \leqslant \alpha$ and $|f_Y(Z) - f_{Y'}(Z)| \leqslant \alpha$. By $\kappa$-strong convexity of $f_Y$ and $f_{Y'}$ (Proposition G.2) it holds

$$\begin{aligned}
\left\| \hat{X} - \hat{X}' \right\|_2^2 &\leqslant 2 \left\| \hat{X} - Z \right\|_2^2 + 2 \left\| Z - \hat{X}' \right\|_2^2 \\
&\leqslant \frac{4}{\kappa} \left( f_Y(Z) - f_Y(\hat{X}) + f_{Y'}(Z) - f_{Y'}(\hat{X}') \right) .
\end{aligned}$$

Suppose w.l.o.g. $f_Y(\hat{X}) \leqslant f_{Y'}(\hat{X}')$, for a symmetric argument works in the other case. Then

$$f_Y(Z) \leqslant f_{Y'}(Z) + \alpha \leqslant f_Y(\hat{X}) + 2\alpha$$

and

$$f_Y(\hat{X}) \leqslant f_{Y'}(\hat{X}') \leqslant f_{Y'}(Z) \leqslant f_Y(\hat{X}) + \alpha .$$

It follows as desired

$$f_Y(Z) - f_Y(\hat{X}) + f_{Y'}(Z) - f_{Y'}(\hat{X}') \leqslant 3\alpha .$$

$\square$

## C  Private recovery for stochastic block models

In this section, we present how to achieve exact recovery in stochastic block models privately and thus prove Theorem 1.3. To this end, we first use the stability of strongly convex optimization (Lemma B.1) to obtain a private weak recovery algorithm in Appendix C.1. Then we show how to privately boost the weak recovery algorithm to achieve exact recovery in Appendix C.2. In Appendix C.4, we complement our algorithmic results by providing an almost tight lower bound on the privacy parameters. We start by defining the relevant notion of adjacent datasets.

**Definition C.1** (Adjacent graphs). *Let $G, G'$ be graphs with vertex set $[n]$. We say that $G, G'$ are adjacent if $|E(G) \triangle E(G')| = 1$.*

**Remark C.2** (Parameters as public information). *We remark that we assume the parameters $n, \gamma, d$ to be public information given in input to the algorithm.*

### C.1  Private weak recovery for stochastic block models

In this section, we show how to achieve weak recovery privately via stability of strongly convex optimization (Lemma B.1). We first introduce one convenient notation. The error rate of an estimate $\hat{x} \in \{\pm 1\}^n$ of the true partition $x \in \{\pm 1\}^n$ is defined as $\operatorname{err}(\hat{x}, x) := \frac{1}{n} \cdot \min\{\operatorname{Ham}(\hat{x}, x), \operatorname{Ham}(\hat{x}, -x)\}$.[24] Our main result is the following theorem.

**Theorem C.3.** *Suppose $\gamma\sqrt{d} \geqslant 12800, \varepsilon, \delta \geqslant 0$. There exists an (Algorithm C.4) such that, for any $x \in \{\pm 1\}^n$, on input $\mathbf{G} \sim \mathsf{SBM}_n(\gamma, d, x)$, outputs $\hat{x}(\mathbf{G}) \in \{\pm 1\}^n$ satisfying*

$$\operatorname{err}(\hat{x}(\mathbf{G}), x) \leqslant O\left(\frac{1}{\gamma\sqrt{d}} + \frac{1}{\gamma d} \cdot \frac{\log(2/\delta)}{\varepsilon^2}\right)$$

*with probability $1 - \exp(-\Omega(n))$. Moreover, the algorithm is $(\varepsilon, \delta)$-differentially private for any input graph and runs in polynomial time.*

Before presenting the algorithm we introduce some notation. Given a graph $G$, let $Y(G) := \frac{1}{\gamma d}(A(G) - \frac{d}{n}J)$ where $A(G)$ is the adjacency matrix of $G$ and $J$ denotes all-one matrices. Define $\mathcal{K} := \{X \in \mathbb{R}^{n \times n} \mid X \succeq 0, X_{ii} = \frac{1}{n} \; \forall i\}$. The algorithm starts with projecting matrix $Y(G)$ to set $\mathcal{K}$. To ensure privacy, then it adds Gaussian noise to the projection $X_1$ and obtains a private matrix $X_2$. The last step applies a standard rounding method.

---

**Algorithm C.4** (Private weak recovery for SBM).
***Input:** Graph G.*

***Operations:***

    *1. Projection: $X_1 \leftarrow \operatorname{argmin}_{X \in \mathcal{K}} \|Y(G) - X\|_F^2$.*

    *2. Noise addition: $\mathbf{X}_2 \leftarrow X_1 + \mathbf{W}$ where $\mathbf{W} \sim \mathcal{N}\left(0, \frac{24}{n\gamma d}\frac{\log(2/\delta)}{\varepsilon^2}\right)^{n \times n}$.*

    *3. Rounding: Compute the leading eigenvector $\mathbf{v}$ of $\mathbf{X}_2$ and return $\operatorname{sign}(\mathbf{v})$.*

---

In the rest of this section, we will show Algorithm C.4 is private in Lemma C.7 and its utility guarantee in Lemma C.8. Then Theorem C.3 follows directly from Lemma C.7 and Lemma C.8.

**Privacy analysis**    Let $\mathcal{Y}$ be the set of all matrices $Y(G) = \frac{1}{\gamma d}(A(G) - \frac{d}{n}J)$ where $G$ is a graph on $n$ vertices. We further define $q : \mathcal{Y} \to \mathcal{K}$ to be the function

$$q(Y) := \operatorname{argmin}_{X \in \mathcal{K}} \|Y - X\|_F^2. \tag{C.1}$$

We first use Lemma B.1 to prove that function $q$ is stable.

**Lemma C.5** (Stability). *The function $q$ as defined in Eq. (C.1) has $\ell_2$-sensitivity $\Delta_{q,2} \leqslant \sqrt{\frac{24}{n\gamma d}}$.*

---
[24]Note $|\langle \hat{x}, x \rangle| = (1 - 2\operatorname{err}(\hat{x}, x)) \cdot n$ for any $\hat{x}, x \in \{\pm 1\}^n$.

*Proof.* Let $g : \mathcal{Y} \times \mathcal{K} \to \mathbb{R}$ be the function $g(Y, X) := \|X\|_{\mathrm{F}}^2 - 2\langle Y, X \rangle$. Applying Lemma B.1 with $f_Y(\cdot) = g(Y, \cdot)$, it suffices to prove that $g$ has $\ell_1$-sensitivity $\frac{4}{n\gamma d}$ with respect to $Y$ and that it is 2-strongly convex with respect to $X$. The $\ell_1$-sensitivity bound follows by observing that adjacent $Y, Y'$ satisfy $\|Y - Y'\|_1 \leqslant \frac{2}{\gamma d}$ and that any $X \in \mathcal{K}$ satisfies $\|X\|_\infty \leqslant \frac{1}{n}$. Thus it remains to prove strong convexity with respect to $X \in \mathcal{K}$. Let $X, X' \in \mathcal{K}$ then

$$
\begin{aligned}
\|X'\|_{\mathrm{F}}^2 &= \|X\|_{\mathrm{F}}^2 + 2\langle X' - X, X\rangle + \|X - X'\|_{\mathrm{F}}^2 \\
&= \|X\|_{\mathrm{F}}^2 + 2\langle X' - X, X + Y - Y\rangle + \|X - X'\|_{\mathrm{F}}^2 \\
&= g(Y, X) + \langle X' - X, \nabla g(X, Y)\rangle + 2\langle X', Y\rangle + \|X - X'\|_{\mathrm{F}}^2.
\end{aligned}
$$

That is $g(Y, X)$ is 2-strongly convex with respect to $X$. Note any $X \in \mathcal{K}$ is symmetric. Then the result follows by Lemma B.1. $\qquad\square$

**Remark C.6.** *In the special case where the contraint set $\mathcal{K}$ does not depend on input dataset $Y$ (e.g. stochastic block models), the proof can be cleaner as follows. Let $f_Y(X) := \|X\|_F^2 - 2\langle X, Y\rangle$. Let $\hat{X} := \arg\min_{X \in \mathcal{K}} f_Y(X)$ and $\hat{X}' := \arg\min_{X \in \mathcal{K}} f_{Y'}(X)$. Suppose without loss of generality $f_Y(\hat{X}) \leqslant f_{Y'}(\hat{X}')$, for a symmetric argument works in the other case. Then*

$$
f_Y(\hat{X}) \leqslant f_{Y'}(\hat{X}') \leqslant f_{Y'}(\hat{X}) \leqslant f_Y(\hat{X}) + \alpha.
$$

Then it is easy to show the algorithm is private.

**Lemma C.7** (Privacy). *The weak recovery algorithm (Algorithm C.4) is $(\varepsilon, \delta)$-DP.*

*Proof.* Since any $X \in \mathcal{K}$ is symmetric, we only need to add a symmetric noise matrix to obtain privacy. Combining Lemma C.5 with Lemma A.12, we immediately get that the algorithm is $(\varepsilon, \delta)$-private. $\qquad\square$

**Utility analysis** Now we show the utility guarantee of our priavte weak recovery algorithm.

**Lemma C.8** (Utility). *For any $x \in \{\pm 1\}^n$, on input $\mathbf{G} \sim \mathsf{SBM}_n(\gamma, d, x)$, Algorithm C.4 efficiently outputs $\hat{x}(\mathbf{G}) \in \{\pm 1\}^n$ satisfying*

$$
\mathrm{err}\,(\hat{x}(\mathbf{G}), x) \leqslant \frac{6400}{\gamma\sqrt{d}} + \frac{7000}{\gamma d} \cdot \frac{\log(2/\delta)}{\varepsilon^2},
$$

*with probability $1 - \exp(-\Omega(n))$.*

To prove Lemma C.8, we need the following lemma which is an adaption of a well-known result in SBM [28, Theorem 1.1]. Its proof is deferred to Appendix H.

**Lemma C.9.** *Consider the settings of Lemma C.8. With probability $1 - \exp(-\Omega(n))$,*

$$
\left\| X_1(\mathbf{G}) - \frac{1}{n}xx^\top \right\|_F^2 \leqslant \frac{800}{\gamma\sqrt{d}}.
$$

*Proof of Lemma C.8.* By Lemma C.9, we have

$$
\left\| X_1(\mathbf{G}) - \frac{1}{n}xx^\top \right\| \leqslant \left\| X_1(\mathbf{G}) - \frac{1}{n}xx^\top \right\|_F \leqslant \sqrt{\frac{800}{\gamma\sqrt{d}}} =: r(\gamma, d)
$$

with probability $1 - \exp(-\Omega(n))$. We condition our following analysis on this event happening.

Let $\mathbf{u}$ be the leading eigenvector of $X_1(\mathbf{G})$. Let $\boldsymbol{\lambda}_1$ and $\boldsymbol{\lambda}_2$ be the largest and second largest eigenvalues of $X_1(\mathbf{G})$. By Weyl's inequality (Lemma F.1) and the assumption $\gamma\sqrt{d} \geqslant 12800$, we have

$$
\boldsymbol{\lambda}_1 - \boldsymbol{\lambda}_2 \geqslant 1 - 2r(\gamma, d) \geqslant \frac{1}{2}.
$$

Let $\mathbf{v}$ be the leading eigenvector of $X_1(\mathbf{G}) + \mathbf{W}$. By Davis-Kahan's theorem (Lemma F.2), we have

$$
\|\mathbf{u} - \mathbf{v}\| \leqslant \frac{2\|\mathbf{W}\|}{\boldsymbol{\lambda}_1 - \boldsymbol{\lambda}_2} \leqslant 4\|\mathbf{W}\|,
$$

$$\left\| \mathbf{u} - x/\sqrt{n} \right\| \leqslant 2 \left\| X_1(\mathbf{G}) - \frac{1}{n}xx^\top \right\| \leqslant 2r(\gamma, d).$$

Putting things together and using Fact E.1, we have

$$\left\| \mathbf{v} - x/\sqrt{n} \right\| \leqslant \left\| \mathbf{u} - \mathbf{v} \right\| + \left\| \mathbf{u} - x/\sqrt{n} \right\| \leqslant \frac{24\sqrt{6}}{\sqrt{\gamma d}} \frac{\sqrt{\log(2/\delta)}}{\varepsilon} + 2r(\gamma, d)$$

with probability $1 - \exp(-\Omega(n))$.

Observe $\mathrm{Ham}(\mathrm{sign}(y), x) \leqslant \|y - x\|^2$ for any $y \in \mathbb{R}^n$ and any $x \in \{\pm 1\}^n$. Then with probability $1 - \exp(-\Omega(n))$,

$$\frac{1}{n} \cdot \mathrm{Ham}(\mathrm{sign}(\mathbf{v}), x) \leqslant \left\| \mathbf{v} - x/\sqrt{n} \right\|^2 \leqslant \frac{6400}{\gamma\sqrt{d}} + \frac{7000}{\gamma d} \cdot \frac{\log(2/\delta)}{\varepsilon^2}.$$

$\square$

*Proof of Theorem C.3.* By Lemma C.7 and Lemma C.8. $\square$

## C.2 Private exact recovery for stochastic block models

In this section, we prove Theorem 1.3. We show how to achieve exact recovery in stochastic block models privately by combining the private weak recovery algorithm we obtained in the previous section and a private majority voting scheme.

Since exact recovery is only possible with logarithmic average degree (just to avoid isolated vertices), it is more convenient to work with the following standard parameterization of stochastic block models. Let $\alpha > \beta > 0$ be fixed constants. The intra-community edge probability is $\alpha \cdot \frac{\log n}{n}$, and the inter-community edge probability is $\beta \cdot \frac{\log n}{n}$. In the language of Model 1.1, it is $\mathsf{SBM}_n(\frac{\alpha+\beta}{2} \cdot \log n, \frac{\alpha-\beta}{\alpha+\beta}, x)$. Our main result is the following theorem.

**Theorem C.10** (Private exact recovery of SBM, restatement of Theorem 1.3). *Let $\varepsilon, \delta \geqslant 0$. Suppose $\alpha, \beta$ are fixed constants satisfying[25]*

$$\sqrt{\alpha} - \sqrt{\beta} \geqslant 16 \quad and \quad \alpha - \beta \geqslant \Omega\left( \frac{1}{\varepsilon^2} \cdot \frac{\log(2/\delta)}{\log n} + \frac{1}{\varepsilon} \right), \tag{C.2}$$

*Then there exists an algorithm (Algorithm C.12) such that, for any balanced[26] $x \in \{\pm 1\}^n$, on input $\mathbf{G} \sim \mathsf{SBM}_n(\frac{\alpha+\beta}{2} \cdot \log n, \frac{\alpha-\beta}{\alpha+\beta}, x)$, outputs $\hat{x}(\mathbf{G}) \in \{x, -x\}$ with probability $1 - o(1)$. Moreover, the algorithm is $(\varepsilon, \delta)$-differentially private for any input graph and runs in polynomial time.*

**Remark C.11.** *In a standard regime of privacy parameters where $\varepsilon \leqslant O(1)$ and $\delta = 1/\mathrm{poly}(n)$, the private exact recovery threshold Eq. (C.2) reads*

$$\sqrt{\alpha} - \sqrt{\beta} \geqslant 16 \quad and \quad \alpha - \beta \geqslant \Omega\left( \varepsilon^{-2} + \varepsilon^{-1} \right),$$

*Recall the non-private exact recovery threshold is $\sqrt{\alpha} - \sqrt{\beta} > \sqrt{2}$. Thus the non-private part in Eq. (C.2), i.e. 16, is close to optimal.*

Algorithm C.12 starts with randomly splitting the input graph $G$ into two subgraphs $\mathbf{G}_1$ and $\mathbf{G}_2$. Setting the graph-splitting probability to $1/2$, each subgraph will contain about half of the edges of $G$. Then we run an $(\varepsilon, \delta)$-DP weak recovery algorithm (Algorithm C.4) on $\mathbf{G}_1$ to get a rough estimate $\tilde{x}(\mathbf{G}_1)$ of accuracy around 90%. Finally, we boost the accuracy to 100% by doing majority voting (Algorithm C.13) on $\mathbf{G}_2$ based on the rough estimate $\tilde{x}(\mathbf{G}_1)$. That is, if a vertex has more neighbors from the opposite community (according to $\tilde{x}(\mathbf{G}_1)$) in $\mathbf{G}_2$, then we assign this vertex to the opposite community. To make the majority voting step private, we add some noise to the vote.

---

[25]In the language of Model 1.1, for any $t$ we have $\sqrt{\alpha} - \sqrt{\beta} \geqslant t$ if and only if $\frac{d}{\log n}(1 - \sqrt{1 - \gamma^2}) \geqslant \frac{t^2}{2}$.

[26]Recall a vector $x \in \{\pm 1\}^n$ is said to be balanced if $\sum_{i=1}^n x_i = 0$.

---

**Algorithm C.12** (Private exact recovery for SBM).
**Input:** *Graph $G$*

**Operations:**

1. *Graph-splitting: Initialize $\mathbf{G}_1$ to be an empty graph on vertex set $V(G)$. Independently put each edge of $G$ in $\mathbf{G}_1$ with probability $1/2$. Let $\mathbf{G}_2 = G \setminus \mathbf{G}_1$.*

2. *Rough estimation on $\mathbf{G}_1$: Run the $(\varepsilon, \delta)$-DP partial recovery algorithm (Algorithm C.4) on $\mathbf{G}_1$ to get a rough estimate $\tilde{x}(\mathbf{G}_1)$.*

3. *Majority voting on $\mathbf{G}_2$: Run the $(\varepsilon, 0)$-DP majority voting algorithm (Algorithm C.13) with input $(\mathbf{G}_2, \tilde{x}(\mathbf{G}_1))$ and get output $\hat{\mathbf{x}}$.*

4. *Return $\hat{\mathbf{x}}$.*

---

**Algorithm C.13** (Private majority voting).
**Input:** *Graph $G$, rough estimate $\tilde{x} \in \{\pm 1\}^n$*

**Operations:**

1. *For each vertex $v \in V(G)$, let $\mathbf{Z}_v = \mathbf{S}_v - \mathbf{D}_v$ where*

   - $\mathbf{D}_v = \sum_{\{u,v\} \in E(G)} \mathbb{1}_{[\tilde{x}_u \neq \tilde{x}_v]}$ ,
   - $\mathbf{S}_v = \sum_{\{u,v\} \in E(G)} \mathbb{1}_{[\tilde{x}_u = \tilde{x}_v]}$ .

   *Set $\hat{\mathbf{x}}_v = \text{sign}(\mathbf{Z}_v + \mathbf{W}_v) \cdot \tilde{x}(\mathbf{G}_1)_v$ where $\mathbf{W}_v \sim \text{Lap}(2/\varepsilon)$.*

2. *Return $\hat{\mathbf{x}}$.*

---

In the rest of this section, we will show Algorithm C.12 is private in Lemma C.15 and it recovers the hidden communities exactly with high probability in Lemma C.17. Then Theorem C.10 follows directly from Lemma C.15 and Lemma C.17.

**Privacy analysis.** We first show the differential privcay of the majority voting algorithm (Algorithm C.13) with respect to input graph $G$ (i.e. assuming fixed the input rough estimate).

**Lemma C.14.** *Algorithm C.13 is $(\varepsilon, 0)$-DP with respect to input $G$.*

*Proof.* Observing the $\ell_1$-sensitivity of the degree count function $Z$ in step is 2, the $(\varepsilon, 0)$-DP follows directly from Laplace mechanism (Lemma A.12) and post-processing (Lemma A.2). $\square$

Then the privacy of the private exact recovery algorithm (Algorithm C.12) is a consequence of composition.

**Lemma C.15** (Privacy). *Algorithm C.12 is $(\varepsilon, \delta)$-DP.*

*Proof.* Let $\mathcal{A}_1 : \mathcal{G}_n \to \{\pm 1\}^n$ denote the $(\varepsilon, \delta)$-DP recovery algorithm in step 2. Let $\mathcal{A}_2 : \mathcal{G}_n \times \{\pm 1\}^n \to \{\pm 1\}^n$ denote the $(\varepsilon, 0)$-DP majority voting algorithm in step 3. Let $\mathcal{A}$ be the composition of $\mathcal{A}_1$ and $\mathcal{A}_2$.

We first make several notations. Given a graph $H$ and an edge $e$, $H_e$ is a graph obtained b adding $e$ to $H$. Given a graph $H$, $\mathbf{G}_1(H)$ is a random subgraph of $H$ by keeping each edge of $H$ with probability $1/2$ independently.

Now, fix two adjacent graphs $G$ and $G_e$ where edge $e$ appears in $G_e$ but not in $G$. Also, fix two arbitrary possible outputs $x_1, x_2 \in \{\pm 1\}^n$ of algorithm $\mathcal{A}$.[27] It is direct to see,

$$\mathbb{P}\left(\mathcal{A}(G) = (x_1, x_2)\right) = \sum_{H \subseteq G} \mathbb{P}\left(\mathcal{A}_1(H) = x_1\right) \mathbb{P}\left(\mathcal{A}_2(G \setminus H, x_1) = x_2\right) \mathbb{P}\left(\mathbf{G}_1(G) = H\right). \quad \text{(C.3)}$$

---

[27]We can imagine that algorithm $\mathcal{A}$ first outputs $(x_1, x_2)$ and then outputs $x_2$ as a post-processing step.

Since $\mathbb{P}(\mathbf{G}_1(G) = H) = \mathbb{P}(\mathbf{G}_1(G_e) = H) + \mathbb{P}(\mathbf{G}_1(G_e) = H_e)$ for any $H \subseteq G$, we have

$$\mathbb{P}\left(\mathcal{A}(G_e) = (x_1, x_2)\right) = \sum_{H \subseteq G} \mathbb{P}\left(\mathcal{A}_1(H) = x_1\right) \mathbb{P}\left(\mathcal{A}_2(G_e \setminus H, x_1) = x_2\right) \mathbb{P}\left(\mathbf{G}_1(G_e) = H\right)$$

$$+ \mathbb{P}\left(\mathcal{A}_1(H_e) = x_1\right) \mathbb{P}\left(\mathcal{A}_2(G_e \setminus H_e, x_1) = x_2\right) \mathbb{P}\left(\mathbf{G}_1(G_e) = H_e\right)$$
(C.4)

Since both $\mathcal{A}_1$ and $\mathcal{A}_2$ are $(\varepsilon, \delta)$-DP, we have for each $H \subseteq G$,

$$\mathbb{P}\left(\mathcal{A}_1(H_e) = x_1\right) \leqslant e^{\varepsilon} \mathbb{P}\left(\mathcal{A}_1(H) = x_1\right) + \delta,$$
(C.5)

$$\mathbb{P}\left(\mathcal{A}_2(G_e \setminus H, x_1) = x_2\right) \leqslant e^{\varepsilon} \mathbb{P}\left(\mathcal{A}_2(G \setminus H, x_1) = x_2\right) + \delta.$$
(C.6)

Plugging Eq. (C.5) and Eq. (C.6) into Eq. (C.4), we obtain

$$\mathbb{P}\left(\mathcal{A}(G_e) = (x_1, x_2)\right) \leqslant \sum_{H \subseteq G} \left[e^{\varepsilon} \mathbb{P}\left(\mathcal{A}_1(H) = x_1\right) \mathbb{P}\left(\mathcal{A}_2(G \setminus H, x_1) = x_2\right) + \delta\right] \mathbb{P}\left(\mathbf{G}_1(G) = H\right)$$

$$= e^{\varepsilon} \mathbb{P}\left(\mathcal{A}(G) = (x_1, x_2)\right) + \delta.$$

Similarly, we can show

$$\mathbb{P}\left(\mathcal{A}(G) = (x_1, x_2)\right) \leqslant e^{\varepsilon} \mathbb{P}\left(\mathcal{A}(G_e) = (x_1, x_2)\right) + \delta.$$
(C.7)

$\square$

**Utility analysis.** We first show the utility guarantee of the priavte majority voting algorithm.

**Lemma C.16.** *Suppose $\mathbf{G}$ is generated by first sampling $\mathbf{G} \sim \mathsf{SBM}_n(\frac{\alpha+\beta}{2} \cdot \log n, \frac{\alpha-\beta}{\alpha+\beta}, x)$ for some balanced $x$ and then for each vertex removing at most $\Delta \leqslant O(\log^2 n)$ adjacent edges arbitrarily. Then on input $\mathbf{G}$ and a balanced rough estimate $\tilde{x}$ satisfying $\mathrm{Ham}(\tilde{x}, x) \leqslant n/16$, Algorithm C.13 efficiently outputs $\hat{x}(\mathbf{G})$ such that for each vertex $v$,*

$$\mathbb{P}\left(\hat{x}(\mathbf{G})_v \neq x_v\right) \leqslant \exp\left(-\frac{1}{64} \cdot \varepsilon(\alpha - \beta) \cdot \log n\right) + 2 \cdot \exp\left(-\frac{1}{16^2} \cdot \frac{(\alpha - \beta)^2}{\alpha + \beta} \cdot \log n\right).$$

*Proof.* Let us fix an arbitrary vertex $v$ and analyze the probability $\mathbb{P}\left(\hat{x}(\mathbf{G})_v \neq x_v\right)$. Let $r := \mathrm{Ham}(\tilde{x}, x)/n$. Then it is not hard to see

$$\mathbb{P}\left(\hat{x}(\mathbf{G})_v \neq x_v\right) \leqslant \mathbb{P}\left(\mathbf{B} + \mathbf{A}' - \mathbf{A} - \mathbf{B}' + \mathbf{W} > 0\right)$$
(C.8)

where

- $\mathbf{A} \sim \mathrm{Binomial}((1/2 - r)n - \Delta, \alpha \frac{\log n}{n})$, corresponding to the number of neighbors that are from the same community and correctly labeled by $\tilde{x}$,

- $\mathbf{B}' \sim \mathrm{Binomial}(rn - \Delta, \beta \frac{\log n}{n})$, corresponding to the number of neighbors that are from the different community but incorrectly labeled by $\tilde{x}$,

- $\mathbf{B} \sim \mathrm{Binomial}((1/2 - r)n, \beta \frac{\log n}{n})$, corresponding to the number of neighbors that are from the different community and correctly labeled by $\tilde{x}$,

- $\mathbf{A}' \sim \mathrm{Binomial}(rn, \alpha \frac{\log n}{n})$, corresponding to the number of neighbors that are from the same community but incorrectly labeled by $\tilde{x}$,

- $\mathbf{W} \sim \mathrm{Lap}(0, 2/\varepsilon)$, independently.

The $\Delta$ term appearing in both $\mathbf{A}$ and $\mathbf{B}'$ corresponds to the worst case where $\Delta$ "favorable" edges are removed. If $r \geqslant \Omega(1)$, then $\Delta = O(\log^2 n)$ is negligible to $rn = \Theta(n)$ and we can safely ignore the effect of removing $\Delta$ edges. If $r = o(1)$, then we can safely assume $\tilde{x}$ is correct on all vertices and ignore the effect of removing $\Delta$ edges as well. Thus, we will assume $\Delta = 0$ in the following analysis.

For any $t, t'$, we have

$$\mathbb{P}\left(\mathbf{A}' + \mathbf{B} - \mathbf{A} - \mathbf{B}' + \mathbf{W} > 0\right) \leqslant \mathbb{P}\left(\mathbf{A}' + \mathbf{B} + \mathbf{W} > t\right) + \mathbb{P}\left(\mathbf{A} + \mathbf{B}' \leqslant t\right)$$

$$\leqslant \mathbb{P}\left(\mathbf{A}' + \mathbf{B} \geqslant t - t'\right) + \mathbb{P}\left(\mathbf{W} \geqslant t'\right) + \mathbb{P}\left(\mathbf{A} + \mathbf{B}' \leqslant t\right).$$

We choose $t, t'$ by first picking two constants $a, b > 0$ satisfying $a + b < 1$ and then solving

- $\mathbb{E}[\mathbf{A}' + \mathbf{B}] - t = a \cdot (\mathbb{E}[\mathbf{A} + \mathbf{B}'] - \mathbb{E}[\mathbf{A}' + \mathbf{B}])$ and
- $t' = (1 - a - b) \cdot (\mathbb{E}[\mathbf{A} + \mathbf{B}'] - \mathbb{E}[\mathbf{A}' + \mathbf{B}])$.

By Fact A.7,

$$\mathbb{P}\left(\mathbf{W} > t'\right) \leqslant \exp\left(-\frac{t'\varepsilon}{2}\right) \leqslant \exp\left(-\frac{(1/4 - r)(1 - a - b)}{2} \cdot \varepsilon(\alpha - \beta) \cdot \log n\right).$$

By Fact E.4 and the assumption $r \leqslant 1/16$, we have

$$\mathbb{P}\left(\mathbf{A} + \mathbf{B}' \leqslant t\right) \leqslant \exp\left(-\frac{(\mathbb{E}[\mathbf{A} + \mathbf{B}'] - t)^2}{2\,\mathbb{E}[\mathbf{A} + \mathbf{B}']}\right) \leqslant \exp\left(-(1/4 - r)^2 a^2 \cdot \frac{(\alpha - \beta)^2}{\alpha + \beta} \cdot \log n\right).$$

Setting $b = 1/2$, by Fact E.4 and the assumption $r \leqslant 1/16$, we have

$$\mathbb{P}\left(\mathbf{A}' + \mathbf{B} \geqslant t - t'\right) \leqslant \exp\left(-\frac{(t - t' - \mathbb{E}[\mathbf{A}' + \mathbf{B}])^2}{t - t' + \mathbb{E}[\mathbf{A}' + \mathbf{B}]}\right) \leqslant \exp\left(-\frac{2(1/4 - r)^2}{7} \cdot \frac{(\alpha - \beta)^2}{\alpha + \beta} \cdot \log n\right).$$

Further setting $a = 1/3$, we have

$$\mathbb{P}\left((\hat{x}(\mathbf{G})_v \neq x_v\right) \leqslant \exp\left(-\frac{1/4 - r}{12} \cdot \varepsilon(\alpha - \beta) \cdot \log n\right) + 2 \cdot \exp\left(-\frac{(1/4 - r)^2}{9} \cdot \frac{(\alpha - \beta)^2}{\alpha + \beta} \cdot \log n\right).$$

Finally, plugging the assumption $r \leqslant 1/16$ to conclude. $\qquad\square$

Then it is not difficult to show the utility guarantee of our priavte exact recovery algorithm.

**Lemma C.17** (Utility). *Suppose* $\alpha, \beta$ *are fixed constants satisfying*

$$\sqrt{\alpha} - \sqrt{\beta} \geqslant 16 \quad and \quad \alpha - \beta \geqslant \Omega\left(\frac{1}{\varepsilon^2} \cdot \frac{\log(2/\delta)}{\log n} + \frac{1}{\varepsilon}\right).$$

*Then for any balanced* $x \in \{\pm 1\}^n$, *on input* $\mathbf{G} \sim \mathsf{SBM}_n(\frac{\alpha+\beta}{2} \cdot \log n, \frac{\alpha-\beta}{\alpha+\beta}, x)$, *Algorithm C.12 efficiently outputs* $\hat{x}(\mathbf{G})$ *satisfying* $\hat{x}(\mathbf{G}) \in \{x, -x\}$ *with probability* $1 - o(1)$.

*Proof.* We will show the probability of a fixed vertex being misclassified is at most $o(1/n)$. Then by union bound, exact recovery can be achieved with probability $1 - o(1)$.

As the graph-splitting probability is $1/2$, $\mathbf{G}_1$ follows $\mathsf{SBM}_n(\frac{\alpha}{2} \cdot \frac{\log n}{n}, \frac{\beta}{2} \cdot \frac{\log n}{n}, x)$. By Theorem C.3, the rough estimate $\tilde{x}(\mathbf{G}_1)$ satisfies[28]

$$\mathrm{err}(\tilde{x}(\mathbf{G}_1), x) \leqslant r := o(1) + \frac{14000}{(\alpha - \beta)\varepsilon^2} \cdot \frac{\log(2/\delta)}{\log n}. \tag{C.9}$$

with probability at least $1 - \exp(-\Omega(n))$. Without loss of generality, we can assume $\mathrm{Ham}(\tilde{x}(\mathbf{G}_1), x) \leqslant rn$, since we consider $-x$ otherwise. By Fact E.2, the maximum degree of $\mathbf{G}_1$ is at most $\Delta := 2\log^2 n$ with probability at least $1 - n\exp(-(\log n)^2/3)$. In the following, we condition our analysis on the above two events regarding $\tilde{x}(\mathbf{G}_1)$ and $\mathbf{G}_1$.

Now, let us fix a vertex and analyze the probability $p_e$ that it is misclassified after majority voting. With $G_1$ being fixed, $\mathbf{G_2}$ can be thought of as being generated by first sampling $\mathbf{G}$ and then removing $G_1$ from $\mathbf{G}$. To make $r \leqslant 1/16$, it suffices to ensure $\alpha - \beta > \frac{500^2}{\varepsilon^2} \cdot \frac{\log(2/\delta)}{\log n}$ by Eq. (C.9). Then by Lemma C.16, we have

$$p_e \leqslant \exp\left(-\frac{1}{64} \cdot \varepsilon(\alpha - \beta) \cdot \log n\right) + 2 \cdot \exp\left(-\frac{1}{16^2} \cdot \frac{(\alpha - \beta)^2}{\alpha + \beta} \cdot \log n\right).$$

To make $p_e$ at most $o(1/n)$, it suffices to ensure

$$\frac{1}{64} \cdot \varepsilon(\alpha - \beta) > 1 \quad and \quad \frac{1}{16^2} \cdot \frac{(\alpha - \beta)^2}{\alpha + \beta} > 1.$$

---

[28]It is easy to make the output of Algorithm C.4 balanced at the cost of increasing the error rate by a factor of at most 2.

Note $(\alpha - \beta)^2/(\alpha + \beta) > (\sqrt{\alpha} - \sqrt{\beta})^2$ for $\alpha > \beta$. Therefore, as long as

$$\sqrt{\alpha} - \sqrt{\beta} \geqslant 16 \quad \text{and} \quad \alpha - \beta \geqslant \frac{500^2}{\varepsilon^2} \cdot \frac{\log(2/\delta)}{\log n} + \frac{64}{\varepsilon},$$

Algorithm C.12 recovers the hidden communities exactly with probability $1 - o(1)$. □

*Proof of Theorem C.10.* By Lemma C.15 and Lemma C.17. □

## C.3  Inefficient recovery using the exponential mechanism

In this section, we will present an inefficient algorithm satisfying pure privacy which succeeds for all ranges of parameters - ranging from weak to exact recovery. The algorithm is based on the exponential mechanism [43] combined with the majority voting scheme introduced in section Appendix C.2. In particular, we will show

**Theorem C.18** (Full version of Theorem 1.4). *Let $\gamma\sqrt{d} \geqslant 12800$ and $x \in \{\pm 1\}^n$ be balanced. Let $\zeta \geqslant 2\exp\left(-\frac{\gamma^2 d}{512}\right)$. For any $\varepsilon \geqslant \frac{64\log(2/\zeta)}{\gamma d}$, there exists an algorithm, Algorithm C.19, which on input $\mathbf{G} \sim \mathsf{SBM}_n(\gamma, d, x^*)$ outputs an estimate $\hat{x}(\mathbf{G}) \in \{\pm 1\}^n$ satisfying*

$$\mathrm{err}\left(\hat{x}(\mathbf{G}), x^*\right) \leqslant \zeta$$

*with probability at least $1 - \zeta$. In addition, the algorithm is $\varepsilon$-private. Further, by slightly modifying the algorithm, we can achieve error $20/\sqrt{\log(1/\zeta)}$ with probability $1 - e^{-n}$.[29]*

A couple of remarks are in order. First, our algorithm works across all degree-regimes in the literature and matches known non-private thresholds and rates up to constants. We remark that for ease of exposition we did not try to optimize these constants. In particular, for $\gamma^2 d$ a constant we achieve weak recovery. We reiterate, that $\gamma^2 d > 1$ is the optimal non-private threshold. For the regime, where $\gamma^2 d = \omega(1)$, it is known that the optimal error rate is $\exp\left(-(1 - o(1))\gamma^2 d\right)$ even non-privately [64], where $o(1)$ goes to zero as $\gamma^2 d$ tends to infinity. We match this up to constants. Moreover, our algorithm achieves exact recovery as soon as $\gamma^2 d \geqslant 512\log n$ since then $\zeta < \frac{1}{n}$. This also matches known non-private threshholds up to constants [5, 47]. Also, our dependence on the privacy parameter $\varepsilon$ is also optimal as shown by the information-theoretic lower bounds in Appendix C.4.

We also emphasize, that if we only aim to achieve error on the order of

$$\frac{1}{\gamma\sqrt{d}} = \Theta\left(\frac{1}{\sqrt{\log(1/\zeta)}}\right),$$

we can achieve exponentially small failure probability in $n$, while keeping the privacy parameter $\varepsilon$ the same. This can be achieved, by ommitting the boosting step in our algorithm and will be clear from the proof of Theorem C.18. We remark that in this case, we can also handle non-balanced communities.

Again, for an input graph $G$, consider the matrix $Y(G) = \frac{1}{\gamma d}\left(A(G) - \frac{d}{n}J\right)$. For $x \in \{\pm 1\}^n$ we define the score function

$$s_G(x) = \langle x, Y(G)x \rangle.$$

Since the entries of $A(G)$ are in $[0, 1]$ and adjacent graphs differ in at most one edge, it follows immediately, that this score function has sensitivity at most

$$\Delta = \max_{\substack{G \sim G', \\ x \in \{\pm 1\}^n}} |s_G(x) - s_{G'}(x)| = \frac{2}{\gamma d} \cdot \max_{\substack{G \sim G', \\ x \in \{\pm 1\}^n}} |\langle x, (A(G) - A(G'))x \rangle| \leqslant \frac{2}{\gamma d}.$$

---

[29]The first, smaller, error guarantee additionally needs the requirement that $\zeta \leqslant \exp(-640)$. The second one does not.

> **Algorithm C.19** (Inefficient algorithm for SBM).
> **Input:** *Graph $G$, privacy parameter $\varepsilon > 0$*
>
> **Operations:**
>
> 1. *Graph-splitting: Initialize $\mathbf{G}_1$ to be an empty graph on vertex set $V(G)$. Independently assign each edge of $G$ to $\mathbf{G}_1$ with probability $1/2$. Let $\mathbf{G}_2 = G \setminus \mathbf{G}_1$.*
>
> 2. *Rough estimation on $\mathbf{G}_1$: Sample $\tilde{x}$ from the distribution with density*
>    $$p(x) \propto \exp\left(\frac{\varepsilon}{2\Delta}\langle x, Y(\mathbf{G}_1)x\rangle\right),$$
>    *where $\Delta = \frac{2}{\gamma d}$.*
>
> 3. *Majority voting on $\mathbf{G}_2$: Run the $\varepsilon$-DP majority voting algorithm (Algorithm C.13) with input $(\mathbf{G}_2, \tilde{x}(\mathbf{G}_1))$. Denote its output by $\hat{\mathbf{x}}$.*
>
> 4. *Return $\hat{\mathbf{x}}$.*

We first analyze the privacy guarantees of the above algorithm.

**Lemma C.20.** *Algorithm C.19 is $\varepsilon$-DP.*

*Proof.* For simplicity and clarity of notation, we will show that the algorithm satisfies $2\varepsilon$-DP. Clearly, the graph splitting step is 0-DP. Step 2 corresponds to the exponential mechanism. Since the sensitivity of the score function is at most $\Delta = \frac{2}{\gamma d}$ it follows by the standard analysis of the mechanism that this step is $\varepsilon$-DP [43]. By Lemma C.14, the majority voting step is also $\varepsilon$-DP. Hence, the result follows by composition (cf. Lemma A.4). □

Next, we will analyze its utility.

**Lemma C.21.** *Let $\gamma\sqrt{d} \geqslant 12800$ and $x \in \{\pm 1\}^n$ be balanced. Let $\exp(-640) \geqslant \zeta \geqslant 2\exp\left(-\frac{\gamma^2 d}{512}\right), \varepsilon \geqslant \frac{64\log(2/\zeta)}{\gamma d}$, and $\mathbf{G} \sim \mathsf{SBM}_n(\gamma, d, x^*)$, the output $\hat{x}(\mathbf{G}) \in \{\pm 1\}^n$ of Algorithm C.19 satisfies*
$$\mathrm{err}\,(\hat{x}(\mathbf{G}), x^*) \leqslant \zeta$$
*with probability at least $1 - \zeta$.*

*Proof.* We will first show that the rough estimate $\tilde{x}$ obtained in step 2 achieves
$$\mathrm{err}\,(\tilde{x}, x^*) \leqslant \frac{20}{\sqrt{\log(1/\zeta)}}$$
with probability $e^{-n}$. This will prove the second part of the theorem - for this we don't need that $\zeta \leqslant \exp(-640)$. In fact, arbitrary $\zeta$ works. The final error guarantee will then follow by Lemma C.16. First, notice that similar to the proof of [28, Lemma 4.1], using Bernstein's inequality and a union bound, we can show that (cf. Fact H.2 for a full proof)
$$\max_{x \in \{\pm 1\}^n} \left|\left\langle x, \left[Y(\mathbf{G}) - \frac{1}{n}x^*(x^*)^\top\right]x\right\rangle\right| \leqslant \frac{100n}{\gamma\sqrt{d}} \leqslant \frac{5}{\sqrt{\log(1/\zeta)}}$$
with probability at least $1 - \exp^{-10n}$. Recall that $s_{\mathbf{G}}(x) = \langle x, Y(\mathbf{G})x\rangle$. Let $\alpha = \frac{5}{\sqrt{\log(1/\zeta)}}$. We call $x \in \{\pm 1\}^n$ *good* if $s_{\mathbf{G}}(x) \geqslant (1 - 3\alpha)n$. It follows that for good $x$ it holds that
$$\frac{1}{n} \cdot \langle x, x^*\rangle^2 \geqslant \langle x, Y(\mathbf{G})x\rangle - \left|\left\langle x, \left[Y(\mathbf{G}) - \frac{1}{n}x^*(x^*)^\top\right]x\right\rangle\right| \geqslant (1 - 4\alpha)n\,.$$
Which implies that
$$2\,\mathrm{err}(x, x^*) \leqslant 1 - \sqrt{1 - 4\alpha} = 1 - \frac{1 - 4\alpha}{\sqrt{1 - 4\alpha}} \leqslant 1 - \frac{1 - 4\alpha}{1 - 2\alpha} = \frac{2\alpha}{1 - 2\alpha} \leqslant 4\alpha\,,$$

where we used that $\alpha \leqslant 1/4$ and that $\sqrt{1-4x} \leqslant 1-2x$ for $x \geqslant 0$. Hence, we have for good $x$ that

$$\mathrm{err}(x, x^*) \leqslant \frac{20}{\sqrt{\log{(1/\zeta)}}} \,.$$

Since $s_{\mathbf{G}}(x^*) \geqslant (1-\alpha)n$, there is at least one good candidate. Hence, we can bound the probability that we do not output a good $x$ as

$$\frac{\exp\left(\frac{\varepsilon}{2\Delta}(1-3\alpha)n\right) \cdot e^n}{\exp\left(\frac{\varepsilon}{2\Delta}(1-\alpha)n\right) \cdot 1} = \exp\left(\left(1 - \frac{2\varepsilon\alpha}{\Delta}\right)n\right) \leqslant e^{-n} \,,$$

where we used that

$$\frac{2\varepsilon\alpha}{\Delta} \geqslant \frac{64\log{(2/\zeta)}}{\gamma d} \cdot \frac{5\gamma d}{\sqrt{\log{(1/\zeta)}}} \geqslant 320\sqrt{\log{(1/\zeta)}} \geqslant 2 \,.$$

We will use Lemma C.16 to proof the final conclusion of the theorem. In what follows, assume without loss of generality that $\mathrm{Ham}\,(x, x^*) < \mathrm{Ham}\,(x, -x^*)$. The above discussion implies that

$$\mathrm{Ham}\,(x, x^*) \leqslant 8\alpha n \leqslant \frac{40n}{\sqrt{\log{(1/\zeta)}}} \leqslant \frac{n}{16} \,,$$

where the last inequality uses $\zeta \leqslant e^{-640}$. Further, by Fact E.2 it also follows that the maximum degree of $\mathbf{G}_2$ is at most $O\left(\log^2 n\right)$ (by some margin). Recall that $\mathbf{G}_2 \sim \mathsf{SBM}\,(d, \gamma, x^*)$. In the parametrization of Lemma C.16 this means that

$$\alpha = \frac{(1+\gamma)\,d}{\log n} \,, \qquad \beta = \frac{(1-\gamma)\,d}{\log n} \,,$$

$$\alpha - \beta = \frac{2\gamma d}{\log n} \,, \qquad \alpha + \beta = \frac{2d}{\log n} \,.$$

Thus, it follows that the output $\hat{x}$ of the majority voting step satisfies for every vertex $v$

$$\mathbb{P}\left(\hat{x}(\mathbf{G})_v \neq x_v\right) \leqslant \exp\left(-\frac{1}{64} \cdot \varepsilon(\alpha - \beta) \cdot \log n\right) + 2 \cdot \exp\left(-\frac{1}{16^2} \cdot \frac{(\alpha-\beta)^2}{\alpha+\beta} \cdot \log n\right)$$

$$\leqslant \exp\left(-\frac{1}{32} \cdot \varepsilon\gamma d\right) + \exp\left(-\frac{1}{16^2} \cdot \gamma^2 d\right)$$

$$\leqslant \zeta^2/4 + \zeta^2/4 \leqslant \zeta^2 \,.$$

By Markov's Inequality it now follows that

$$\mathbb{P}\left(\mathrm{err}\,(\hat{x}(\mathbf{G}), x^*) \geqslant \zeta\right) \leqslant \zeta \,.$$

$\square$

## C.4  Lower bound on the parameters for private recovery

In this section, we prove a tight lower bound for private recovery for stochastic block models. Recall the definition of error rate, $\mathrm{err}(u, v) := \frac{1}{n} \cdot \min\{\mathrm{Ham}(u, v), \mathrm{Ham}(u, -v)\}$ for $u, v \in \{\pm 1\}^n$. Our main result is the following theorem.

**Theorem C.22** (Full version of Theorem 1.5). *Suppose there exists an $\varepsilon$-differentially private algorithm such that for any balanced $x \in \{\pm 1\}^n$, on input $\mathbf{G} \sim \mathsf{SBM}_n(d, \gamma, x)$, outputs $\hat{x}(\mathbf{G}) \in \{\pm 1\}^n$ satisfying*

$$\mathbb{P}\left(\mathrm{err}(\hat{x}(\mathbf{G}), x) < \zeta\right) \geqslant 1 - \eta,$$

*where[30] $1/n \leqslant \zeta \leqslant 0.04$ and the randomness is over both the algorithm and stochastic block models. Then,*

$$e^{2\varepsilon} - 1 \geqslant \Omega\left(\frac{\log(1/\zeta)}{\gamma d} + \frac{\log(1/\eta)}{\zeta n \gamma d}\right). \tag{C.10}$$

---

[30]Error rate less than $1/n$ already means exact recovery. Thus it does not make sense to set $\zeta$ to any value strictly smaller than $1/n$. The upper bound $\zeta \leqslant 0.04$ is just a technical condition our proof needs for Eq. (C.12).

**Remark C.23.** *Both terms in lower bound Eq.* (C.10) *are tight up to constants by the following argument. Considering typical privacy parameters $\varepsilon \leqslant 1$, then $e^{2\varepsilon} - 1 \approx 2\varepsilon$. For exponentially small failure probability, i.e. $\eta = 2^{-\Omega(n)}$, the lower bound reads $\varepsilon \geqslant \Omega(\frac{1}{\gamma d} \cdot \frac{1}{\zeta})$, which is achieved by Algorithm C.19 without the boosting step - see the discussion after Theorem C.18. For polynomially small failure probability, i.e. $\eta = 1/\operatorname{poly}(n)$, the lower bound Eq.* (C.10) *reads $\varepsilon \geqslant \Omega(\frac{1}{\gamma d} \cdot \log \frac{1}{\zeta})$, which is achieved by Theorem C.18.*

By setting $\zeta = 1/n$ in Theorem C.22, we directly obtain a tight lower bound for private exact recovery as a corollary.

**Corollary C.24.** *Suppose there exists an $\varepsilon$-differentially private algorithm such that for any balanced $x \in \{\pm 1\}^n$, on input $\mathbf{G} \sim \mathsf{SBM}_n(d, \gamma, x)$, outputs $\hat{x}(\mathbf{G}) \in \{\pm 1\}^n$ satisfying*

$$\mathbb{P}\left(\hat{x}(\mathbf{G}) \in \{x, -x\}\right) \geqslant 1 - \eta,$$

*where the randomness is over both the algorithm and stochastic block models. Then,*

$$e^{2\varepsilon} - 1 \geqslant \Omega\left(\frac{\log(n) + \log \frac{1}{\eta}}{\gamma d}\right). \tag{C.11}$$

**Remark C.25.** *The lower bound Eq.* (C.11) *for priavte exact recovery is tight up to constants, since there exists an (inefficient) $\varepsilon$-differentially priavte exact recovery algorithm with $\varepsilon \leqslant O(\frac{\log n}{\gamma d})$ and $\eta = 1/\operatorname{poly}(n)$ by Theorem C.18 and [55, Theorem 3.7].*

In rest of this section, we will prove Theorem C.22. The proof applies the packing lower bound argument similar to [29, Theorem 7.1]. To this end, we first show $\operatorname{err}(\cdot, \cdot)$ is a semimetric over $\{\pm 1\}^n$.

**Lemma C.26.** $\operatorname{err}(\cdot, \cdot)$ *is a semimetric over $\{\pm 1\}^n$.*

*Proof.* Symmetry and non-negativity are obvious from the definition. We will show $\operatorname{err}(\cdot, \cdot)$ satisfies triangle inequality via case analysis. Let $u, v, w \in \{\pm 1\}^n$ be three arbitrary sign vectors. By symmetry, we only need to consider the following four cases.

*Case 1:* $\operatorname{Ham}(u, v), \operatorname{Ham}(u, w), \operatorname{Ham}(v, w) \leqslant n/2$. This case is reduced to showing Hamming distance satisfies triangle inequality, which is obvious.

*Case 2:* $\operatorname{Ham}(u, v), \operatorname{Ham}(u, w) \leqslant n/2$ and $\operatorname{Ham}(v, w) \geqslant n/2$. We need to check two subcases. First,

$$\operatorname{err}(u, v) \leqslant \operatorname{err}(u, w) + \operatorname{err}(v, w) \Leftrightarrow \operatorname{Ham}(u, v) + \operatorname{Ham}(v, w) \leqslant \operatorname{Ham}(u, w) + n$$
$$\Leftarrow \operatorname{Ham}(u, v) + H(u, v) + H(u, w) \leqslant \operatorname{Ham}(u, w) + n$$
$$\Leftrightarrow \operatorname{Ham}(u, v) \leqslant n/2.$$

Second,

$$\operatorname{err}(v, w) \leqslant \operatorname{err}(u, v) + \operatorname{err}(u, w) \Leftrightarrow n \leqslant \operatorname{Ham}(v, w) + \operatorname{Ham}(u, v) + \operatorname{Ham}(u, w)$$
$$\Leftarrow n \leqslant 2 \operatorname{Ham}(v, w).$$

*Case 3:* $\operatorname{Ham}(u, v) \leqslant n/2$ and $\operatorname{Ham}(u, w), \operatorname{Ham}(v, w) \geqslant n/2$. This case can be reduced to case 1 by considering $u, v, -w$.

*Case 4:* $\operatorname{Ham}(u, v), \operatorname{Ham}(u, w), \operatorname{Ham}(v, w) \geqslant n/2$. This case can be reduced to case 2 by considering $-u, v, w$. $\qquad \square$

*Proof of Theorem C.22.* Suppose there exists an $\varepsilon$-differentially private algorithm satisfying the theorem's assumption.

We first make the following notation. Given a semimetric $\rho$ over $\{\pm 1\}^n$, a center $v \in \{\pm 1\}^n$, and a radius $r \geqslant 0$, define $B_\rho(v, r) := \{w \in \{\pm 1\}^n : \mathbf{1}^\top w = 0, \rho(w, v) \leqslant r\}$.

Pick an arbitrary balanced $x \in \{\pm 1\}^n$. Let $M = \{x^1, x^2, \ldots, x^m\}$ be a maximal $2\zeta$-packing of $B_{\operatorname{err}}(x, 4\zeta)$ in semimetric $\operatorname{err}(\cdot, \cdot)$. By maximality of $M$, we have $B_{\operatorname{err}}(x, 4\zeta) \subseteq \cup_{i=1}^m B_{\operatorname{err}}(x^i, 2\zeta)$, which implies

$$|B_{\operatorname{err}}(x, 4\zeta)| \leqslant \sum_{i=1}^m \left|B_{\operatorname{err}}(x^i, 2\zeta)\right|$$

$$\implies |B_{\text{Ham}}(x, 4\zeta)| \leqslant \sum_{i=1}^{m} 2 \cdot |B_{\text{Ham}}(x^i, 2\zeta)| = 2m \cdot |B_{\text{Ham}}(x, 2\zeta)|$$

$$\implies 2m \geqslant \frac{|B_{\text{Ham}}(x, 4\zeta n)|}{|B_{\text{Ham}}(x, 2\zeta n)|} = \frac{\binom{n/2}{2\zeta n}^2}{\binom{n/2}{\zeta n}^2} \geqslant \frac{\left(\frac{1}{4\zeta}\right)^{4\zeta n}}{\left(\frac{e}{2\zeta}\right)^{2\zeta n}} = \left(\frac{1}{8e\zeta}\right)^{2\zeta n} \tag{C.12}$$

For each $i \in [m]$, define $Y_i := \{w \in \{\pm 1\}^n : \text{err}(w, x^i) \leqslant \zeta\}$. Then $Y_i$'s are pairwise disjoint. For each $i \in [m]$, let $P_i$ be the distribution over $n$-vertex graphs generated by $\text{SBM}_n(d, \gamma, x^i)$. By our assumption on the algorithm, we have for any $i \in [m]$ that

$$\mathbb{P}_{\mathbf{G} \sim P_i} (\hat{x}(\mathbf{G}) \in Y_i) \geqslant 1 - \eta.$$

Combining the fact that $Y_i$'s are pairwise disjoint, we have

$$\sum_{i=1}^{m} \mathbb{P}_{\mathbf{G} \sim P_1} (\hat{x}(\mathbf{G}) \in Y_i) = \mathbb{P}_{\mathbf{G} \sim P_1} (\hat{x}(\mathbf{G}) \in \cup_{i=1}^{m} Y_i) \leqslant 1 \implies \sum_{i=2}^{m} \mathbb{P}_{\mathbf{G} \sim P_1} (\hat{x}(\mathbf{G}) \in Y_i) \leqslant \eta. \tag{C.13}$$

In the following, we will lower bound $\mathbb{P}_{\mathbf{G} \sim P_1}(\hat{x}(\mathbf{G}) \in Y_i)$ for each $i \in [m] \setminus \{1\}$ using group privacy.

Note each $P_i$ is a product of $\binom{n}{2}$ independent Bernoulli distributions. Thus for any $i, j \in [m]$, there exists a coupling $\omega_{ij}$ of $P_i$ and $P_j$ such that, if $(\mathbf{G}, \mathbf{H}) \sim \omega$, then

$$\text{Ham}(\mathbf{G}, \mathbf{H}) \sim \text{Binomial}(N_{ij}, p),$$

where $p = 2\gamma d/n$ and $N_{ij} = \text{Ham}(x^i, x^j) \cdot (n - \text{Ham}(x^i, x^j))$. Applying group privacy, we have for any two graphs $G, H$ and for any $S \subseteq \{\pm 1\}^n$ that[31]

$$\mathbb{P}(\hat{x}(G) \in S) \leqslant \exp(\varepsilon \cdot \text{Ham}(G, H)) \cdot \mathbb{P}(\hat{x}(H) \in S). \tag{C.14}$$

For each $i \in [m]$, taking expectations on both sides of Eq. (C.14) with respect to coupling $\omega_{i1}$ and setting $S = Y_i$, we have

$$\mathbb{E}_{(\mathbf{G}, \mathbf{H}) \sim \omega_{i1}} \mathbb{P}(\hat{x}(\mathbf{G}) \in Y_i) \leqslant \mathbb{E}_{(\mathbf{G}, \mathbf{H}) \sim \omega_{i1}} \exp(\varepsilon \cdot \text{Ham}(\mathbf{G}, \mathbf{H})) \cdot \mathbb{P}(\hat{x}(\mathbf{H}) \in Y_i). \tag{C.15}$$

The left side of Eq. (C.15) is equal to

$$\mathbb{E}_{(\mathbf{G}, \mathbf{H}) \sim \omega_{i1}} \mathbb{P}(\hat{x}(\mathbf{G}) \in Y_i) = \mathbb{P}_{\mathbf{G} \sim P_i} (\hat{x}(\mathbf{G}) \in Y_i) \geqslant 1 - \eta.$$

Upper bounding the right side of Eq. (C.15) by Cauchy-Schwartz inequality, we have

$$\mathbb{E}_{(\mathbf{G}, \mathbf{H}) \sim \omega_{i1}} \exp(\varepsilon \cdot \text{Ham}(\mathbf{G}, \mathbf{H})) \cdot \mathbb{P}(\hat{x}(\mathbf{H}) \in Y_i)$$

$$\leqslant \left( \mathbb{E}_{(\mathbf{G}, \mathbf{H}) \sim \omega_{i1}} \exp(2\varepsilon \cdot \text{Ham}(\mathbf{G}, \mathbf{H})) \right)^{1/2} \cdot \left( \mathbb{E}_{(\mathbf{G}, \mathbf{H}) \sim \omega_{i1}} \mathbb{P}(\hat{x}(\mathbf{H}) \in Y_i)^2 \right)^{1/2}$$

$$= \left( \mathbb{E}_{\mathbf{X} \sim \text{Binomial}(N_{i1}, p)} \exp(2\varepsilon \cdot \mathbf{X}) \right)^{1/2} \cdot \left( \mathbb{E}_{\mathbf{H} \sim P_1} \mathbb{P}(\hat{x}(\mathbf{H}) \in Y_i)^2 \right)^{1/2}.$$

Using the formula for the moment generating function of binomial distributions, we have

$$\mathbb{E}_{\mathbf{X} \sim \text{Binomial}(N_{i1}, p)} \exp(2\varepsilon \cdot \mathbf{X}) = (1 - p + p \cdot e^{2\varepsilon})^{N_{i1}},$$

and it is easy to see

$$\mathbb{E}_{\mathbf{H} \sim P_1} \mathbb{P}(\hat{x}(\mathbf{H}) \in Y_i)^2 = \mathbb{E}_{\mathbf{H} \sim P_1} \left( \mathbb{E} \, \mathbb{1}_{[\hat{x}(\mathbf{H}) \in Y_i]} \right)^2 \leqslant \mathbb{P}_{\mathbf{H} \sim P_1} (\hat{x}(\mathbf{H}) \in Y_i).$$

Putting things together, Eq. (C.15) implies for each $i \in [m]$ that

$$\mathbb{P}_{\mathbf{H} \sim P_1} (\hat{x}(\mathbf{H}) \in Y_i) \geqslant \frac{(1 - \eta)^2}{(1 - p + p \cdot e^{2\varepsilon})^{N_{i1}}}. \tag{C.16}$$

---

[31]In Eq. (C.14), the randomness only comes from the algorithm.

Since $x^i \in B_{\mathrm{err}}(x, 4\zeta)$ for $i \in [m]$, by assuming $\zeta \leqslant 1/16$, we have

$$N_{i1} = \mathrm{Ham}(x^i, x^1) \cdot (n - \mathrm{Ham}(x^1, x^i)) \leqslant 8\zeta n(n - 8\zeta n). \tag{C.17}$$

Recalling $p = 2\gamma d/n$ and combining Eq. (C.12), Eq. (C.13), Eq. (C.16) and Eq. (C.17), we have

$$(m-1) \cdot \frac{(1-\eta)^2}{(1 - p + p \cdot e^{2\varepsilon})^{8\zeta n(n - 8\zeta n)}} \leqslant \eta.$$

By taking logarithm on both sides, using $t \geqslant \log(1 + t)$ for any $t > -1$, and assuming $\zeta \leqslant 1/(8e)$, we have

$$e^{2\varepsilon} - 1 \gtrsim \frac{\log \frac{1}{8e\zeta}}{\gamma d} + \frac{\log \frac{1}{\eta}}{\zeta n \gamma d}.$$

$\square$

# D  Private algorithms for learning mixtures of spherical Gaussians

In this section we present a private algorithm for recovering the centers of a mixtures of $k$ Gaussians (cf. Model 1.2). Let $\mathcal{Y} \subseteq (\mathbb{R}^d)^{\otimes n}$ be the collection of sets of $n$ points in $\mathbb{R}^d$. We consider the following notion of adjacency.

**Definition D.1** (Adjacent datasets). *Two datasets $Y, Y' \in \mathcal{Y}$ are said to be adjacent if $|Y \cap Y'| \geqslant n - 1$.*

**Remark D.2** (Problem parameters as public information). *We consider the parameters $n, k, \Delta$ to be public information given as input to the algorithm.*

Next we present the main theorem of the section.

**Theorem D.3** (Privately learning spherical mixtures of Gaussians). *Consider an instance of Model 1.2. Let $t \in \mathbb{N}$ be such that $\Delta \geqslant O\left(\sqrt{t}k^{1/t}\right)$. For $n \geqslant \Omega\left(k^{O(1)} \cdot d^{O(t)}\right)$, $k \geqslant (\log n)^{1/5}$, there exists an algorithm, running in time $(nd)^{O(t)}$, that outputs vectors $\hat{\boldsymbol{\mu}}_1, \ldots, \hat{\boldsymbol{\mu}}_\ell$ satisfying*

$$\max_{\ell \in [k]} \left\| \hat{\boldsymbol{\mu}}_\ell - \mu_{\pi(\ell)} \right\|_2 \leqslant O(k^{-12}),$$

*with high probability, for some permutation $\pi : [k] \to [k]$.[32] Moreover, for $\varepsilon \geqslant k^{-10}, \delta \geqslant n^{-10}$, the algorithm is $(\varepsilon, \delta)$-differentially private for any input $Y$.*

We remark that our algorithm not only works for mixtures of Gaussians but for all mixtures of $2t$-explicitly bounded distributions (cf. Definition A.19).

Our algorithm is based on the sum-of-squares hierarchy and at the heart lies the following sum-of-squares program. The indeterminates $z_{11}, \ldots, z_{1k}, \ldots, z_{nk}$ and vector-valued indeterminates $\mu'_1, \ldots, \mu'_k$, will be central to the proof of Theorem D.3. Let $n, k, t$ be fixed parameters.

$$\begin{cases} z_{i\ell}^2 = z_{i\ell} & \forall i \in [n], \ell \in [k] \quad \text{(indicators)} \\ \displaystyle\sum_{\ell \in [k]} z_{i\ell} \leqslant 1 & \forall i \in [n] \quad \text{(cluster mem.)} \\ z_{i\ell} \cdot z_{i\ell'} = 0 & \forall i \in [n], \ell \in [k] \quad \text{(uniq. mem.)} \\ \displaystyle\sum_i z_{i\ell} \leqslant n/k & \forall \ell \in [k] \quad \text{(size of clusters)} \\ \mu'_\ell = \dfrac{k}{n} \displaystyle\sum_i z_{i\ell} \cdot y_i & \forall \ell \in [k] \quad \text{(means of clusters)} \\ \forall v \in \mathbb{R}^d: \dfrac{k}{n} \displaystyle\sum_{i=1}^n z_{i\ell}\langle y_i - \mu'_\ell, v\rangle^{2s} + \left\| Q v^{\otimes s} \right\|^2 = (2s)^s \cdot \|v\|_2^{2s} & \forall s \leqslant t, \ell \in [k] \quad \text{($t$ moment)} \end{cases}$$

$$(\mathcal{P}_{n,k,t}(Y))$$

---

[32]We remark that we chose constants to optimize readability and not the smallest possible ones.

We remark that the moment constraint encodes the $2t$-explicit 2-boundedness constraint introduced in Definition A.19. Note that in the form stated above there are infinitely many constraints, one for each vector $v$. This is just for notational convenience. This constraint postulates equality of two polynomials in $v$. Formally, this can also be encoded by requiring there coefficients to agree and hence eliminating the variable $v$. It is not hard to see that this can be done adding only polynomially many constraints. Further, the matrix variable $Q$ represents the SOS proof of the $2t$-explicit 2-boundedness constraint and we can hence deduce that for all $0 \leqslant s \leqslant t$

$$\mathcal{P} \Big|\frac{v}{2s} \left\{ \frac{k}{n} \sum_{i=1}^{n} z_{i\ell} \langle y_i - \mu'_\ell, v \rangle^{2s} \leqslant (2s)^s \|s\|_2^{2s} \right\} .$$

Before presenting the algorithm we will introduce some additional notation which will be convenient. We assume $t, n, k$ to be *fixed* throughout the section and drop the corresponding subscripts. For $Y \in \mathcal{Y}$, let $\mathcal{Z}(Y)$ be the set of degree-$10t$ pseudo-distributions satisfying $\mathcal{P}(Y)$. For each $\zeta \in \mathcal{Z}(Y)$ define $W(\zeta)$ as the $n$-by-$n$ matrix satisfying

$$W(\zeta)_{ij} = \tilde{\mathbb{E}}_\zeta \left[ \sum_{\ell \in [k]} z_{i\ell} \cdot z_{j\ell} \right] .$$

We let $\mathcal{W}(Y) := \{ W(\zeta) \mid \zeta \in \mathcal{Z}(Y) \}$ .

Recall that $J$ denotes the all-ones matrix. We define the function $g : \mathbb{R}^{n \times n} \to \mathbb{R}$ as

$$g(W) = \|W\|_{\mathrm{F}}^2 - (10)^{10} k^{300} \langle J, W \rangle$$

and let

$$W(\hat{\zeta}(Y)) := \mathrm{argmin}_{W \in \mathcal{W}(Y)} g(W) .$$

We also consider the following function

**Definition D.4** (Soft thresholding function). *We denote by $\phi : [0, 1] \to [0, 1]$ the function*

$$\phi(x) = \begin{cases} 0 & \text{if } x \leqslant 0.8 \,, \\ 1 & \text{if } x \geqslant 0.9 \,, \\ \frac{x - 0.8}{0.9 - 0.8} & \text{otherwise} \,. \end{cases}$$

Notice that $\phi(\cdot)$ is $\frac{1}{0.9 - 0.8} = 10$ Lipschitz. Next we introduce our algorithm. Notice the algorithm relies on certain private subroutines. We describe them later in the section to improve the presentation.

**Algorithm D.5** (Private algorithm for learning mixtures of Gaussians).
**Input:** Set of $n$ points $Y \subseteq \mathbb{R}^d$ , $\varepsilon, \delta > 0$ , $k, t \in \mathbb{N}$ , $d^* = 100 \log n$ , $b = k^{-15}$ .

1. Compute $W = W(\hat{\zeta}(Y))$.

2. Pick $\boldsymbol{\tau} \sim tLap\left(-n^{1.6}\left(1 + \frac{\log(1/\delta)}{\varepsilon}\right), \frac{n^{1.6}}{\varepsilon}\right)$.

3. If $|\boldsymbol{\tau}| \geqslant n^{1.7}$ or $\|\phi(W)\|_1 \leqslant \frac{n^2}{k} \cdot \left(1 - \frac{1}{n^{0.1}} - \frac{1}{k^{100}}\right) + \boldsymbol{\tau}$ reject.

4. For all $i \in [n]$ , compute the $n$-dimensional vector

$$\nu^{(i)} = \begin{cases} \mathbf{0} & \text{if } \|\phi(W_i)\|_1 = 0 \\ \|\phi(W_i)\|_1^{-1} \sum_j \phi(W_{ij}) \cdot y_j & \text{otherwise.} \end{cases}$$

5. Pick a set $\boldsymbol{\mathcal{S}}$ of $n^{0.01}$ indices $i \in [n]$ uniformly at random.

6. For each $i \in \boldsymbol{\mathcal{S}}$ let $\bar{\boldsymbol{\nu}}^{(i)} = \nu^{(i)} + \mathbf{w}$ where $\mathbf{w} \sim N\left(0, n^{-0.18} \cdot \frac{\log(2/\delta)}{\varepsilon^2} \cdot \text{Id}\right)$ .

7. Pick $\boldsymbol{\Phi} \sim N\left(0, \frac{1}{d^*}\right)^{d^* \times d}$ , $\mathbf{q} \overset{u.a.r.}{\sim} [0, b]$ and run the histogram learner of Lemma A.13 with input $\boldsymbol{\Phi}\bar{\boldsymbol{\nu}}^{(1)}, \ldots, \boldsymbol{\Phi}\bar{\boldsymbol{\nu}}^{(n^{0.01})}$ and parameters

$$\mathbf{q}, b, \alpha = k^{-10}, \beta = n^{-10}, \delta^* = \frac{\delta}{n}, \varepsilon^* = \varepsilon \cdot \frac{10k^{50}}{n^{0.01}} .$$

Let $\mathbf{B}_1, \ldots, \mathbf{B}_k$ be the resulting $d^*$-dimensional bins with highest counts. Break ties randomly.

8. Reject if $\min_{i \in [k]} \left|\left\{j \mid \boldsymbol{\Phi}\bar{\boldsymbol{\nu}}^{(j)} \in \mathbf{B}_i\right\}\right| < \frac{n^{0.01}}{2k}$.

9. For each $l \in [k]$ output

$$\hat{\boldsymbol{\mu}}_l := \frac{1}{\left|\left\{j \mid \boldsymbol{\Phi}\bar{\boldsymbol{\nu}}^{(j)} \in \mathbf{B}_i\right\}\right|} \cdot \left(\sum_{\boldsymbol{\Phi}\bar{\boldsymbol{\nu}}^{(j)} \in \mathbf{B}_l} \bar{\boldsymbol{\nu}}^{(j)}\right) + \mathbf{w}' ,$$

where $\mathbf{w}' \sim N\left(0, N\left(0, 32 \cdot k^{-120} \cdot \frac{\log(2kn/\delta)}{\varepsilon^2} \cdot \text{Id}\right)\right)$.

For convenience, we introduce some preliminary facts.

**Definition D.6** (Good $Y$). *Let $\mathbf{Y}$ be sampled according to Model 1.2. We say that $\mathbf{Y}$ is good if:*

1. *for each $\ell \in [k]$, there are at least $\frac{n}{k} - n^{0.6}$ and most $\frac{n}{k} + n^{0.6}$ points sampled from $D_\ell$ in $\mathbf{Y}$. Let $\mathbf{Y}_\ell \subseteq \mathbf{Y}$ be such set of points.*

2. *Each $\mathbf{Y}_\ell$ is $2t$-explicitly 2-bounded.*

It turns out that typical instances $\mathbf{Y}$ are indeed good.

**Lemma D.7** ([30, 36]). *Consider the settings of Theorem D.3. Then $\mathbf{Y}$ is good with high probability. Further, in this case the sets $\mathcal{Z}(Y)$ and $\mathcal{W}(Y)$ are non-empty.*

## D.1 Privacy analysis

In this section we show that our clustering algorithm is private.

**Lemma D.8** (Differential privacy of the algorithm). *Consider the settings of Theorem D.3. Then Algorithm D.5 is $(\varepsilon, \delta)$-differentially private.*

We split our analysis in multiple steps and combine them at the end. On a high level, we will argue that on adjacent inputs $Y, Y'$ many of the vectors $\nu^{(i)}$ by the algorithm are close to each other and a small part can be very far. We can then show that we can mask this small difference using the

Gaussian mechanism and afterwards treat this subset of the vectors as privatized (cf. Lemma I.4). Then we can combine this with known histogram learners to deal with the small set of $\nu^{(i)}$'s that is far from each other on adjacent inputs.

### D.1.1 Sensitivity of the matrix W

Here we use Lemma B.1 to reason about the sensitivity of $\phi(W(\hat{\zeta}(Y)))$. For adjacent datasets $Y, Y' \in \mathcal{Y}$ we let $\hat{\zeta}, \hat{\zeta}'$ be the pseudo-distribution corresponding to $W(\hat{\zeta}(Y))$ and $W(\hat{\zeta}(Y'))$ computed in step 1 of the algorithm, respectively. We prove the following result.

**Lemma D.9** ($\ell_1$-sensitivity of $\phi(W)$). *Consider the settings of Theorem D.3. Let $W, W'$ be respectively be the matrices computed in step 1 by Algorithm D.5 on adjacent inputs $Y, Y' \in \mathcal{Y}$. Then*

$$\|\phi(W) - \phi(W')\|_1 \leqslant n^{1.6}.$$

*For all but $n^{0.8}$ rows $i$ of $\phi(W), \phi(W')$, it holds*

$$\|\phi(W)_i - \phi(W')_i\|_1 \leqslant n^{0.8}.$$

*Proof.* The second inequality is an immediate consequence of the first via Markov's inequality. Thus it suffices to prove the first. Since $\phi(\cdot)$ is 10-Lipschitz, we immediately obtain the result if

$$\left\|W(\hat{\zeta}(Y)) - W(\hat{\zeta}(Y'))\right\|_1 \leqslant n^{1.55}.$$

Thus we focus on this inequality. To prove it, we verify the two conditions of Lemma B.1. First notice that $g$ is 2-strongly convex with respect to its input $W$. Indeed for $W, W' \in \mathcal{W}(Y)$, since $\forall i, j \in [n], W_{ij} \geqslant 0$ it holds that

$$
\begin{aligned}
\|W'\|_F^2 &= \|W\|_F^2 + \|W - W'\|_F^2 + 2\langle W' - W, W\rangle \\
&= \|W\|_F^2 + \|W - W'\|_F^2 + 2\langle W' - W, W\rangle + \langle W' - W, (10)^{10}k^{300}(J - J)\rangle \\
&= g(W) + \|W - W'\|_F^2 + \langle W' - W, \nabla g(W)\rangle + \langle W', (10)^{10}k^{300}J\rangle,
\end{aligned}
$$

where we used that $\nabla g(W) = 2W - (10)^{10}k^{300}J$. Thus it remain to prove *(i)* of Lemma B.1.

Let $\hat{\zeta} \in \mathcal{Z}(Y), \hat{\zeta}' \in \mathcal{Z}(Y')$ be the pseudo-distributions such that $W_Y(\hat{\zeta}) = W$ and $W_Y(\hat{\zeta}') = W'$. We claim that there always exists $\zeta_{\text{adj}} \in \mathcal{Z}(Y) \cap \mathcal{Z}(Y')$ such that

1. $|g(W(\zeta)) - g(W(\zeta_{\text{adj}}))| \leqslant \frac{2n}{k} \cdot \left((10)^{10}k^{300} + 1\right) \leqslant 3 \cdot (10)^{10}k^{300}n$,

2. $|g_{Y'}(W(\zeta_{\text{adj}})) - g(W(\zeta_{\text{adj}}))| = 0$.

Note that in this case the second point is always true since $g$ doesn't depend on $Y$. Together with Lemma B.1 these two inequalities will imply that

$$\left\|W(\hat{\zeta}(Y)) - W(\hat{\zeta}(Y'))\right\|_F^2 \leqslant 18 \cdot (10)^{10}k^{300}n.$$

By assumption on $n$, an application of Cauchy-Schwarz will give us the desired result.

So, let $i$ be the index at which $Y, Y'$ differ. We construct $\zeta_{\text{adj}}$ as follows: for all polynomials $p$ of degree at most $10t$ we let

$$\tilde{\mathbb{E}}_{\zeta_{\text{adj}}}[p] = \begin{cases} \tilde{\mathbb{E}}_{\zeta}[p] & \text{if } p \text{ does not contain variables } z_{i\ell} \text{ for any } \ell \in [k] \\ 0 & \text{otherwise.} \end{cases}$$

By construction $\zeta_{\text{adj}} \in \mathcal{Z}(Y) \cap \mathcal{Z}(Y')$. Moreover, $W(\zeta), W(\zeta_{\text{adj}})$ differ in at most $2n/k$ entries. Since all entries of the two matrices are in $[0, 1]$, the first inequality follows by definition of the objective function. $\qquad\square$

### D.1.2 Sensitivity of the resulting vectors

In this section we argue that if the algorithm does not reject in step 3 then the vectors $\nu^{(i)}$ are stable on adjacent inputs. Concretely our statement goes as follows:

**Lemma D.10** (Stability of the $\nu^{(i)}$'s)**.** *Consider the settings of Theorem D.3. Suppose Algorithm D.5 does not reject in step 3, on adjacent inputs $Y$, $Y' \in \mathcal{Y}$. Then for all but $\frac{6n}{k^{50}}$ indices $i \in [n]$, it holds:*

$$\left\| \nu_Y^{(i)} - \nu_{Y'}^{(i)} \right\|_2 \leqslant O\left(n^{-0.1}\right).$$

The proof of Lemma D.10 crucially relies on the next statement.

**Lemma D.11** (Covariance bound)**.** *Consider the settings of Theorem D.3. Let $W$ be the matrix computed by Algorithm D.5 on input $Y \in \mathcal{Y}$. For $i \in [n]$, if $\|\phi(W_i)\|_1 \geqslant \frac{n}{k} \cdot \left(1 - \frac{10}{k^{50}}\right)$ then $\nu^{(i)}$ induces a 2-explicitly 40-bounded distribution over $Y$.*

*Proof.* First, by assumption notice that there must be at least $\frac{n}{k} \cdot \left(1 - \frac{10}{k^{50}}\right)$ entries of $\phi(W_i)$ larger than 0.8. We denote the set of $j \in [n]$ such that $W_{ij} \geqslant 0.8$ by $\mathcal{G}$. Let $\zeta \in \mathcal{Z}(Y)$ be the degree $10t$ pseudo-distribution so that $W = W(\zeta(Y))$. Since $\zeta$ satisfies $\mathcal{P}(Y)$, for $\ell \in [k]$ it follows from the moment bound constraint for $s = 1$ that for all unit vectors $u$ it holds that

$$\mathcal{P} \mathrel{\Big|\frac{}{4}} \left\{ 0 \leqslant \frac{k}{n} \sum_{j=1}^{n} z_{j\ell} \langle \mathbf{y}_j - \mu'_l, u \rangle^2 \leqslant 2 \right\},$$

Using the SOS triangle inequality (cf. Fact I.2) $\mathrel{\Big|\frac{a,b}{2}} (a+b)^2 \leqslant 2(a^2 + b^2)$ it now follows that

$$\mathbf{0} \preceq \tilde{\mathbb{E}}_\zeta \left[ \frac{k^2}{n^2} \sum_{j,j' \in [n]} z_{j\ell} z_{j'\ell} \cdot (y_j - y_{j'})^{\otimes 2} \right] \preceq 8\mathrm{Id}$$

and thus

$$\mathbf{0} \preceq \tilde{\mathbb{E}}_\zeta \left[ \frac{k^2}{n^2} \sum_{\ell \in [k]} \sum_{j,j' \in [n]} z_{i\ell} z_{j\ell} z_{j'\ell} \cdot (y_j - y_{j'})^{\otimes 2} \right] \preceq 8\mathrm{Id}.$$

Furthermore using $\mathcal{P}(Y) \mathrel{\big|\frac{}{2}} \{z_{i\ell} z_{i\ell'} = 0\}$ for $\ell \neq \ell'$ we have

$$\tilde{\mathbb{E}}_\zeta \left[ \sum_{\ell \in [k]} \sum_{j,j' \in [n]} z_{i\ell} z_{j\ell} z_{j'\ell} \right] = \tilde{\mathbb{E}}_\zeta \left[ \left( \sum_{\ell \in [k],\, j \in [n]} z_{i\ell} z_{j\ell} \right) \cdot \left( \sum_{\ell' \in [k],\, j' \in [n]} z_{i\ell'} z_{j'\ell'} \right) \right].$$

Now, for fixed $j, j' \in [n]$, using

$$\left\{ a^2 = a,\, b^2 = b \right\} \mathrel{\big|\frac{}{O(1)}} \left\{ 1 + ab - a - b = 1 - ab - (a-b)^2 \geqslant 0 \right\}$$

with $a = \sum_{\ell \in [k]} z_{i\ell} z_{j\ell}$ and $b = \sum_{\ell' \in [k]} z_{i\ell'} z_{j'\ell'}$ we get

$$\tilde{\mathbb{E}}_\zeta \left[ \left( \sum_{\ell \in [k]} z_{i\ell} z_{j\ell} \right) \left( \sum_{\ell' \in [k]} z_{i\ell'} z_{j'\ell'} \right) \right] \geqslant \tilde{\mathbb{E}}_\zeta \left[ \sum_{\ell \in [k]} z_{i\ell} z_{j\ell} + \sum_{\ell' \in [k]} z_{i\ell'} z_{j'\ell'} \right] - 1$$
$$= W_{ij} + W_{ij'} - 1.$$

Now if $j, j' \in \mathcal{G}$ we must have

$$\sum_{\ell \in [k]} \tilde{\mathbb{E}}_\zeta \left[ z_{i\ell} z_{j\ell} z_{j'\ell} \right] = \tilde{\mathbb{E}}_\zeta \left[ \left( \sum_{\ell \in [k]} z_{i\ell} z_{j\ell} \right) \left( \sum_{\ell' \in [k]} z_{i\ell'} z_{j'\ell'} \right) \right] \geqslant 0.6.$$

Since $\phi(W_{ij}) \leqslant 1$ by definition and $\|\phi(W_i)\|_1 \geqslant \frac{n}{k} \cdot \left(1 - \frac{10}{k^{50}}\right)$, we conclude

$$\|\phi(W_i)\|_1^{-2} \left[ \sum_{j,j' \in [n]} \phi(W_{ij}) \phi(W_{ij'}) (y_j - y_{j'})^{\otimes 2} \right]$$

$$\preceq 5 \cdot \frac{k^2}{n^2} \sum_{j,j' \in [n], \ell \in [k]} \tilde{\mathbb{E}}_\zeta \left[ z_{i\ell} z_{j\ell} z_{j'\ell} \right] \cdot (y_j - y_{j'})^{\otimes 2}$$

$$\preceq 40 \mathrm{Id} \,.$$

as desired. $\qquad \square$

We can now prove Lemma D.10.

*Proof of Lemma D.10.* Let $W, W'$ be the matrices computed by Algorithm D.5 in step 1 on input $Y, Y'$, respectively. Let $\mathcal{G} \subseteq [n]$ be the set of indices $i$ such that

$$\left\| \phi(W)_i - \phi(W')_i \right\|_1 \leqslant n^{0.8} \,.$$

Notice that $|\mathcal{G}| \geqslant n - n^{0.8}$ by Lemma D.9. Since on input $Y$ the algorithm did not reject in step 3 we must have

$$\| \phi(W) \|_1 \geqslant \frac{n^2}{k} \cdot \left( 1 - \frac{1}{n^{0.1}} - \frac{1}{k^{100}} \right) - n^{1.7} \geqslant \frac{n^2}{k} \cdot \left( 1 - \frac{2}{k^{100}} \right) \,.$$

Let $g_W$ be the number of indices $i \in \mathcal{G}$ such that $\| \phi(W)_i \|_1 \geqslant \frac{n}{k} \cdot \left( 1 - \frac{1}{k^{50}} \right)$. It holds that

$$\frac{n^2}{k} \cdot \left( 1 - \frac{2}{k^{100}} \right) \leqslant g_W \cdot \frac{n}{k} + (n - |\mathcal{G}|) \cdot \frac{n}{k} + (|G| - g_w) \frac{n}{k} \cdot \left( 1 - \frac{1}{k^{50}} \right)$$

$$\leqslant g_W \cdot \frac{n}{k} \cdot \frac{1}{k^{50}} + \frac{n^{1.8}}{k} + \frac{n^2}{k} \cdot \left( 1 - \frac{1}{k^{50}} \right)$$

$$\leqslant g_W \cdot \frac{n}{k} \cdot \frac{1}{k^{50}} + \frac{n^2}{k} \cdot \left( 1 + \frac{1}{k^{100}} - \frac{1}{k^{50}} \right) \,.$$

Rearring now yields

$$g_W \geqslant n \cdot \left( 1 - \frac{3}{k^{50}} \right) \,.$$

Similarly, let $g_{W'}$ be the number of indices $i \in \mathcal{G}$ such that $\| \phi(W')_i \|_1 \geqslant \frac{n}{k} \cdot \left( 1 - \frac{1}{k^{50}} \right)$. By an analogous argument it follows that $g_{W'} \geqslant n \cdot \left( 1 - \frac{3}{k^{50}} \right)$. Thus, by the pigeonhole principle there are at least $g_W \geqslant n \cdot \left( 1 - \frac{6}{k^{50}} \right)$ indices $i$ such that

1. $\| \phi(W)_i \|_1 \geqslant \frac{n}{k} \left( 1 - \frac{1}{k^{50}} \right)$ ,

2. $\| \phi(W')_i \|_1 \geqslant \frac{n}{k} \left( 1 - \frac{1}{k^{50}} \right)$ ,

3. $\| \phi(W)_i - \phi(W')_i \|_1 \leqslant n^{0.8}$ .

Combining these with Lemma D.11 we may also add

4. the distribution induced by $\| \phi(W_i) \|_1^{-1} \phi(W_i)$ is 2-explicitly 40-bounded,

5. the distribution induced by $\| \phi(W'_i) \|_1^{-1} \phi(W'_i)$ is 2-explicitly 40-bounded.

Using that for non-zero vectors $x, y$ it holds that $\left\| \frac{x}{\|x\|} - \frac{y}{\|y\|} \right\| \leqslant \frac{2}{\|x\|} \|x - y\|$ points 1 to 3 above imply that

$$\left\| \| \phi(W_i) \|_1^{-1} \phi(W_i) - \| \phi(W'_i) \|_1^{-1} \phi(W'_i) \right\|_1 \leqslant \frac{2 n^{0.8}}{\frac{n}{k} \cdot \left( 1 - \frac{1}{k^{50}} \right)} = O\left( n^{-0.2} \right) \,.$$

Hence, applying Theorem A.21 with $t = 1$ it follows that

$$\left\| \nu_Y^{(i)} - \nu_{Y'}^{(i)} \right\|_2 \leqslant O\left( n^{-0.1} \right) \,.$$

$\qquad \square$

### D.1.3  From low sensitivity to privacy

In this section we argue privacy of the whole algorithm, proving Lemma D.8. Before doing that we observe that low-sensitivity is preserved with high probability under subsampling.

**Fact D.12** (Stability of $\mathcal{S}$). *Consider the settings of Theorem D.3. Suppose Algorithm D.5 does not reject in step 3, on adjacent inputs $Y, Y' \in \mathcal{Y}$. With probability at least $1 - e^{-n^{\Omega(1)}}$ over the random choices of $\mathcal{S}$, for all but $\frac{10n^{0.01}}{k^{50}}$ indices $i \in \mathcal{S}$, it holds:*

$$\left\| \nu_Y^{(i)} - \nu_{Y'}^{(i)} \right\|_2 \leqslant O\left(n^{-0.1}\right) .$$

*Proof.* There are at most $\frac{6n}{k^{50}}$ such indices in $[n]$ by Lemma D.10. By Chernoff's bound, cf. Fact E.4, the claim follows. $\square$

Finally, we prove our main privacy lemma.

*Proof of Lemma D.8.* For simplicity, we will prove that the algorithm is $(5\varepsilon, 5\delta)$-private. Let $Y, Y' \in \mathcal{Y}$ be adjacent inputs. By Lemma A.10 and Lemma D.9 the test in step 3 of Algorithm D.5 is $(\varepsilon, \delta)$-private.

Thus suppose now the algorithm did not reject in step 3 on inputs $Y, Y'$. By composition (cf. Lemma A.4) it is enough to show that the rest of the algorithm is $(\varepsilon, \delta)$-private with respect to $Y, Y'$ under this condition. Next, let $\nu_Y^{(1)}, \ldots, \nu_Y^{(n)}$ and $\nu_{Y'}^{(1)}, \ldots, \nu_{Y'}^{(n)}$ be the vectors computed in step 4 of the algorithm and $\mathcal{S}$ be the random set of indices computed in step 5.[33] By Lemma D.10 and Fact D.12 with probability $1 - e^{-n^{\Omega(1)}}$ over the random choices of $\mathcal{S}$ we get that for all but $\frac{10n^{0.01}}{k^{50}}$ indices $i \in \mathcal{S}$, it holds that

$$\left\| \nu_Y^{(i)} - \nu_{Y'}^{(i)} \right\|_2 \leqslant O\left(n^{-0.1}\right) .$$

Denote this set of indices by $\mathcal{G}$. Note, that we may incorporate the failure probability $e^{-n^{\Omega(1)}} \leqslant \min\{\varepsilon/2, \delta/2\}$ into the final privacy parameters using Fact I.3.

Denote by $\mathbf{V}, \mathbf{V}'$ the $|\mathcal{S}|$-by-$d$ matrices respectively with rows $\nu_Y^{(i_1)}, \ldots, \nu_Y^{(i_{|\mathcal{S}|})}$ and $\nu_{Y'}^{(i_1)}, \ldots, \nu_{Y'}^{(i_{|\mathcal{S}|})}$, where $i_1, \ldots, i_{|\mathcal{S}|}$ are the indices in $\mathcal{S}$. Recall, that $|\mathcal{G}|$ rows of $\mathbf{V}$ and $\mathbf{V}'$ differ by at most $O\left(n^{-0.1}\right)$ in $\ell_2$-norm. Thus, by the Gaussian mechanism used in step 6 (cf. Lemma A.12) and Lemma I.4 it is enough to show that step 7 to step 9 of the algorithm are private with respect to pairs of inputs $V$ and $V'$ differing in at most 1 row.[34] In particular, suppose these steps are $(\varepsilon_1, \delta_1)$-private. Then, for $m = n^{0.01} - |\mathcal{G}| \leqslant \frac{10n^{0.01}}{k^{50}}$, by Lemma I.4 it follows that step 6 to step 9 are $(\varepsilon', \delta')$-differentially private with

$$\varepsilon' := \varepsilon + m\varepsilon_1 ,$$
$$\delta' := e^{\varepsilon} m e^{(m-1)\varepsilon_1} \delta_1 + \delta .$$

Consider steps 7 and 8. Recall, that in step 7 we invoke the histogram learner with parameters

$$b = k^{-15}, \mathbf{q} \overset{u.a.r.}{\sim} [0, b], \alpha = k^{-10}, \beta = n^{-10}, \delta^* = \frac{\delta}{n}, \varepsilon^* = \varepsilon \cdot \frac{10k^{50}}{n^{0.01}} .$$

Hence, by Lemma A.13 this step is $(\varepsilon^*, \delta^*)$-private since

$$\frac{8}{\varepsilon^* \alpha} \cdot \log\left(\frac{2}{\delta^* \beta}\right) \leqslant \frac{200 \cdot k^{10} \cdot n^{0.01}}{10 \cdot k^{50} \cdot \varepsilon} \cdot \log n = \frac{20 \cdot n^{0.01}}{k^{40} \cdot \varepsilon} \cdot \log n \leqslant n ,$$

for $\varepsilon \geqslant k^{-10}$. Step 8 is private by post-processing.

---

[33]Note that since this does not depend on $Y$ or $Y'$, respectively, we can assume this to be the same in both cases. Formally, this can be shown, e.g., via a direct calculation or using Lemma A.4.

[34]Note that for the remainder of the analysis, these do *not* correspond to $\mathbf{V}$ and $\mathbf{V}'$, since those differ in $m$ rows. Lemma I.4 handles this difference.

Next, we argue that step 9 is private by showing that the average over the bins has small $\ell_2$-sensitivity. By Lemma A.4 we can consider the bins $\mathbf{B}_1, \ldots, \mathbf{B}_k$ computed in the previous step as fixed. Further, we can assume that the algorithm did not reject in step 8, i.e., that each bin contains at least $\frac{n^{0.01}}{2k}$ points of $V$ and $V'$ respectively. As a consequence, every bin contains at least two (projections of) points of the input $V$ or $V'$ respectively. In particular, it contains at least one (projection of a) point which is present in both $V$ and $V'$. Fix a bin $\mathbf{B}_l$ and let $\bar{\nu}^*$ be such that it is both in $V$ and $V'$ and $\Phi\bar{\nu}^* \in \mathbf{B}_l$. Also, define

$$S_l := \left| \left\{ j \mid \Phi\bar{\nu}_Y^{(j)} \in \mathbf{B}_i \right\} \right| ,$$

$$S_l' := \left| \left\{ j \mid \Phi\bar{\nu}_{Y'}^{(j)} \in \mathbf{B}_i \right\} \right| .$$

Assume $V$ and $V'$ differ on index $j$. We consider two cases. First, assume that $\Phi\bar{\nu}_Y^{(j)}$ and $\Phi\bar{\nu}_{Y'}^{(j)}$ both lie in $\mathbf{B}_l$. In this case, $S_l = S_l'$ and using Lemma E.5 it follows that with probability $n^{-100} \leqslant \min\{\varepsilon/2, \delta/2\}$ it holds that

$$\left\| \bar{\nu}_Y^{(j)} - \bar{\nu}_{Y'}^{(j)} \right\|_2 \leqslant \left\| \bar{\nu}_Y^{(j)} - \bar{\nu}^* \right\|_2 + \left\| \bar{\nu}^* - \bar{\nu}_{Y'}^{(j)} \right\| \leqslant 10 \cdot \left( \left\| \Phi\bar{\nu}_Y^{(j)} - \Phi\bar{\nu}^* \right\|_2 + \left\| \Phi\bar{\nu}_{Y'}^{(j)} - \Phi\bar{\nu}^* \right\|_2 \right)$$

$$\leqslant 20 \cdot \sqrt{d^*} \cdot b \leqslant 200 \cdot k^{-12} .$$

And hence we can bound

$$\left\| \frac{1}{S_l} \cdot \left( \sum_{\Phi\bar{\nu}_Y^{(j)} \in \mathbf{B}_l} \bar{\nu}_Y^{(j)} \right) - \frac{1}{S_l'} \cdot \left( \sum_{\Phi\bar{\nu}_{Y'}^{(j)} \in \mathbf{B}_l} \bar{\nu}_{Y'}^{(j)} \right) \right\|_2 \leqslant \frac{\left\| \bar{\nu}_Y^{(j)} - \bar{\nu}_{Y'}^{(j)} \right\|_2}{S_l} \leqslant \frac{400 \cdot k^{-11}}{n^{0.01}} .$$

Next, assume that $\Phi\bar{\nu}_Y^{(j)} \notin \mathbf{B}_l$ and $\Phi\bar{\nu}_{Y'}^{(j)} \in \mathbf{B}_l$ (the other case works symetrically). It follows that $S_l = S_l' - 1$ and we can bound

$$\left\| \frac{1}{S_l} \cdot \left( \sum_{\Phi\bar{\nu}_Y^{(j)} \in \mathbf{B}_l} \bar{\nu}_Y^{(j)} \right) - \frac{1}{S_l'} \cdot \left( \sum_{\Phi\bar{\nu}_{Y'}^{(j)} \in \mathbf{B}_l} \bar{\nu}_{Y'}^{(j)} \right) \right\|_2 = \frac{1}{S_l \cdot S_l'} \cdot \left\| S_l' \left( \sum_{\Phi\bar{\nu}_Y^{(j)} \in \mathbf{B}_l} \bar{\nu}_Y^{(j)} \right) - (S_l' - 1) \left( \sum_{\Phi\bar{\nu}_{Y'}^{(j)} \in \mathbf{B}_l} \bar{\nu}_{Y'}^{(j)} \right) \right\|_2$$

$$= \frac{1}{S_l \cdot S_l'} \cdot \left\| S_l' \cdot \bar{\nu}_{Y'}^{(j)} + \left( \sum_{\Phi\bar{\nu}_{Y'}^{(j)} \in \mathbf{B}_l} \bar{\nu}_{Y'}^{(j)} \right) \right\|_2$$

$$= \frac{1}{S_l} \cdot \left\| \bar{\nu}_{Y'}^{(j)} - \frac{1}{S_l'} \left( \sum_{\Phi\bar{\nu}_{Y'}^{(j)} \in \mathbf{B}_l} \bar{\nu}_{Y'}^{(j)} \right) \right\|_2$$

$$\leqslant \frac{\sqrt{d^*} \cdot b}{S_l} \leqslant \frac{20 \cdot k^{-11}}{n^{0.01}} .$$

Hence, the $\ell_2$-sensitivity is at most $\Delta := \frac{400 \cdot k^{-11}}{n^{0.01}}$. Since

$$2\Delta^2 \cdot \frac{\log(2/(\delta^*/k))}{(\varepsilon^*/k)^2} = 32 \cdot k^{-120} \cdot \frac{\log(2kn/\delta)}{\varepsilon^2}$$

and $\mathbf{w}' \sim N\left( 0, 32 \cdot k^{-120} \cdot \frac{\log(2kn/\delta)}{\varepsilon^2} \cdot \mathrm{Id} \right)$ it follows that outputing $\hat{\boldsymbol{\mu}}_l$ is $(\varepsilon^*/k, \delta^*/k)$-DP by the Gaussian Mechanism that. By Lemma A.4 it follows step 9 is $(\varepsilon^*, \delta^*)$-private.

Hence, by Lemma A.4 it follows that step 7 to step 9 are $(2\varepsilon^*, 2\delta^*)$-differentially private. Using $m \leqslant \frac{10n^{0.01}}{k^{10}}$ it now follows by Lemma I.4 that step 6 to step 9 are $(\varepsilon', \delta')$-private for

$$\varepsilon' = \varepsilon + 2m\varepsilon^* \leqslant 3\varepsilon ,$$

$$\delta' = 2e^\varepsilon m e^{(m-1)2\varepsilon^*} \delta^* + \delta \leqslant 2m e^{3\varepsilon} \cdot \frac{\delta}{n} + \delta \leqslant 3\delta .$$

Thus, combined with the private check and Fact I.3 in step 3 the whole algorithm is $(5\varepsilon, 5\delta)$-private.

$\square$

## D.2 Utility analysis

In this section we reason about the utility of Algorithm D.5 and prove Theorem D.3. We first introduce some notation.

**Definition D.13** (True solution). *Let* $\mathbf{Y}$ *be an input sampled from Model 1.2. Denote by* $W^*(\mathbf{Y}) \in \mathcal{W}(\mathbf{Y})$ *the matrix induced by the true solution (or ground truth). I.e., let*

$$W^*(\mathbf{Y})_{ij} = \begin{cases} 1 & \text{if } i, j \text{ were both sampled from the same component of the mixture,} \\ 0 & \text{otherwise.} \end{cases}$$

*Whenever the context is clear, we simply write* $\mathbf{W}^*$ *to ease the notation.*

First, we show that in the utility case step 3 of Algorithm D.5 rejects only with low probability.

**Lemma D.14** (Algorithm does not reject on good inputs). *Consider the settings of Theorem D.3. Suppose* $\mathbf{Y}$ *is a good set as per Definition D.6. Then* $\left\| W(\hat{\zeta}(\mathbf{Y})) \right\|_1 \geqslant \frac{n^2}{k} \cdot \left( 1 - n^{-0.4} - \frac{1}{(10)^{10}k^{300}} \right)$ *and Algorithm D.5 rejects with probability at most* $\exp\left( -\Omega\left( n^{1.7} \right) \right)$.

*Proof.* Since $\mathbf{Y}$ is good, there exists $\mathbf{W}^* \in \mathcal{W}(\mathbf{Y})$, corresponding to the indicator matrix of the true solution, such that

$$g(\mathbf{W}^*) = \|\mathbf{W}^*\|_{\mathrm{F}}^2 - 10^{10}k^{300}\langle J, \mathbf{W}^* \rangle \leqslant \frac{n^2}{k} + n^{1.6} - (10)^{10}k^{300}\left( \frac{n^2}{k} - n^{1.6} \right)$$

$$= \frac{n^2}{k}\left( 1 + \frac{k}{n^{0.4}} - (10)^{10}k^{300}\left( 1 - \frac{k}{n^{0.4}} \right) \right) \,.$$

Since $g(W(\hat{\zeta}(\mathbf{Y}))) \leqslant g(\mathbf{W}^*)$ it follows that

$$(10)^{10}k^{300}\langle J, W(\hat{\zeta}(\mathbf{Y})) \rangle \geqslant |g(W(\hat{\zeta}(\mathbf{Y})))| \geqslant \frac{n^2}{k}\left( (10)^{10}k^{300}\left( 1 - \frac{k}{n^{0.4}} \right) - 1 - \frac{k}{n^{0.4}} \right) \,.$$

Since, $\left\| W(\hat{\zeta}(\mathbf{Y})) \right\|_1 \geqslant \langle J, W(\hat{\zeta}(\mathbf{Y})) \rangle$ the first claim follows rearranging the terms. This means that the algorithm rejects only if $|\boldsymbol{\tau}| \geqslant n^{1.7}$. Recall that $\boldsymbol{\tau} \sim \mathrm{tLap}\left( -n^{1.6}\left( 1 + \frac{\log(1/\delta)}{\varepsilon} \right), \frac{n^{1.6}}{\varepsilon} \right)$. Hence, by Lemma A.11 it follows that

$$\mathbb{P}\left( |\boldsymbol{\tau}| \geqslant n^{1.7} \right) \leqslant \frac{\exp\left( -n^{1.7} + \varepsilon + \log(1/\delta) \right)}{2 - \exp\left( -\varepsilon - \log(1/\delta) \right)} = \exp\left( -\Omega\left( n^{1.7} \right) \right) \,.$$

$\square$

The next step shows that on a good input $\mathbf{Y}$ the matrix $\phi(W(\hat{\zeta}(\mathbf{Y})))$ is close to the true solution.

**Lemma D.15** (Closeness to true solution on good inputs). *Consider the settings of Theorem D.3. Suppose* $\mathbf{Y}$ *is a good set as per Definition D.6. Let* $W(\mathbf{Y}) \in \mathcal{W}(\mathbf{Y})$ *be the matrix computed by Algorithm D.5. Suppose the algorithm does not reject. Then*

$$\|\phi(W(\mathbf{Y})) - \mathbf{W}^*\|_1 \leqslant \frac{n^2}{k} \cdot \frac{3}{k^{98}} \,.$$

The proof is similar to the classical utility analysis of the sum-of-squares program found, e.g., in [30, 26]. We defer it to Appendix I.

Together, the above results imply that the vectors $\nu^{(i)}$ computed by the algorithm are close to the true centers of the mixture.

**Lemma D.16** (Closeness to true centers). *Consider the settings of Theorem D.3. Suppose* $\mathbf{Y}$ *is a good set as per Definition D.6. Let* $\mathbf{W} \in \mathcal{W}(\mathbf{Y})$ *be the matrix computed by Algorithm D.5. Suppose the algorithm does not reject in step 3. Then for each* $\ell \in [k]$, *there exists* $\frac{n}{k} \cdot \left( 1 - \frac{2}{k^{47}} \right)$ *indices* $i \in [n]$, *such that*

$$\left\| \nu^{(i)}(\mathbf{W}) - \mu_\ell \right\|_2 \leqslant O\left( k^{-25} \right) \,.$$

*Proof.* We aim to show that for most indices $i \in [n]$ the vectors $\|\phi(\mathbf{W}_i)\|_1^{-1} \phi(\mathbf{W}_i)$ and $\|\mathbf{W}_i^*\|_1^{-1} \mathbf{W}_i^*$ induce a 2-explicitly 40-bounded distribution over $\mathbf{Y}$. If additionally the two vectors are close in $\ell_1$-norm, the result will follow by Theorem A.21.

Note that $\|\mathbf{W}_i^*\|_1^{-1} \mathbf{W}_i^*$ induces a 2-explicitly 40-bounded distribution by Lemma D.7. By Markov's inequality and Lemma D.15 there can be at most $n/k^{48}$ indices $j \in [n]$ such that

$$\left\| \phi(\mathbf{W})_j - \mathbf{W}_j^* \right\|_1 \geqslant \frac{n}{k} \cdot \frac{3}{k^{50}} \ .$$

Consider all remaining indices $i$. It follows that

$$\|\phi(\mathbf{W}_i)\|_1 \geqslant \|\mathbf{W}_i^*\|_1 - \|\phi(\mathbf{W})_i - \mathbf{W}_i^*\|_1 \geqslant \frac{n}{k} \cdot \left( 1 - \frac{k}{n^{0.4}} - \frac{3}{k^{50}} \right) \geqslant \frac{n}{k} \cdot \left( 1 - \frac{10}{k^{50}} \right) \ .$$

Hence, by Lemma D.11 the distribution induced by $\|\phi(\mathbf{W}_i)\|_1^{-1} \phi(\mathbf{W}_i)$ is 2-explicitly 40-bounded distribution. Further, using $\|\mathbf{W}_i^*\|_1 \geqslant \frac{n}{k} \left( 1 - \frac{k}{n^{0.4}} \right)$ we can bound

$$\left\| \|\phi(\mathbf{W}_i)\|_1^{-1} \phi(\mathbf{W}_i) - \|\mathbf{W}_i^*\|_1^{-1} \mathbf{W}_i^* \right\|_1 = \|\phi(\mathbf{W}_i)\|_1^{-1} \|\mathbf{W}_i^*\|_1^{-1} \cdot \left\| \|\mathbf{W}_i^*\|_1 \phi(\mathbf{W}_i) - \|\phi(\mathbf{W}_i)\|_1 \mathbf{W}_i^* \right\|_1$$

$$\leqslant \|\phi(\mathbf{W}_i)\|_1^{-1} \|\mathbf{W}_i^*\|_1^{-1} \cdot \left( |\|\phi(\mathbf{W}_i)\|_1 - \|\mathbf{W}_i^*\|_1| \cdot \|\phi(\mathbf{W}_i)\|_1 + \|\phi(\mathbf{W}_i)\|_1 \cdot \|\phi(\mathbf{W}_i) - \mathbf{W}_i^*\|_1 \right)$$

$$\leqslant \|\mathbf{W}_i^*\|_1^{-1} \cdot 2 \|\phi(\mathbf{W}_i) - \mathbf{W}_i^*\|_1 \leqslant \frac{6}{k^{50} \cdot \left( 1 - \frac{k}{n^{0.4}} \right)} \leqslant \frac{7}{k^{50}} \ .$$

Hence, by Theorem A.21 for each $l \in [k]$ there are at least $\frac{n}{k} - n^{0.6} - \frac{n}{k^{48}} \geqslant \frac{n}{k} \cdot \left( 1 - \frac{2}{k^{47}} \right)$ indices $i$ such that

$$\left\| \nu^{(i)}(\mathbf{W}) - \|\mathbf{W}_i^*\|_1^{-1} \sum_{j=1}^n \mathbf{W}_{i,j}^* \mathbf{y}_j \right\|_2 \leqslant O\left( k^{-25} \right) \ .$$

The result now follows by standard concentration bounds applied to the distribution induced by $\|\mathbf{W}_i^*\|_1^{-1} \mathbf{W}_i^*$. $\qquad\square$

An immediate consequence of Lemma D.16 is that the vectors $\bar{\boldsymbol{\nu}}^{(i)}$ inherits the good properties of the vectors $\nu^{(i)}$ with high probability.

**Corollary D.17** (Closeness to true centers after sub-sampling)**.** *Consider the settings of Theorem D.3. Suppose $\mathbf{Y}$ is a good set as per Definition D.6. Let $\mathbf{W} \in \mathcal{W}(\mathbf{Y})$ be the matrix computed by Algorithm D.5. Suppose the algorithm does not reject. Then with high probability for each $\ell \in [k]$, there exists $\frac{n^{0.01}}{k} \cdot \left( 1 - \frac{150}{k^{47}} \right)$ indices $i \in \mathcal{S}$, such that*

$$\left\| \bar{\boldsymbol{\nu}}^{(i)} - \mu_\ell \right\|_2 \leqslant O\left( k^{-25} \right) \ .$$

*Proof.* For each $\ell \in [k]$, denote by $\mathcal{T}_\ell$ the set of indices in $[n]$ satisfying

$$\left\| \nu^{(i)}(\mathbf{W}) - \mu_\ell \right\|_2 \leqslant O\left( k^{-25} \right) \ .$$

By Lemma D.16 we know that $\mathcal{T}_\ell$ has size at least $\frac{n}{k} \cdot \left( 1 - \frac{2}{k^{47}} \right)$. Further, let $\mathcal{S}$ be the set of indices selected by the algorithm. By Chernoff's bound Fact E.4 with probability $1 - e^{-n^{\Omega(1)}}$, we have $|\mathcal{S} \cap \mathcal{T}_\ell| \geqslant \frac{n^{0.01}}{k} \cdot \left( 1 - \frac{150}{k^{47}} \right)$. Taking a union bound over all $\ell \in [k]$ we get that with probability $1 - e^{-n^{\Omega(1)}}$, for each $\ell \in [k]$, there exists $\frac{n^{0.01}}{k} \cdot \left( 1 - \frac{150}{k^{47}} \right)$ indices $i \in \mathcal{S}$ such that

$$\left\| \nu^{(i)}(\mathbf{W}) - \mu_\ell \right\|_2 \leqslant O\left( k^{-25} \right) \ .$$

Now, we obtain the corollary observing (cf. Fact E.1 with $m = 1$) that with probability at least $1 - e^{-n^{\Omega(1)}}$, for all $i \in \mathcal{S}$

$$\left\| \bar{\boldsymbol{\nu}}^{(i)} - \nu^{(i)}(\mathbf{W}) \right\|_2 = \|\mathbf{w}\|_2 \leqslant n^{-0.05} \cdot \frac{\sqrt{\log(2/\delta)}}{\varepsilon} \cdot \sqrt{d} \leqslant n^{-0.04} \leqslant O\left( k^{-25} \right) \ .$$

$\qquad\square$

For each $\ell$, denote by $\mathcal{G}_\ell \subseteq \mathcal{S}$ the set of indices $i \in \mathcal{S}$ satisfying

$$\left\| \bar{\boldsymbol{\nu}}^{(i)} - \mu_\ell \right\|_2 \leqslant O\left(k^{-25}\right) .$$

Let $\mathcal{G} := \bigcup_{\ell \in [k]} \mathcal{G}_\ell$. We now have all the tools to prove utility of Algorithm D.5. We achieve this by showing thst with high probability, each bin returned by the algorithm at step 7 satisfies $\mathcal{G}_{\ell'} \subseteq \mathbf{B}_\ell$ for some $\ell, \ell' \in [k]$. Choosing the bins small enough will yield the desired result.

**Lemma D.18** (Closeness of estimates). *Consider the settings of Theorem D.3. Suppose $\mathbf{Y}$ is a good set as per Definition D.6. Let $\mathbf{W} \in \mathcal{W}(\mathbf{Y})$ be the matrix computed by Algorithm D.5. Suppose the algorithm does not reject. Then with high probability, there exists a permutation $\pi : [k] \to [k]$ such that*

$$\max_{\ell \in [k]} \left\| \mu_\ell - \hat{\boldsymbol{\mu}}_{\pi(\ell)} \right\|_2 \leqslant O\left(k^{-20}\right)$$

*Proof.* Consider distinct $\ell, \ell' \in [k]$. By Corollary D.17 for each $\bar{\boldsymbol{\nu}}^{(i)}, \bar{\boldsymbol{\nu}}^{(j)} \in \mathcal{G}_\ell$ it holds that

$$\left\| \bar{\boldsymbol{\nu}}^{(i)} - \bar{\boldsymbol{\nu}}^{(j)} \right\|_2 \leqslant C \cdot k^{-25} ,$$

for some universal constant $C > 0$. Moreover, by assumption on $\mu_\ell, \mu_{\ell'}$ for each $\bar{\boldsymbol{\nu}}^{(i)} \in \mathcal{G}_\ell$ and $\bar{\boldsymbol{\nu}}^{(j)} \in \mathcal{G}_{\ell'}$

$$\left\| \bar{\boldsymbol{\nu}}^{(i)} - \bar{\boldsymbol{\nu}}^{(j)} \right\|_2 \geqslant \Delta - O\left(k^{-25}\right) .$$

Thus, by Lemma E.5 with probability at least $1 - e^{\Omega(d^*)} \geqslant 1 - n^{-100}$ it holds that or each $\bar{\boldsymbol{\nu}}^{(i)}, \bar{\boldsymbol{\nu}}^{(j)} \in \mathcal{G}_\ell$ and $\bar{\boldsymbol{\nu}}^r \in \mathcal{G}_{\ell'}$ with $\ell' \neq \ell$,

$$\left\| \boldsymbol{\Phi}\bar{\boldsymbol{\nu}}^{(i)} - \boldsymbol{\Phi}\bar{\boldsymbol{\nu}}^{(j)} \right\|_2 \leqslant C^* \cdot k^{-25} \qquad \text{and} \qquad \left\| \boldsymbol{\Phi}\bar{\boldsymbol{\nu}}^{(i)} - \boldsymbol{\Phi}\bar{\boldsymbol{\nu}}^{(r)} \right\|_2 \geqslant \Delta - C^* \cdot k^{-25}$$

for some other universal constant $C^* > C$. Let $Q_{\boldsymbol{\Phi}}(\mathcal{G}_\ell) \subseteq \mathbb{R}^{d^*}$ be a ball of radius $C^* \cdot \left(k^{-25}\right)$ such that $\forall i \in \mathcal{G}_\ell$ it holds $\boldsymbol{\Phi}\bar{\boldsymbol{\nu}}^{(i)} \in Q_{\boldsymbol{\Phi}}(\mathcal{G}_\ell)$. That is, $Q_{\boldsymbol{\Phi}}(\mathcal{G}_\ell)$ contains the projection of all points in $\mathcal{G}_\ell$.

Recall that $d^* = 100 \log(n) \leqslant 100k^5$ and $b = k^{-15}$. Let $\mathcal{B} = \{\mathbf{B}_i\}_{i=1}^\infty$ be the sequence of bins computed by the histogram learner of Lemma A.13 for $\mathbb{R}^{d^*}$ at step 7 of the algorithm. By choice of $b$, and since $\mathbf{q}$ is chosen uniformly at random in $[0, b]$, the probability that there exists a bin $\mathbf{B} \in \mathcal{B}$ containing $Q_{\boldsymbol{\Phi}}(\mathcal{G}_\ell)$ is at least

$$1 - d^* \cdot \frac{C^*}{b} \cdot \left(k^{-25}\right) \geqslant 1 - \frac{100C^*}{b} \cdot k^{-20} \geqslant 1 - O\left(k^{-5}\right) ,$$

where we used that $d^* = 100 \log n \leqslant 100k^5$. A simple union bound over $\ell \in [k]$ yields that with high probability for all $\ell \in [k]$, there exists $\mathbf{B} \in \mathcal{B}$ such that $Q_{\boldsymbol{\Phi}}(\mathcal{G}_\ell) \subseteq \mathbf{B}$. For simplicity, denote such bin by $\mathbf{B}_\ell$.

We continue our analysis conditioning on the above events, happening with high probability. First, notice that for all $l \in [k]$

$$\max_{u, u' \in \mathbf{B}_\ell} \|u - u'\|_2^2 \leqslant d^* \cdot b^2 \leqslant 100k^{-25} \leqslant \frac{\Delta - C^* k^{-25}}{k^{10}} ,$$

and thus there cannot be $\ell, \ell' \in [k]$ such that $Q_{\boldsymbol{\Phi}}(\mathcal{G}_\ell) \subseteq \mathbf{B}_\ell$ and $Q_{\boldsymbol{\Phi}}(\mathcal{G}'_\ell) \subseteq \mathbf{B}_\ell$. Moreover, by Corollary D.17 and

$$\min_{\ell \in [k]} |\mathcal{G}_\ell| \geqslant \frac{n^{0.01}}{k} \cdot \left(1 - \frac{150}{k^{47}}\right) ,$$

and hence

$$|\mathcal{S} \setminus \mathcal{G}| \leqslant n^{0.01} \cdot \frac{150}{k^{47}} = \frac{n^{0.01}}{k} \cdot \frac{150}{k^{46}}$$

it must be that step 7 returned bins $\mathbf{B}_1, \ldots, \mathbf{B}_k$. This also implies that the algorithm does not reject. Further, by Lemma E.5 for all $\bar{\boldsymbol{\nu}}^{(i)}, \bar{\boldsymbol{\nu}}^{(j)}$ such that $\boldsymbol{\Phi}\bar{\boldsymbol{\nu}}^{(i)}, \boldsymbol{\Phi}\bar{\boldsymbol{\nu}}^{(j)} \in \mathbf{B}_l$ it holds that

$$\left\| \bar{\boldsymbol{\nu}}^{(i)} - \bar{\boldsymbol{\nu}}^{(j)} \right\|_2 \leqslant C^* \cdot \left\| \boldsymbol{\Phi}\bar{\boldsymbol{\nu}}^{(i)} - \boldsymbol{\Phi}\bar{\boldsymbol{\nu}}^{(j)} \right\|_2 \leqslant C^* \cdot \sqrt{d^*} \cdot b \leqslant O\left(k^{-12}\right) .$$

And hence, by triangle inequality, we get

$$\left\|\bar{\boldsymbol{\nu}}^{(i)} - \mu_l\right\|_2 \leqslant O\left(k^{-12}\right) .$$

Finally, recall that for each $\ell \in [k]$,

$$\hat{\boldsymbol{\mu}}_l := \frac{1}{|\{j \mid \boldsymbol{\Phi}\bar{\boldsymbol{\nu}}^{(j)} \in \mathbf{B}_i\}|} \cdot \left(\sum_{\boldsymbol{\Phi}\bar{\boldsymbol{\nu}}^{(j)} \in \mathbf{B}_l} \bar{\boldsymbol{\nu}}^{(j)}\right) + \mathbf{w}' ,$$

where $\mathbf{w}' \sim N\left(0, N\left(0, 32 \cdot k^{-120} \cdot \frac{\log(2kn/\delta)}{\varepsilon^2} \cdot \mathrm{Id}\right)\right)$. Since by choice of $n, k, \varepsilon$ it holds that

$$32 \cdot k^{-120} \cdot \frac{\log(2kn/\delta)}{\varepsilon^2} \leqslant O\left(k^{-90}\right) ,$$

we get with probability at least $1 - e^{-k^{\Omega(1)}}$ for each $\ell \in [k]$, by Fact E.1, with $m = 1$, and a union bound that

$$\|\mathbf{w}'\| \leqslant O\left(k^{-20}\right) .$$

Since all $\bar{\boldsymbol{\nu}}^{(i)}$ such that $\boldsymbol{\Phi}\bar{\boldsymbol{\nu}}^{(i)} \in \mathbf{B}_l$ are at most $O\left(k^{-12}\right)$-far from $\mu_l$, also their average is. We conclude that

$$\|\hat{\boldsymbol{\mu}}_\ell - \mu_l\|_2 \leqslant O(k^{-12}) + \|\mathbf{w}\|_2 \leqslant O(k^{-12}) .$$

This completes the proof. $\qquad\square$

Now Theorem D.3 is a trivial consequence.

*Proof of Theorem D.3.* The error guarantees and privacy guarantees immediately follows combining Lemma D.8, Lemma D.15, Lemma D.14 and Lemma D.18. The running time follows by Fact A.16.
$\qquad\square$

# E   Concentration inequalities

We introduce here several useful and standard concentration inequalities.

**Fact E.1** (Concentration of spectral norm of Gaussian matrices). *Let $\mathbf{W} \sim \mathcal{N}(0,1)^{m \times n}$. Then for any $t$, we have*

$$\mathbb{P}\left(\sqrt{m} - \sqrt{n} - t \leqslant \sigma_{\min}(\mathbf{W}) \leqslant \sigma_{\max}(\mathbf{W}) \leqslant \sqrt{m} + \sqrt{n} + t\right) \geqslant 1 - 2\exp\left(-\frac{t^2}{2}\right),$$

*where $\sigma_{\min}(\cdot)$ and $\sigma_{\max}(\cdot)$ denote the minimum and the maximum singular values of a matrix, respectively.*

*Let $\mathbf{W}'$ be an $n$-by-$n$ symmetric matrix with independent entries sampled from $N(0, \sigma^2)$. Then $\|\mathbf{W}'\| \leqslant 3\sigma\sqrt{n}$ with probability at least $1 - \exp(-\Omega(n))$.*

**Fact E.2** (Maximum degree of Erdős-Rényi graphs). *Let $G$ be an Erdős-Rényi graph on $n$ vertices with edge probability $p$. Then with probability at least $1 - n\exp(-np/3)$, any vertex in $G$ has degree at most $2np$.*

**Fact E.3** (Gaussian concentration bounds). *Let $\mathbf{X} \sim \mathcal{N}(0, \sigma^2)$. Then for any $t \geqslant 0$,*

$$\max\left\{\mathbb{P}\left(\mathbf{X} \geqslant t\right), \mathbb{P}\left(\mathbf{X} \leqslant -t\right)\right\} \leqslant \exp\left(-\frac{t^2}{2\sigma^2}\right).$$

**Fact E.4** (Chernoff bound). *Let $\mathbf{X_1}, \ldots, \mathbf{X_n}$ be independent random variables taking values in $\{0, 1\}$. Let $\mathbf{X} := \sum_{i=1}^n \mathbf{X_i}$ and let $\mu := \mathbb{E}\,\mathbf{X}$. Then for any $\delta > 0$,*

$$\mathbb{P}\left(\mathbf{X} \leqslant (1-\delta)\mu\right) \leqslant \exp\left(-\frac{\delta^2\mu}{2}\right),$$

$$\mathbb{P}\left(\mathbf{X} \geqslant (1+\delta)\mu\right) \leqslant \exp\left(-\frac{\delta^2\mu}{2+\delta}\right).$$

**Lemma E.5** ([31]). *Let $\Phi$ be a $d$-by-$n$ Gaussian matrix, with each entry independently chosen from $N(0, 1/d)$. Then, for every vector $u \in \mathbb{R}^n$ and every $\alpha \in (0, 1)$*

$$\mathbb{P}\left(\|\Phi u\| = (1 \pm \alpha)\|u\|\right) \geqslant 1 - e^{-\Omega(\alpha^2 d)} .$$

## F    Linear algebra

**Lemma F.1** (Weyl's inequality). *Let $A$ and $B$ be symmetric matrices. Let $R = A - B$. Let $\alpha_1 \geqslant \cdots \geqslant \alpha_n$ be the eigenvalues of $A$. Let $\beta_1 \geqslant \cdots \geqslant \beta_n$ be the eigenvalues of $B$. Then for each $i \in [n]$,*

$$|\alpha_i - \beta_i| \leqslant \|R\|.$$

**Lemma F.2** (Davis-Kahan's theorem). *Let $A$ and $B$ be symmetric matrices. Let $R = A - B$. Let $\alpha_1 \geqslant \cdots \geqslant \alpha_n$ be the eigenvalues of $A$ with corresponding eigenvectors $v_1, \ldots, v_n$. Let $\beta_1 \geqslant \cdots \geqslant \beta_n$ be the eigenvalues of $B$ with corresponding eigenvectors $u_1, \ldots, u_n$. Let $\theta_i$ be the angle between $\pm v_i$ and $\pm u_i$. Then for each $i \in [n]$,*

$$\sin(2\theta_i) \leqslant \frac{2\|R\|}{\min_{j \neq i} |\alpha_i - \alpha_j|}.$$

## G    Convex optimization

**Proposition G.1.** *Let $f : \mathbb{R}^m \to \mathbb{R}$ be a convex function. Let $\mathcal{K} \subseteq \mathbb{R}^m$ be a convex set. Then $y^* \in \mathcal{K}$ is a minimizer of $f$ over $\mathcal{K}$ if and only if there exists a subgradient $g \in \partial f(y^*)$ such that*

$$\langle y - y^*, g \rangle \geqslant 0 \quad \forall y \in \mathcal{K}.$$

*Proof.* Define indicator function

$$I_{\mathcal{K}}(y) = \begin{cases} 0, & y \in \mathcal{K}, \\ \infty, & y \notin \mathcal{K}. \end{cases}$$

Then for $y \in \mathcal{K}$, one has

$$\partial I_{\mathcal{K}}(y) = \{g \in \mathbb{R}^m : \langle g, y - y' \rangle \geqslant 0 \ \forall y' \in \mathcal{K}\}.$$

Note $y^*$ is a minimizer of $f$ over $\mathcal{K}$, if and only if $y^*$ is a minimizer of $f + I_{\mathcal{K}}$ over $\mathbb{R}^m$, if and only if $\mathbf{0}_m \in \partial(f + I_{\mathcal{K}})(y^*) = \partial f(y^*) + \partial I_{\mathcal{K}}(y^*)$, if and only if there exists $g \in \partial f(y^*)$ such that $\langle g, y - y^* \rangle \geqslant 0$ for any $y \in \mathcal{K}$. $\square$

**Proposition G.2** (Pythagorean theorem from strong convexity). *Let $f : \mathbb{R}^m \to \mathbb{R}$ be a convex function. Let $\mathcal{K} \subseteq \mathbb{R}^m$ be a convex set. Suppose $f$ is $\kappa$-strongly convex over $\mathcal{K}$. Let $x^* \in \mathcal{K}$ be a minimizer of $f$ over $\mathcal{K}$. Then for any $x \in \mathcal{K}$, one has*

$$\|x - x^*\|^2 \leqslant \frac{2}{\kappa}(f(x) - f(x^*)).$$

*Proof.* By strong convexity, for any subgradient $g \in \partial f(x^*)$ one has

$$f(x) \geqslant f(x^*) + \langle x - x^*, g \rangle + \frac{\kappa}{2}\|x - x^*\|^2.$$

By Proposition G.1, $\langle x - x^*, g \rangle \geqslant 0$ for some $g \in \partial f(x^*)$. Then the result follows. $\square$

## H    Deferred proofs SBM

We prove Lemma C.9 restated below.

**Lemma H.1** (Restatement of Lemma C.9). *Consider the settings of Lemma C.8. With probability $1 - \exp(-\Omega(n))$ over $\mathbf{G} \sim \mathsf{SBM}_n(\gamma, d, x)$,*

$$\left\|\hat{X}(Y(\mathbf{G})) - \frac{1}{n}xx^\top\right\|_F^2 \leqslant \frac{800}{\gamma\sqrt{d}}.$$

*Proof.* Recall $\mathcal{K} = \{X \in \mathbb{R}^{n \times n} : X \succeq 0, X_{ii} = 1/n \ \forall i\}$. Let $X^* := \frac{1}{n}xx^\top$. Since $\hat{X} = \hat{X}(Y(\mathbf{G}))$ is a minimizer of $\min_{X \in \mathcal{K}} \|Y(\mathbf{G}) - X\|_F^2$ and $X^* \in \mathcal{K}$, we have

$$\left\|\hat{X} - Y(\mathbf{G})\right\|_F^2 \leqslant \|X^* - Y(\mathbf{G})\|_F^2 \iff \left\|\hat{X} - X^*\right\|_F^2 \leqslant 2\left\langle \hat{X} - X^*, Y(\mathbf{G}) - X^*\right\rangle.$$

The infinity-to-one norm of a matrix $M \in \mathbb{R}^{m \times n}$ is defined as

$$\|M\|_{\infty \to 1} := \max \left\{ \langle u, Mv \rangle : u \in \{\pm 1\}^m, v \in \{\pm 1\}^n \right\}.$$

By [28, Fact 3.2], every $Z \in \mathcal{K}$ satisfies

$$|\langle Z, Y(\mathbf{G}) - X^* \rangle| \leqslant \frac{K_G}{n} \cdot \|Y(\mathbf{G}) - X^*\|_{\infty \to 1},$$

where $K_G \leqslant 1.783$ is Grothendieck's constant. Similar to the proof of [28, Lemma 4.1], using Bernstein's inequality and union bound, we can show (cf. Fact H.2)

$$\|Y(\mathbf{G}) - X^*\|_{\infty \to 1} \leqslant \frac{100n}{\gamma \sqrt{d}}$$

with probability $1 - \exp(-\Omega(n))$. Putting things together, we have

$$\left\| \hat{X}(Y(\mathbf{G})) - \frac{1}{n} xx^\top \right\|_F^2 \leqslant \frac{400 \cdot K_G}{\gamma \sqrt{d}},$$

with probability $1 - \exp(-\Omega(n))$. $\hfill \square$

**Fact H.2.** *Let* $\gamma > 0, d \in \mathbb{N}, x^* \in \{\pm 1\}^n$, *and* $\mathbf{G} \sim \mathsf{SBM}(\gamma, d, x^*)$. *Let* $Y(\mathbf{G}) = \frac{1}{\gamma d} \left( A(\mathbf{G}) - \frac{d}{n} J \right)$, *where* $A((G))$ *is the adjacency matrix of* $(G)$ *with entries* $d/n$ *on the diagonal. Then*

$$\max_{x \in \{\pm 1\}^n} \left| x^\top \left( Y(\mathbf{G}) - \tfrac{1}{n} x^* (x^*)^\top \right) x \right| \leqslant \frac{100n}{\gamma \sqrt{d}}$$

*with probability at least* $1 - e^{-10n}$.

*Proof.* The result will follow using Bernstein's Inequality and a union bound. Define $\mathbf{E} := Y(\mathbf{G}) - \frac{1}{n} x^* (x^*)^\top$. Fix $x \in \{\pm 1\}^n$ and for $1 \leqslant i < j \leqslant n$, let $\mathbf{Z}_{i,j} := \mathbf{E}_{i,j} x_i x_j$. Then $x^\top \mathbf{E} x = 2 \sum_{1 \leqslant i < j \leqslant n} \mathbf{Z}_{i,j}$. Note that

$$\mathbb{E} \, \mathbf{Z}_{i,j} = 0,$$

$$|\mathbf{Z}_{i,j}| \leqslant \frac{1}{\gamma n} \cdot \left( \frac{n}{d} - 1 \right) + \frac{1}{\gamma dn} \leqslant \frac{1}{\gamma d},$$

$$\mathbb{E} \, \mathbf{Z}_{i,j}^2 = \mathrm{Var} \left[ \mathbf{Y}(\mathbf{G})_{i,j} \right] \leqslant \mathbb{E} \, \mathbf{Y}(\mathbf{G})_{i,j}^2 \leqslant (1 + \gamma) \frac{d}{n} \cdot \frac{1}{\gamma^2 n^2} \left[ \left( \frac{n}{d} - 1 \right)^2 - \frac{1}{\gamma^2 n^2} \right] + \frac{1}{\gamma^2 n^2}$$

$$\leqslant (1 + \gamma) \frac{1}{d \gamma^2 n} + \frac{1}{\gamma^2 n^2} \leqslant \frac{3}{\gamma^2 dn}.$$

By Bernstein's Inequality (cf. [62, Proposition 2.14]) it follows that

$$\mathbb{P} \left( \sum_{i < j} \mathbf{Z}_{i,j} \geqslant \frac{50n}{\gamma \sqrt{d}} \right) \leqslant \mathbb{P} \left( \sum_{i < j} \mathbf{Z}_{i,j} \geqslant \frac{n^2}{2} \cdot \frac{100n}{\gamma \sqrt{d}} \right) \leqslant 2 \exp \left( - \frac{\frac{10^4}{\gamma^2 d}}{\frac{3}{\gamma^2 dn} + \frac{100}{3 \gamma^2 d^{3/2} n}} \right)$$

$$= 2 \exp \left( - \frac{10^4 n}{3 + \frac{100}{\sqrt{d}}} \right) \leqslant \exp \left( -50n \right).$$

Hence, by a union bound over all $x \in \{\pm 1\}^n$ it follows that

$$\max_{x \in \{\pm 1\}^n} \left| x^\top \left( Y(\mathbf{G}) - \tfrac{1}{n} x^* (x^*)^\top \right) x \right| \leqslant \frac{100n}{\gamma \sqrt{d}}$$

with probability at least $1 - e^{-10n}$. $\hfill \square$

# I Deferred proofs for clustering

In this section, we will prove Lemma D.15 restated below.

**Lemma** (Restatement of Lemma D.15). *Consider the settings of Theorem D.3. Suppose $\mathbf{Y}$ is a good set as per Definition D.6. Let $W(\mathbf{Y}) \in \mathcal{W}(\mathbf{Y})$ be the matrix computed by Algorithm D.5. Suppose the algorithm does not reject. Then*

$$\|\phi(W(\mathbf{Y})) - \mathbf{W}^*\|_1 \leqslant \frac{n^2}{k} \cdot \frac{3}{k^{98}} .$$

We will need the following fact about our clustering program. Similar facts where used, e.g., in [30, 26]. One difference for us is that we don't have a constraint on the lower bound on the cluster size indicated by our SOS variables. However, since we maximize a variant of the $\ell_1$ norm of the second moment matrix of the pseudo-distribution this will make up for this.

**Fact I.1.** *Consider the same setting as in Lemma D.15. Let $0 < \delta \leqslant \frac{1}{1.5 \cdot 10^{10}} \cdot \frac{1}{k^{201}}$ and denote by $\mathbf{C}_1, \ldots, \mathbf{C}_k \subseteq [n]$ the indices belonging to each true cluster. Then $W(\mathbf{Y})$ satisfies the following three properties:*

1. *For all $i, j \in [n]$ it holds that $0 \leqslant \mathbf{W}_{i,j} \leqslant 1$,*

2. *for all $i \in [n]$ it holds that $\sum_{j=1}^n \mathbf{W}_{i,j} \leqslant \frac{n}{k}$ and for at least $(1 - \frac{1}{1000k^{100}})n$ indices $i \in [n]$ it holds that $\sum_{j=1}^n \mathbf{W}_{i,j} \geqslant (1 - \frac{1}{(10)^6 k^{200}}) \cdot \frac{n}{k}$,*

3. *for all $r \in [k]$ it holds that $\sum_{i \in \mathbf{C}_r, j \notin \mathbf{C}_r} \mathbf{W}_{i,j} \leqslant \delta \cdot \frac{n^2}{k}$.*

We will prove Fact I.1 at the end of this section. With this in hand, we can proof Lemma D.15.

*Proof of Lemma D.15.* For brevity, we write $\mathbf{W} = W(\mathbf{Y})$. Since $\phi(\mathbf{W}^*) = \mathbf{W}^*$ and $\phi$ is 10-Lipschitz we can also bound

$$\|\phi(\mathbf{W}) - \mathbf{W}^*\|_1 \leqslant 10 \cdot \|\mathbf{W} - \mathbf{W}^*\|_1 .$$

Let $\delta \leqslant \frac{1}{1.5 \cdot 10^{10}} \cdot \frac{1}{k^{201}}$ and again let $\mathbf{C}_1, \ldots, \mathbf{C}_k \subseteq [n]$ denote the indices belonging to each true cluster. Note that by assumption that $\mathbf{Y}$ is a good sample it holds for each $r \in [k]$ that $\frac{n}{k} - n^{0.6} \leqslant |\mathbf{C}_r| \leqslant \frac{n}{k} + n^{0.6}$.

Let $r, r' \in [k]$. We can write

$$\|\mathbf{W} - \mathbf{W}^*\|_1 = \sum_{r=1}^k \sum_{i,j \in \mathbf{C}_r} |\mathbf{W}_{i,j} - 1| + \sum_{r=1}^k \sum_{i \in \mathbf{C}_r, j \notin \mathbf{C}_r} |\mathbf{W}_{i,j} - 0| \tag{I.1}$$

Note that we can bound the second sum by $k \cdot \delta \frac{n^2}{k}$ using Item 3. Further, in what follows consider only indices $i$ such that $\sum_{j=1}^n \mathbf{W}_{i,j} \geqslant (1 - \frac{1}{(10)^6 k^{200}}) \cdot \frac{n}{k}$. By Item 2 we can bound the contribution of the other indices by

$$\frac{1}{1000k^{100}} n \cdot \left(\frac{n}{k} + n^{0.6}\right) \leqslant \frac{2}{1000k^{100}} \cdot \frac{n^2}{k} .$$

Focusing only on such indices, for the first sum in Eq. (I.1), fix $r \in [k]$. We will aim to show that most entries of $\mathbf{W}$ are large if and only if the corresponding entry of $\mathbf{W}^*$ is 1. By Item 3 and Markov's Inequality, it follows that for at least a $(1 - \frac{1}{1000k^{100}})$-fraction of the indices $i \in \mathbf{C}_r$ it holds that

$$\sum_{j \notin \mathbf{C}_r} \mathbf{W}_{i,j} \leqslant 1000k^{100} \cdot \delta \frac{n^2}{k \cdot |\mathbf{C}_r|} \leqslant 1000k^{100}\delta \cdot \frac{n}{1 - k \cdot n^{-0.4}} \leqslant 2000k^{101}\delta \cdot \frac{n}{k} ,$$

where we used that $|\mathbf{C}_r| \geqslant \frac{n}{k} - n^{0.6}$. Call such indices *good*. Notice that for good indices it follows using Item 2 that

$$\sum_{j \in \mathbf{C}_r} \mathbf{W}_{i,j} \geqslant \frac{n}{k} \cdot (1 - \frac{1}{(10)^6 k^{200}} - 2000k^{101}\delta) .$$

Denote by $G$ the number of $j \in \mathbf{C}_r$ such that $\mathbf{W}_{i,j} \geqslant 1 - \frac{1}{1000k^{100}}$. Using the previous display and that $\mathbf{W}_{i,j} \leqslant 1$ we obtain

$$\frac{n}{k} \cdot \left(1 - \frac{1}{(10)^6 k^{200}} - 2000k^{101}\delta\right) \leqslant \sum_{j \in \mathbf{C}_r} \mathbf{W}_{i,j} \leqslant G \cdot 1 + (|\mathbf{C}_r| - G) \cdot (1 - \tfrac{1}{1000k^{100}})$$

$$\leqslant G \cdot \tfrac{1}{1000k^{100}} + \tfrac{n}{k} \cdot (1 + \tfrac{1}{kn^{0.4}}) \cdot (1 - \tfrac{1}{1000k^{100}})$$

$$\leqslant G \cdot \tfrac{1}{1000k^{100}} + \tfrac{n}{k} \cdot (1 + \tfrac{1}{kn^{0.4}}),$$

where we also used $|\mathbf{C}_r| \leqslant \frac{n}{k} + n^{0.6}$. Rearranging now yields

$$G \geqslant \frac{n}{k} \cdot \left(1 - \frac{1}{1000k^{100}} - \frac{10^3 k^{99}}{n^{0.4}} - 2 \cdot 10^6 k^{101}\delta\right) \geqslant \frac{n}{k} \cdot \left(1 - \frac{2}{1000k^{100}} - 2 \cdot 10^6 k^{101}\delta\right).$$

We can now bound

$$\sum_{i,j \in \mathbf{C}_r} |\mathbf{W}_{i,j} - 1| = \sum_{i,j \in \mathbf{C}_r, i \text{ is good}} |\mathbf{W}_{i,j} - 1| + \sum_{i,j \in \mathbf{C}_r, i \text{ is not good}} |\mathbf{W}_{i,j} - 1|$$

$$\leqslant |\mathbf{C}_r| \cdot \left((|\mathbf{C}_r| - G) \cdot 1 + |\mathbf{C}_r| \cdot \tfrac{1}{1000k^{100}}\right) + \tfrac{1}{1000k^{100}} \cdot |\mathbf{C}_r|^2$$

$$\leqslant |\mathbf{C}_r|^2 \left(1 + \tfrac{1}{500k^{100}}\right) - G \cdot |\mathbf{C}_r|$$

$$\leqslant \tfrac{n^2}{k^2}(1 + \tfrac{k}{n^{0.4}})^2(1 + \tfrac{1}{500k^{100}}) - \tfrac{n^2}{k^2}(1 - \tfrac{2}{1000k^{100}} - 2 \cdot 10^6 k^{101}\delta)(1 - \tfrac{k}{n^{0.4}})$$

$$\leqslant \tfrac{n^2}{k^2} \cdot (30 \cdot 10^6 k^{101}\delta + \tfrac{11}{500k^{100}}) \leqslant \tfrac{n^2}{k} \cdot (30 \cdot 10^6 k^{100}\delta + \tfrac{11}{500k^{101}})$$

$$\leqslant \tfrac{n^2}{k} \cdot \tfrac{3}{125k^{101}}.$$

Putting everything together, it follows that

$$\|\phi(\mathbf{W}) - \mathbf{W}^*\|_F^2 \leqslant \|\phi(\mathbf{W}) - \mathbf{W}^*\|_1 \leqslant 10 \cdot \frac{n^2}{k} \left(\delta k + \frac{2}{1000k^{100}} + \frac{3}{125k^{100}}\right) \leqslant \frac{n^2}{k} \cdot \frac{4}{k^{100}} \leqslant \frac{n^2}{k} \cdot \frac{3}{k^{98}}.$$

$$\square$$

It remains to verify Fact I.1.

*Proof of Fact I.1.* Let $\mathcal{P} = \mathcal{P}_{n,k,t}(\mathbf{Y})$ be the system of Eq. $(\mathcal{P}_{n,k,t}(Y))$. Recall that $\mathbf{W}_{i,j} = \tilde{\mathbb{E}} \sum_{l \in [k]} z_{i,l} z_{j,l}$. Since

$$\mathcal{P} \left|\frac{}{4}\right. \left\{0 \leqslant \sum_{l \in [k]} z_{i,l} z_{j,l} \leqslant \sum_{l \in [k]} z_{i,l} \leqslant 1\right\},$$

it follows that $0 \leqslant \mathbf{W}_{i,j} \leqslant 1$. Further, for each $i \in [n]$ it holds that

$$\mathcal{P} \left|\frac{}{4}\right. \left\{\sum_{j \in [n], l \in [k]} z_{j,l} z_{i,l} \leqslant \frac{n}{k} \sum_{l \in [k]} z_{i,l} \leqslant \frac{n}{k}\right\}$$

implying that $\sum_{j \in [n]} \mathbf{W}_{i,j} \leqslant \frac{n}{k}$. Further, by Lemma D.14

$$\|\mathbf{W}\|_1 \geqslant \frac{n^2}{k} \cdot \left(1 - n^{-0.4} - \frac{1}{(10)^{10} k^{300}}\right) \geqslant \frac{n^2}{k} \cdot \left(1 - \frac{1}{(10)^9 k^{300}}\right).$$

Denote by $\mathbf{W}_i$ the $i$-th row of $\mathbf{W}$ and by $L$ the number of rows which have $\ell_1$ norm at least $(1 - \frac{1}{(10)^6 k^{200}}) \cdot \frac{n}{k}$. Since for all $i$ it holds that $\|\mathbf{W}_i\|_1 \leqslant \frac{n}{k}$ it follows that

$$\frac{n^2}{k} \cdot \left(1 - \frac{1}{(10)^9 k^{300}}\right) \leqslant \sum_{i \in [n]} \|\mathbf{W}_i\|_1 \leqslant L \cdot \frac{n}{k} + (n - L) \cdot \left(1 - \frac{1}{(10)^6 k^{200}}\right) \cdot \frac{n}{k}$$

$$= L \cdot \frac{1}{(10)^6 k^{200}} \cdot \frac{n}{k} + \frac{n^2}{k} \cdot \left(1 - \frac{1}{(10)^6 k^{200}}\right)$$

Rearranging then yields $L \geqslant (1 - \frac{1}{1000k^{100}}) \cdot n$ which proofs Item 2.

It remains to verify Item 3. Fix $r, l \in [k]$ and define $z_l(\mathbf{C}_r) = \frac{k}{n} \sum_{i \in \mathbf{C}_r} z_{i,l}$. Let $t > 0$ be an integer. We aim to show that for all unit vectors $v$ it holds that

$$\mathcal{P} \left|\frac{}{10t} \left\{ z_l(\mathbf{C}_r) \cdot \frac{1}{\Delta^{2t}} \sum_{r' \neq r} z_l(\mathbf{C}_{r'}) \langle \mu_r - \mu_{r'}, v \rangle^{2t} \leqslant \frac{\delta}{k} \right\}, \right. \tag{I.2}$$

where $\Delta$ is the minimal separation between the true means. Before proving this, let us examine how we can use this fact to prove Item 3. Note, that for all $r \neq r'$ it holds that

$$\sum_{s,u \in [k]} \left\langle \mu_r - \mu_{r'}, \frac{\mu_s - \mu_u}{\|\mu_s - \mu_u\|} \right\rangle^{2t} \geqslant \Delta^{2t}.$$

Hence, if the above SOS proof indeed exists, we obtain

$$\sum_{i \in \mathbf{C}_r, j \notin \mathbf{C}_r} \mathbf{W}_{i,j} = \sum_{l=1}^{k} \tilde{\mathbb{E}} \sum_{i \in \mathbf{C}_r, j \notin \mathbf{C}_r} z_{i,l} z_{j,l} = \frac{n^2}{k^2} \tilde{\mathbb{E}} z_l(\mathbf{C}_r) \cdot \sum_{r' \neq r} z_l(\mathbf{C}_{r'})$$

$$\leqslant \frac{n^2}{\Delta^{2t} k^2} \sum_{s,u \in [k]} \tilde{\mathbb{E}} z_l(\mathbf{C}_r) \cdot \sum_{r' \neq r} z_l(\mathbf{C}_r) \left\langle \mu_r - \mu_{r'}, \frac{\mu_s - \mu_u}{\|\mu_s - \mu_u\|} \right\rangle^{2t}$$

$$\leqslant \frac{\delta}{k} k^2 \cdot \frac{n^2}{k^2} = \delta \cdot \frac{n^2}{k}.$$

In the remainder of this proof we will prove Eq. (I.2). We will use the following SOS version of the triangle Inequality (cf. Fact I.2)

$$\left|\frac{x,y}{2t} (x + y)^{2t} \leqslant 2^{2t-1}(x^{2t} + y^{2t}). \right.$$

Recall that $\mu'_l = \frac{k}{n} \sum_{i=1}^{n} z_{i,l} y_i$ and denote by $\mu_{\pi(i)}$ the true mean corresponding to the $i$-th sample. Let $v$ be an arbitrary unit vector, it follows that

$$\mathcal{P} \left|\frac{}{10t} \left\{ z_l(\mathbf{C}_r) \cdot \frac{1}{\Delta^{2t}} \sum_{r' \neq r} z_l(\mathbf{C}_{r'}) \langle \mu_r - \mu_{r'}, v \rangle^{2t} \right.\right.$$

$$\leqslant z_l(\mathbf{C}_r) \cdot \frac{2^{2t-1}}{\Delta^{2t}} \sum_{r' \neq r} z_l(\mathbf{C}_{r'}) \left( \langle \mu_r - \mu'_l, v \rangle^{2t} + \langle \mu_{r'} - \mu'_l, v \rangle^{2t} \right)$$

$$\leqslant \frac{2^{2t-1}}{\Delta^{2t}} \sum_{r=1}^{k} z_l(\mathbf{C}_r) \langle \mu_r - \mu'_l, v \rangle^{2t} = \frac{2^{2t-1}}{\Delta^{2t}} \cdot \frac{k}{n} \sum_{i=1}^{n} z_{i,l} \langle \mu_{\pi(i)} - \mu'_l, v \rangle^{2t} \right\},$$

where we used that $\mathcal{P} \left|\frac{}{1} \sum_{r=1}^{k} z_l(\mathbf{C}_r) \leqslant 1\right.$. Using the SOS triangle inequality again and that $\mathcal{P} \left|\frac{}{2} z_{i,l} \leqslant 1\right.$ we obtain

$$\mathcal{P} \left|\frac{10t}{} \left\{ z_l(\mathbf{C}_r) \cdot \frac{1}{\Delta^{2t}} \sum_{r' \neq r} z_l(\mathbf{C}_{r'}) \langle \mu_r - \mu_{r'}, v \rangle^{2t} \right.\right.$$

$$\leqslant \frac{2^{4t-1}}{\Delta^{2t}} \cdot \left( k \cdot \frac{1}{n} \sum_{i=1}^{n} \langle \mathbf{y}_i - \mu_{\pi(i)}, v \rangle^{2t} + \frac{k}{n} \sum_{i=1}^{n} z_{i,l} \langle \mathbf{y}_i - \mu'_l, v \rangle^{2t} \right) \right\}.$$

We start by bounding the first sum. Recall that by assumption the uniform distribution over each true cluster is $2t$-explicitly $2$-bounded. It follows that

$$\left|\frac{}{2t} \left\{ \frac{1}{n} \sum_{i=1}^{n} \langle \mathbf{y}_i - \mu_{\pi(i)}, v \rangle^{2t} = \frac{1}{k} \sum_{r=1}^{k} \frac{k}{n} \sum_{i \in \mathbf{C}_r} \langle \mathbf{y}_i - \mu_r, v \rangle^{2t} \leqslant \frac{1}{k} \sum_{r=1}^{k} \frac{k}{n} \cdot |\mathbf{C}_r| \cdot (2t)^t \cdot \|v\|_2^{2t} \right. \tag{I.3}$$

$$\leqslant \left( 1 + \frac{k}{n^{0.4}} \right) \cdot (2t)^t \leqslant 2(2t)^t \right\}, \tag{I.4}$$

where we used that $|\mathbf{C}_r| \leqslant \frac{n}{k} + n^{0.6}$. To bound the second sum, we will use the moment bound constraints. In particular, we know that

$$\mathcal{P} \left|\frac{}{10t} \left\{ \frac{k}{n} \sum_{i=1}^{n} z_{i,l} \langle \mathbf{y}_i - \mu_l', v \rangle^{2t} \leqslant (2t)^t \right\} \right. . \tag{I.5}$$

Combining Eq. (I.4) and Eq. (I.5) now yields

$$\mathcal{P} \left|\frac{}{10t} \left\{ z_l(\mathbf{C}_r) \cdot \frac{1}{\Delta^{2t}} \sum_{r' \neq r} z_l(\mathbf{C}_{r'}) \langle \mu_r - \mu_{r'}, v \rangle^{2t} \leqslant k \frac{2^{2t+1}(2t)^t}{\Delta^{2t}} \leqslant k \left( \frac{8t}{\Delta^2} \right)^t \right\} \right. .$$

Note that by assumption $\Delta \geqslant O(\sqrt{t k^{1/t}})$. Overloading notation, we can choose the $t$ parameter in the SOS proof to be 202 times the $t$ parameter in the lower bound in the separation to obtain[35]

$$\sum_{i \in \mathbf{C}_r, j \notin \mathbf{C}_r} \mathbf{W}_{i,j} \leqslant \delta \cdot \frac{n^2}{k} .$$

$\square$

## I.1 Small Lemmas

**Fact I.2** (Lemma A.2 in [36]). *For all integers $t > 0$ it holds that*

$$\left|\frac{x,y}{2t} (x + y)^{2t} \leqslant 2^{2t-1}(x^{2t} + y^{2t}) \right. .$$

**Fact I.3.** *Let $\varepsilon, \delta > 0$. Let $\mathcal{M} \colon \mathcal{Y} \to \mathcal{O}$ be a randomized algorithm that, for every pair of adjacent inputs, with probability at least $1 - \gamma \geqslant 1/2$ over the internal randomness of $\mathcal{Y}$[36] satisfies $(\varepsilon, \delta)$-privacy. Then $\mathcal{M}$ is $(\varepsilon + 2\gamma, \delta + \gamma)$-private.*

*Proof.* Let $X, X'$ be adjacent input and let $B$ be the event under which $\mathcal{M}$ is $(\varepsilon, \delta)$-private. By assumption, we know that $\mathbb{P}(B) \geqslant 1 - \gamma$. Let $S \in \mathcal{O}$, it follows that

$$\begin{aligned}
\mathbb{P}(\mathcal{M}(X) \in S) &= \mathbb{P}(B) \cdot \mathbb{P}(\mathcal{M}(X) \in S \mid B) + \mathbb{P}(B^c) \cdot \mathbb{P}(\mathcal{M}(X) \in S \mid B^c) \\
&\leqslant \mathbb{P}(\mathcal{M}(X) \in S \mid B) + \gamma \\
&\leqslant e^\varepsilon \mathbb{P}(\mathcal{M}(X) \in S \mid B) + \delta + \gamma \\
&\leqslant \frac{e^\varepsilon}{\mathbb{P}(B)} \cdot \mathbb{P}(\mathcal{M}(X) \in S) + \delta + \gamma \\
&\leqslant e^{\varepsilon + \log\left(\frac{1}{1-\gamma}\right)} \cdot \mathbb{P}(\mathcal{M}(X) \in S) + (\delta + \gamma) \\
&\leqslant e^{\varepsilon + 2\gamma} \cdot \mathbb{P}(\mathcal{M}(X) \in S) + (\delta + \gamma) ,
\end{aligned}$$

where we used that $\log(1 - \gamma) \geqslant -2\gamma$ for $\gamma \in [0, 1/2]$. $\square$

## I.2 Privatizing input using the Gaussian Mechanism

In this section, we will proof the following helpful lemma used in the privacy analysis of our clustering algorithm (Algorithm D.5). In summary, it says that when restricted to some set our input has small $\ell_2$ sensitivity, we can first add Gaussian noise proportional to this sensitivity and afterwards treat this part of the input as "privatized". In particular, for the remainder of the privacy analysis we can treat this part as the same on adjacent inputs. Note that we phrase the lemma in terms of matrix inputs since this is what we use in our application. Of course, it also holds for more general inputs.

---

[35]Note that this influences the exponent in the running time and sample complexity only by a constant factor and hence doesn't violate the assumptions of Theorem D.3.

[36]In particular, this randomness is independent of the input

**Lemma I.4.** *Let $V, V' \in \mathbb{R}^{n \times d}, m \in [n]$ and $\Delta > 0$ be such that there exists a set $S$ of size at least $n - m$ satisfying*

$$\forall i \in S. \; \|V_i - V_i'\|_2^2 \leqslant \Delta^2 \,,$$

*where $V_i, V_i'$ denote the rows of $V, V'$, respectively. Let $\mathcal{A}_2 \colon \mathbb{R}^{n \times d} \to \mathcal{O}$ be an algorithm that is $(\varepsilon_2, \delta_2)$-differentially private in the standard sense, i.e., for all sets $\mathcal{S} \subseteq \mathcal{O}$ and datasets $X, X' \in \colon \mathbb{R}^{n \times d}$ differing only in a single row it holds that*

$$\mathbb{P}\left(\mathcal{A}_2\left(X\right) \in S\right) \leqslant e^{\varepsilon_2} \mathbb{P}\left(\mathcal{A}_2\left(X'\right) \in S\right) + \delta_2 \,.$$

*Further, let $\mathcal{A}_1 \colon \mathbb{R}^{n \times d} \to \mathbb{R}^{n \times d}$ be the Gaussian Mechanism with parameters $\Delta, \varepsilon_1, \delta_1$. I.e., on input $M$ it samples $\mathbf{W} \sim N\left(0, 2\Delta^2 \cdot \frac{\log(2/\delta_1)}{\varepsilon_1^2}\right)^{n \times d}$ and outputs $M + \mathbf{W}$.*

*Then for*

$$\varepsilon' := \varepsilon_1 + m\varepsilon_2 \,,$$
$$\delta' := e^{\varepsilon_1} m e^{(m-1)\varepsilon_2} \delta_2 + \delta_1 \,.$$

*$\mathcal{A}_2 \circ \mathcal{A}_1$ is $(\varepsilon', \delta')$-differentially private with respect to $V$ and $V'$, i.e., for all sets $\mathcal{S} \subseteq \mathcal{O}$ it holds that*

$$\mathbb{P}\left((\mathcal{A}_2 \circ \mathcal{A}_1)\left(V\right) \in S\right) \leqslant e^{\varepsilon'} \mathbb{P}\left((\mathcal{A}_2 \circ \mathcal{A}_1)\left(V'\right) \in S\right) + \delta' \,.$$

*Proof.* Without loss of generality, assume that $S = \{1, \ldots, m\}$. Denote by $V_1, V_2$ the first $m$ and last $n - m$ rows of $V$ respectively. Analogously for $V_1', V_2'$. We will later partitin the noise $\mathbf{W}$ of the Gaussian mechanism in the same way. Further, for a subset $A$ of $\mathbb{R}^{n \times n}$ and $Y \in \mathbb{R}^{m \times n}$ define

$$T_{A,Y} = \left\{ X \in \mathbb{R}^{(n-m) \times n} \; \middle| \; \begin{pmatrix} X \\ Y \end{pmatrix} \in A \right\} \subseteq \mathbb{R}^{(n-m) \times n} \,.$$

Note that $\begin{pmatrix} X \\ Y \end{pmatrix} \in A$ if and only if $X \in T_{A,Y}$.

Let $\mathcal{S} \subseteq \mathcal{O}$. It now follows that

$$\mathbb{P}_{\mathcal{A}_2, \mathbf{W}}\left[(\mathcal{A}_2 \circ \mathcal{A}_1)\left(V\right) \in S\right] = \mathop{\mathbb{E}}_{\mathcal{A}_2, \mathbf{W}}\left[\mathbb{1}\left\{V + \mathbf{W} \in \mathcal{A}_2^{-1}(S)\right\}\right]$$

$$= \mathop{\mathbb{E}}_{\mathcal{A}_2, \mathbf{W}_2}\left[\mathop{\mathbb{E}}_{\mathbf{W}_1}\left[\mathbb{1}\left\{\begin{pmatrix} V_1 + \mathbf{W}_1 \\ V_2 + \mathbf{W}_2 \end{pmatrix} \in \mathcal{A}_2^{-1}(S)\right\}\right] \,\middle|\, \mathbf{W}_2\right]$$

$$= \mathop{\mathbb{E}}_{\mathcal{A}_2, \mathbf{W}_2}\left[\mathop{\mathbb{E}}_{\mathbf{W}_1}\left[\mathbb{1}\left\{V_1 + \mathbf{W}_1 \in T_{\mathcal{A}_2^{-1}(S), V_2 + \mathbf{W}_2}\right\}\right] \,\middle|\, \mathbf{W}_2\right]$$

$$\leqslant e^{\varepsilon_1} \cdot \mathop{\mathbb{E}}_{\mathcal{A}_2, \mathbf{W}_2}\left[\mathop{\mathbb{E}}_{\mathbf{W}_1}\left[\mathbb{1}\left\{V_1' + \mathbf{W}_1 \in T_{\mathcal{A}_2^{-1}(S), V_2 + \mathbf{W}_2}\right\}\right] \,\middle|\, \mathbf{W}_2\right] + \delta_1$$

$$= e^{\varepsilon_1} \cdot \mathop{\mathbb{E}}_{\mathcal{A}_2, \mathbf{W}}\left[\mathbb{1}\left\{\begin{pmatrix} V_1' + \mathbf{W}_1 \\ V_2 + \mathbf{W}_2 \end{pmatrix} \in \mathcal{A}_2^{-1}(S)\right\}\right] + \delta_1 \,,$$

where the inequality follows by the guarantees of the Gaussian Mechanism. Further, we can bound

$$\mathop{\mathbb{E}}_{\mathcal{A}_2, \mathbf{W}}\left[\mathbb{1}\left\{\begin{pmatrix} V_1' + \mathbf{W}_1 \\ V_2 + \mathbf{W}_2 \end{pmatrix} \in \mathcal{A}_2^{-1}(S)\right\}\right] = \mathop{\mathbb{E}}_{\mathbf{W}}\left[\mathop{\mathbb{E}}_{\mathcal{A}_2}\left[\mathbb{1}\left\{\mathcal{A}_2\begin{pmatrix} V_1' + \mathbf{W}_1 \\ V_2 + \mathbf{W}_2 \end{pmatrix} \in S\right\} \,\middle|\, \mathbf{W}\right]\right]$$

$$\leqslant e^{m\varepsilon_2} \cdot \mathop{\mathbb{E}}_{\mathbf{W}}\left[\mathop{\mathbb{E}}_{\mathcal{A}_2}\left[\mathbb{1}\left\{\mathcal{A}_2\begin{pmatrix} V_1' + \mathbf{W}_1 \\ V_2' + \mathbf{W}_2 \end{pmatrix} \in S\right\} \,\middle|\, \mathbf{W}\right]\right] + m e^{(m-1)\varepsilon_2} \delta_2$$

$$= e^{m\varepsilon_2} \cdot \mathop{\mathbb{E}}_{\mathcal{A}_2, \mathbf{W}}\left[\mathbb{1}\left\{\begin{pmatrix} V_1' + \mathbf{W}_1 \\ V_2' + \mathbf{W}_2 \end{pmatrix} \in \mathcal{A}_2^{-1}(S)\right\}\right] + m e^{(m-1)\varepsilon_2} \delta_2 \,,$$

where the inequality follows by the privacy guarantees of $\mathcal{A}_2$ combined with standard group privacy arguments.

Putting the above two displays together and plugging in the definition of $\varepsilon', \delta'$ we finally obtain

$$\mathbb{P}_{\mathcal{A}_2, \mathbf{W}}\left[(\mathcal{A}_2 \circ \mathcal{A}_1)\left(V\right) \in S\right] \leqslant e^{\varepsilon'} \mathbb{P}_{\mathcal{A}_2, \mathbf{W}}\left[(\mathcal{A}_2 \circ \mathcal{A}_1)\left(V'\right) \in S\right] + \delta' \,.$$

$\square$

