# OpenReview forum: "Private estimation algorithms for stochastic block models and mixture models"
_NeurIPS.cc/2023/Conference — NeurIPS 2023 spotlight_

### Official Review · Reviewer_cXQY · 2023-07-02

**Soundness:** 3 good
**Presentation:** 3 good
**Contribution:** 4 excellent
**Rating:** 8
**Confidence:** 4

**Summary:**

In this paper, the authors investigate graph clustering with the Stochastic Block Model (SBM) and metric clustering with the Gaussian Mixture Model (GMM) within a novel privacy-preserving framework. They propose differentially private (DP) algorithms for both problems, which offer significantly improved performance guarantees compared to state-of-the-art DP algorithms.

Specifically, the authors present efficient  $(\epsilon,\delta)$-differentially private SBM algorithms for both weak recovery and exact recovery. These algorithms operate in polynomial time, outperforming the previous quasi-polynomial time DP algorithms. Additionally, they provide a lower bound that demonstrates the optimality of the privacy guarantees from an information-theoretic perspective.

Furthermore, the authors develop private algorithms for learning the mixture of $k$ spherical Gaussians under the condition that the minimum separation is at least $O(k^{1/t}\sqrt{t})$. Notably, these algorithms do not require an additional additive term of $\Omega(\sqrt{\log n})$ in the minimum separation or an explicit upper bound on the Euclidean norm of the centers, which were assumed in prior works.

The proposed algorithms are built upon a simple yet powerful connection between robustness and privacy. The authors formalize this connection as a framework for transforming robust estimation algorithms into private algorithms. They show that if two strongly convex functions over constrained sets, where both the function and the set may depend on the input, are point-wise close, their minimizers are also close. Conversely, they establish that projections of points that are close to each other onto convex sets that are point-wise close must also be close.

**Strengths:**

The paper is well-written and addresses fundamental clustering problems, including SBM and GMM, providing novel differentially private algorithms that make remarkable advancements in the field. The lower bound also demonstrates the optimality of the privacy guarantees of the SBM algorithm.

Moreover, their techniques of translating robust estimation algorithms to DP algorithms seem readily extended to some other estimation algorithms. Though the ideas are intuitively simple, the formal connections and applications to clustering algorithms are highly nontrivial.

**Weaknesses:**

There is no experiment in the paper (but it is a strong theory paper).

**Questions:**

No.

**Limitations:**

It might be useful to briefly discuss why the current framework does not achieve $(\log k)^{c+1/2}$ separation in polynomial time.

---

> ### Author Rebuttal · Authors · 2023-08-09
>
> We thank the reviewer for their constructive feedback and suggestions.
>
> **comment:**
> > It might be useful to briefly discuss why the current framework does not achieve $\log(k)^{1/2 + c}$ separation in polynomial time.
>
> **response:** On a high level, our framework converts non-private algorithms that are based on convex optimization into private ones. Currently, the only known (non-private) algorithm that can achieve separation $\log(k)^{1/2 + c}$ in polynomial time is not based on such convex optimization techniques (but rather intricate moment calculations). We believe it is likely, that there also exists an algorithm based on convex optimization that (non-privately) matches these guarantees. Given such an algorithm, we could likely use this in our framework and achieve the same guarantees privately.

---

> > ### Comment · Reviewer_cXQY · 2023-08-16
> >
> > Many thanks for your response. I will keep my score.

---

### Official Review · Reviewer_SDAx · 2023-07-06

**Soundness:** 4 excellent
**Presentation:** 4 excellent
**Contribution:** 4 excellent
**Rating:** 8
**Confidence:** 3

**Summary:**

This paper introduces a general framework for designing efficient private estimation algorithms from robust non-private algorithms. Their approach, based on a certain stability analysis for convex optimization, is applied to weak and exact recovery in the stochastic block model (SBM) and for learning mixtures of spherical Gaussians. They obtain the first efficient $(\varepsilon,\delta)$-private algorithms for estimation in the first setting (with a new lower bound suggesting that there is little room for improvement), as well as one for the second setting which allows for far lower separation between centers than before.

**Strengths:**

- The approach introduced in this paper appears very useful and flexible, and I anticipate future work building off this approach to address a variety of high-dimensional private estimation tasks
- They make significant progress on two important problems in private estimation
- Presentation is strong, prior work and guarantees are clearly delineated, their results are clearly stated, and, while there is a bit of technicality in the second application, the authors build up to it nicely with the first SBM results
- The Gaussian mixture application requires some clever sum-of-squares analysis. More broadly, there is certainly technical depth to this work beyond the general framework

**Weaknesses:**

- I sometimes lost track of the role of different parameters. I think some may be obvious to those more familiar with this subfield of privacy, and the many special cases mentioned in the discussion were helpful. In any extended version of this work, I would suggest adding more discussion of terms appearing in bounds from the classic (non-private) settings.
- The choice of objective function in the Gaussian mixture application wasn't clear to me - where does the all ones matrix come from?
- A fair bit of technical detail was pushed to the supplement, which was a bit tricky to parse. In particular, while the high level approach was clear to me, I was unable to verify correctness for the Gaussian mixture proofs (though I mostly blame this on my unfamiliarity with the SoS program they consider)

Minor nits:
- On lines 207-208, it was unclear to me why $k \geq \sqrt{\log d}$ should be considered a high-dimensional regime
- On line 272, the objective function $f$ is not defined (though clear enough from context)
- "the basic" repeated on line 360


**Questions:**

In addition to those above:
- Are the authors aware of their sensitivity analysis appearing in any previous convex programming literature? It is simple enough that I am a bit surprised it was not previously known. For instance, if their Condition (ii) was replaced with a Hausdorff distance bound between K(Y) and K(Y'), I think this is essentially folklore (though of course Condition (ii) is more sophisticated than this).

**Limitations:**

The authors were clear about all assumptions and I don't anticipate any negative impacts.

---

> ### Author Rebuttal · Authors · 2023-08-09
>
> We thank the reviewer for their constructive feedback and suggestions.
>
> **comment:**
> > The choice of objective function in the Gaussian mixture application wasn't clear to me - where does the all ones matrix come from?
>
> **response:** We choose the objective function $||W-J||\_F^2$ (where $J$ is the $n\times n$ all-one matrix) for both privacy and utility. Intuitively, $W_{ij}\in[0,1]$ indicates whether the $i$-th and $j$-th datapoints are sampled from the same Gaussian component.)
> Let $W^*\in\set{0,1}^{n\times n}$ be the ground-truth membership matrix. To recover $W^*$, we at least want $W$ to have (roughly) the same number of ones as $W^*$. One way to achieve this goal is to simply put an upper bound on $|\sum_{i,j}W_{ij}-W^*_{ij}|$. However, such a constraint would put a lower bound on $\sum_{i,j}W_{ij}$, which could cause privacy issues. Because there would be two neighboring datasets, such that the constraint set induced by one dataset is satisfiable, but the constraint set induced by the other dataset is not satisfiable. Thus we avoid lower bounding $\sum_{i,j}W_{ij}$ and instead minimize the distance between $W$ and the all-one matrix $J$. In this way, the constraint set induced by any dataset is satisfiable and the utility is kept.
>
> **comment:**
> > Are the authors aware of their sensitivity analysis appearing in any previous convex programming literature? It is simple enough that I am a bit surprised it was not previously known. For instance, if their Condition (ii) was replaced with a Hausdorff distance bound between $K(Y)$ and $K(Y')$, I think this is essentially folklore (though of course Condition (ii) is more sophisticated than this).
>
> **response:** We are aware of similar sensitivity analysis in the literature (e.g. [1,2,3]) and we believe our sensitivity analysis is not known before. (There is one paragraph comparing our work with previous works in our manuscript, but was removed from our submission due to page limitation.)
> More specifically, most previous results focus on the special case of unconstrained optimization of strongly convex functions. In contrast, our sensitivity bound applies to the significantly more general settings where both the objective functions and the constrained set may depend on the input.
> We will add back such a comparison in an updated version of our paper.
>
> [1] Kamalika Chaudhuri, Claire Monteleoni, and Anand D Sarwate, Differentially private empirical risk minimization., Journal of Machine Learning Research 12 (2011), no. 3.
>
> [2] Raef Bassily, Adam Smith, and Abhradeep Thakurta, Private empirical risk minimization: Efficient algorithms and tight error bounds, 2014 IEEE 55th annual symposium on foundations of computer science, IEEE, 2014, pp. 464–473.
>
> [3] Di Wang, Minwei Ye, and Jinhui Xu, Differentially private empirical risk minimization revisited: Faster and more general, Advances in Neural Information Processing Systems 30 (2017).
>
> **comment:**
> > - In any extended version of this work, I would suggest adding more discussion of terms appearing in bounds from the classic (non-private) settings.
> > - A fair bit of technical detail was pushed to the supplement, which was a bit tricky to parse.
> > - On lines 207-208, it was unclear to me why  should be considered a high-dimensional regime
> > - On line 272, the objective function is not defined (though clear enough from context)
> > - "the basic" repeated on line 360
>
> **response:** We will fix them in an updated version of our paper. We thank the reviewer for these helpful suggestions!

---

> > ### Comment · Reviewer_SDAx · 2023-08-10
> >
> > Thanks for providing these details. These sufficiently address the points I raised, and I intend to keep my score.

---

### Official Review · Reviewer_NQm5 · 2023-07-10

**Soundness:** 4 excellent
**Presentation:** 2 fair
**Contribution:** 4 excellent
**Rating:** 7
**Confidence:** 3

**Summary:**

The paper studies private parameter estimation in certain high-dimensional statistical inference tasks. In particular, community detection in stochastic block models (where the privacy is with respect to edge changes) and estimation of unknown means in isotropic Gaussian mixture models.
Although these problems have been studied in the literature, the paper provides new algorithms with improved sample and computational complexities, which closely match that of non-private estimation.
Although seemingly disjoint, the paper's results follow a common methodology of privatizing the existing non-private algorithms that rely on (strongly) convex optimization.


**Strengths:**

The paper proposes a generic privatization procedure relying on strongly convex optimization, and then the paper applies it to two problems of interest, leading to improved sample and computational complexities. The paper's results are novel and interesting.


**Weaknesses:**

I have the following comments:

1. The paper's writing needs to be improved and it seems that the submitted version is missing many details. For example, the notation for the indicator function is not displayed due to LaTeX error and many algorithmic environments are referenced as theorems by cleveref.


2. The comparison with work of Kothari-Manurangsi-Velingker is still not clear. Perhaps the paper should begin by explaining the similarities and then the differences.

Please also see the comments below.


**Questions:**

# Major Comments

1. I suggest adding the notion of privacy (node/sample differential privacy) in the theorem statements.

2. Page, Line 45: "However, for the more challenging task of graph clustering, node-DP is sometimes impossible to achieve."
    I do not follow. For the SBM, I believe that node-DP is possible to achieve since the information is not concentrated on a single node.

3. Footnote 11: I suggest explicitly adding the notion of privacy in the theorem statement (see the comment above as well).


4. Line 259: Where is the first step (i) from Line 253 used in the result here? Does it not follow from (ii) alone? Moreover, what if $K(Y)$ is empty?

5. Line 270, The convex program for SBM. The program from the citation [24] is different from the one displayed here. Especially, it is linear in X and not quadratic. Please clarify.

# Other Comments

1. Page 2, Line 49: What is meant by "non-robust setting"



3. Page 2, Line 64: I believe there are more representative papers for outlier robust estimation.

4. Page 3, Model 1.1: Please give intuition about the parameters in the model, for example, d is the average degree.

5. Theorem 1.3: Please add the weak recovery guarantee in the theorem statement.

6. Line 266: Describe explicitly the relation between the input and Y

3. Lines 182-188: I found the whole paragraph confusing. I suggest rephrasing.

4. Theorem 1.6: Why does $k$ need to be large?

6. Line 210: Put $t^*$ in the runtime or say that it is constant.

7. Line 250-251 Please add citations here "Our sensitivity bound is simple, yet it generalizes previously known bounds for strongly convex optimization problems"

8. Line 277: This was confusing and what is the norm $\|\cdot\|_1?$. Please rephrase and explain that the function $f$ does not contain the quadratic in $Y$. Although it does not affect the optimization problem, but it is important for the sensitivity analysis.

9. Line 312: "While this program suggests a path to recover the first property,..." Is it even feasible?

10. Line 321: Briefly explain that it is so by setting $z_i$ to zero.

11. Line 341-347: This was confusing for me. What histograms are being referred to here? I understand the space might have been an issue originally but the accepted version can be longer.

---

> ### Author Rebuttal · Authors · 2023-08-09
>
> We thank the reviewer for their constructive feedback and suggestions.
>
> **comment:**
> > Line 45: "However, for the more challenging task of graph clustering, node-DP is sometimes impossible to achieve." I do not follow. For the SBM, I believe that node-DP is possible to achieve since the information is not concentrated on a single node.
>
> **response:** What we wanted to convey is that *node-DP exact recovery is impossible to achieve*. The proof is as follows. For any graph $G$, we could obtain a neighboring graph $G'$ by picking an arbitrary vertex $v$ and removing all of its adjacent edges. Since vertex $v$ is isolated in $G'$, it is impossible to cluster this vertex correctly. By privacy, it is also impossible to cluster this vertex correctly in the original graph $G$. We will state this claim more accurately in an updated version of our paper.
>
> **comment:**
> > Line 259: Where is the first step (i) from Line 253 used in the result here? Does it not follow from (ii) alone? Moreover, what if $K(Y)$ is empty?
>
> **response:** Both step (i) and (ii) are used to show the inequality on Line 259. There is actually a typo on Line 254: $|f_Y(X)-f_{Y}(Z)|\leq\alpha$ should be $|f_Y(X)-f_{Y'}(Z)|\leq\alpha$. We will fix it in an updated version of our paper.
> The proof is in the proof of Lemma B.1 in the supplementary material of our submission. We reproduce it here. Let $Z\in K(Y)\cap K(Y')$ be a point such that $|f_Y(Z)-f_{Y'}(Z)|\leq \alpha$ by (i) and $|f_{Y}(\hat{X}) - f_{Y'}(Z)|\leq \alpha$ by (ii). Then $f_{Y}(Z) \leq f_{Y'}(Z)+\alpha \leq f_{Y}(\hat{X})+2\alpha$, where the first inequality follows from (i) and the second inequality follows from (ii). It is easy to see $f_{Y}(\hat{X})-f_{Y}(Z) \leq 0$ by optimality of $\hat{X}$. Thus $|f_{Y}(\hat{X})-f_{Y}(Z)| \leq 2\alpha$.
> Suppose without loss of generality $f_Y(\hat{X}) \leq f_{Y'}(\hat{X}')$, for a symmetric argument works in the other case. Then
> $f_{Y}(\hat{X}) \leq f_{Y'}(\hat{X}') \leq f_{Y'}(Z) \leq f_{Y}(\hat{X})+\alpha$,
> where the second inequality follows from the optimality of $\hat{X}'$ and the third inequality follows from (ii). Since $f_{Y'}(\hat{X}')$ and $f_{Y'}(Z)$ are sandwiched between $f_{Y}(\hat{X})$ and $f_{Y}(\hat{X})+\alpha$, then $|f_{Y'}(\hat{X}') - f_{Y'}(Z)| \leq \alpha$.
> Therefore, $|f_{Y}(\hat{X})-f_{Y}(Z)| + |f_{Y'}(\hat{X}') - f_{Y'}(Z)| \leq 3\alpha$.
>
> **comment:**
> > Line 270, The convex program for SBM. The program from the citation [24] is different from the one displayed here. Especially, it is linear in $X$ and not quadratic. Please clarify.
>
> **response:** The objective function in the citation [24] (i.e. GV [1]) is indeed linear in $X$, while the objective function of our program is quadratic in $X$. However, both programs have similar utility guarantees and the utility proof of our program is an adaption of that in GV. (See Lemma H.1 in the supplementary material of our submission.)
> The reason why we used the quadratic objective function (instead of the linear one in GV) is to achieve privacy via strong convexity.
>
> [1] Olivier Guédon and Roman Vershynin. Community detection in sparse networks via Grothendieck’s inequality. Probab. Theory Related Fields, 165(3-4):1025–1049, 2016.
>
>
> **comment:**
> > Line 312: "While this program suggests a path to recover the first property,..." Is it even feasible?
>
> **response:** More precisely, the 4-th constraint in $P(Y)$ on line 301 should be $(1-\alpha)n/k \leq \sum_{i}z_{i\ell} \leq (1+\alpha)n/k$ (for some appropriately chosen $\alpha$) instead of $\sum_{i}z_{i\ell}=n/k$. Here we were being a bit informal to make the discussion cleaner and more accessible. We will clarify it in an updated version of our paper.
>
>
> **comment:**
> > Line 341-347: What histograms are being referred to here?
>
> **response:** We consider the histogram in $\mathbb{R}^d$ where each bin is a $d$-dimensional cube.
> Formally, let $q\in\mathbb{R}$ and $\set{I_i}\_{i\in\mathbb{Z}}$ be a partition of $\mathbb{R}$ into intervals of length $b$, where $I_i:=\set{x\in\mathbb{R}: q+ (i-1) \cdot b\leq x< q + i\cdot b}$.
> Partition $\mathbb{R}^d$ into sets $\set{B_{i_1,\dots,i_d}}\_{i_1,\dots, i_d\in\mathbb{Z}}$ where $B_{i_1,\dots,i_d} := \set{x\in\mathbb{R}^d : \forall j\in [d], x_j \in I_{i_j}}$.
> We will clarify it in an updated version of our paper.
>
>
> **We will fix all the remaining issues pointed out by the reviewer in an updated version of our paper.**

---

> > ### Comment · Reviewer_NQm5 · 2023-08-10
> >
> > I thank the authors for their detailed response.
> >
> > Could you please also explain the similarities and dissimilarities with the Kothari-Manurangsi-Velingker paper in detail?

---

> > > ### Author Response · Authors · 2023-08-14
> > > **Comparison with Kothari-Manurangsi-Velingker**
> > >
> > > Both Kothari-Manurangsi-Velingker (KMV) and our work obtained private algorithms for high-dimensional statistical estimation problems by privatizing strongly convex programs, more specifically, sum-of-squares (SoS) programs. On a high level, we both first solve an SoS program, and then add Gaussian noise to the solution, and return the result.
> > >
> > > The main difference between KMV and our work lies in how we choose the SoS program. Such a difference saves us a computationally-expensive but necessary step for KMV.
> > > For the problem of robust moment estimation, KMV considered the canonical SoS program [1] which contains a minimum cardinality constraint. In robust statistics literature, such minimum cardinality constraints are added to ensure good utility, without privacy concerns [1]. However, from the perspective of privacy, such a constraint is problematic, since there will always exist two adjacent input datasets such that the constraints are satisfiable for one but not for the other. KMV and us resolve this privacy issue in different ways as follows:
> > > - KMV uses an exponential mechanism to pick the lower bound of the minimum cardinality constraint. This step also ensures that solutions to the resulting SoS program will have low sensitivity.
> > > - In contrast, we simply drop the minimum cardinality constraint. Then the resulting SoS program is always feasible for any input dataset! However, the resulting SoS program will not have good utility if we do nothing else. To have good utility, we need to pick an appropriate objective function. For example, in Gaussian mixture models, we dropped the constraint that $\sum_{j}W_{ij}$ should be lower bounded and turned to minimize $||W-J||_F^2$ (where $J$ is the all-one matrix).
> > >
> > > Our approach provides some advantages over KMV.
> > > From a purely algorithmic point of view, the exponential mechanism in KMV requires computing $O(n)$ scores. Computing each of these scores requires solving a large semidefinite program. In contrast, our algorithm completely avoids this step. This difference can provide significant improvement in the running time.
> > > From the point of view of algorithm analysis, proving that the exponential mechanism in KMV works requires several steps: 1) defining a (clever) score function, 2) bounding the sensitivity of this score function and, 3) showing existence of a large range of parameters with high score.
> > > Our approach completely bypasses these steps by directly bounding the sensitivity of the program via the key lemma, hence significantly simplifying the proofs.
> > >
> > > [1] Kothari, Pravesh K., and David Steurer. "Outlier-robust moment-estimation via sum-of-squares." _arXiv preprint arXiv:1711.11581_ (2017).

---

### Official Review · Reviewer_GVoS · 2023-08-01

**Soundness:** 4 excellent
**Presentation:** 4 excellent
**Contribution:** 3 good
**Rating:** 7
**Confidence:** 4

**Summary:**

The paper considers the problems of weak and strong recovery for stochastic block models, as well as the problem of estimating the means of Gaussian Mixture Models with sphrerical Gaussians, under the constraint of differential privacy. Their contributions include:

* A computationally efficient algorithm that performs strong recovery for the stochastic block model under $(\varepsilon, \delta)$-edge privacy.
* A computationally inefficient algorithm that performs both weak and strong recovery for the stochastic block model under $\varepsilon$-edge privacy, as well as a matching information-theoretic lower bound.
* A computationally efficient algorithm for estimating the means of mixtures of spherical Gaussians under $(\varepsilon, \delta)$-DP.

The algorithms are all derived by applying a general framework that is developed as part of the work which involves privatizing the solutions of strongly convex optimization programs. Additionally, the results given in the paper improve upon the prior work of Seif et. al. and Kamath et. al. on SBMs and GMMs, respectively.

**Strengths:**

As mentioned in my summary, the paper provides a general framework for privatizing the solutions of convex programs, which they apply in a multitude of settings. I consider this to be a significant contribution in the private statistics literature. In addition to that, the paper is very clearly written and the description of the techniques if quite accessible.

**Weaknesses:**

No concrete weaknesses.

**Questions:**

There has been recent work by Arbas et. al. on private estimation of GMMs (see https://arxiv.org/abs/2303.04288). I read both works and understand that they go in different directions (meaning that the work of Arbas et. al. doesn't supersede the present paper). However, unless I missed something the paper is not mentioned at all in the related work section (or anywhere else in the submitted pdfs), which feels like an important ommission. Thus, it'd be great if the authors could provide a comparison of their work with that paper (and also include it in their discussion of related work).

**Limitations:**

This is a purely theoretical work with no obvious societal impact.

---

> ### Author Rebuttal · Authors · 2023-08-09
>
> We thank the reviewer for their constructive feedback and suggestions.
>
> ## Comparison with Arbas-Ashtiani-Liaw (AAL)
> AAL [1] is a subsequent and independent work to ours. Their technique (as well as many previous papers, e.g. [2,3]) is based on the Subsample-And-Aggregate framework, while our approach is based on a different framework which directly privatizes convex optimization programs. We will provide a comparison with AAL [1] in an updated version of our paper.
>
> [1] Arbas, Jamil, Hassan Ashtiani, and Christopher Liaw. "Polynomial time and private learning of unbounded Gaussian Mixture Models." _arXiv preprint arXiv:2303.04288_ (2023).
>
> [2] Cohen, Edith, et al. "Differentially-private clustering of easy instances." _International Conference on Machine Learning_. PMLR, 2021.
>
> [3] Tsfadia, Eliad, et al. "Friendlycore: Practical differentially private aggregation." _International Conference on Machine Learning_. PMLR, 2022.

---

> > ### Comment · Reviewer_GVoS · 2023-08-18
> >
> > Thanks! I have no other questions. My score of course remains unchanged.

---

### Decision · Program_Chairs · 2023-09-21

**Decision:**

Accept (spotlight)

**Comment:**

This work develops sample efficient and (in some cases) computationally efficient algorithms for privately learning a number of probabilistic models, specifically mixtures of spherical Gaussians and stochastic block models. The reviewers agreed that the results and techniques are novel and interesting.  Overall, the paper is a solid contribution to this field.